# Robust group and simultaneous inferences for high-dimensional single index model

**Weichao Yang**
School of Statistics
Beijing Normal University, Beijing, China
yangweichao@mail.bnu.edu.cn

**Hongwei Shi**
School of Statistics
Beijing Normal University, Beijing, China
shihongwei21@mail.bnu.edu.cn

**Xu Guo** *
School of Statistics
Beijing Normal University, Beijing, China
xustat12@bnu.edu.cn

**Changliang Zou**
NITFID, School of Statistics and Data Science,
LPMC and KLMDASR and LEBPS,
Nankai University, Tianjin, China
zoucl@nankai.edu.cn

## Abstract

The high-dimensional single index model (SIM), which assumes that the response is independent of the predictors given a linear combination of predictors, has drawn attention due to its flexibility and interpretability, but its efficiency is adversely affected by outlying observations and heavy-tailed distributions. This paper introduces a robust procedure by recasting the SIM into a pseudo-linear model with transformed responses. It relaxes the distributional conditions on random errors from sub-Gaussian to more general distributions and thus it is robust with substantial efficiency gain for heavy-tailed random errors. Under this paradigm, we provide asymptotically honest group inference procedures based on the idea of orthogonalization, which enjoys the feature that it does not require the zero and nonzero coefficients to be well-separated. Asymptotic null distribution and bootstrap implementation are both established. Moreover, we develop a multiple testing procedure for determining if the individual coefficients are relevant simultaneously, and show that it is able to control the false discovery rate asymptotically. Numerical results indicate that the new procedures can be highly competitive among existing methods, especially for heavy-tailed errors.

## 1 Introduction

With fast development of information technologies, high-dimensional data are frequently collected in various scientific disciplines. To efficiently analyze high-dimensional data, various regularization methods were developed and thoroughly investigated in both theory and methodology. For a comprehensive review, see [19]. Though significant progress has been made about the theory of estimation and variable selection, statistical inference for a high-dimensional model remains largely unexplored until the seminal works of [58], [47] and [29]. They developed debiased estimators that correct the bias introduced by LASSO and established the estimators' asymptotic normality. [40]

---

*The first two authors contribute equally and are co-first authors. Xu Guo is the corresponding author. Guo was supported by National Key R & D Program of China (Grant Nos. 2023YFA1008702, 2023YFA1011100), National Natural Science Foundation of China Grant (Nos. 12322112, 12071038), and the Fundamental Research Funds for the Central Universities. Zou was supported by the National Key R & D Program of China (Grant Nos. 2022YFA1003703, 2022YFA1003800), the National Natural Science Foundation of China (Grant Nos. 11925106, 12231011, 11931001, 12226007, 12326325).

38th Conference on Neural Information Processing Systems (NeurIPS 2024).

proposed a decorrelated score method which is a general procedure and is applicable to a large family of penalized M-estimators. [45] introduced a recursive online-score estimation for high-dimensional generalized linear model. [46] proposed the nuisance penalized regression which does not penalize the parameters of interest. [21] and [20] studied inference problems for the high-dimensional Cox model and longitudinal data, respectively. Logistic regression model was investigated in [36]. Other recent developments include [44], [50], [5], and [19].

Most of current inference procedures focus on parametric regression models. To address the potential mis-specification issue of parametric models, the single index model (SIM) which is one of the most popular semiparametric modelling techniques, has received extensive attention and in-depth research in the past decade. Let $Y \in \mathbb{R}$ be the response variable along with predictor vector $\mathbf{X} = (X_1, \cdots, X_p)^\top \in \mathbb{R}^p$. We consider a general SIM,

$$Y = g(\mathbf{X}^\top \boldsymbol{\beta}, \epsilon) \quad \text{with } \epsilon \perp \mathbf{X}, \tag{1.1}$$

where $\boldsymbol{\beta} = (\beta_1, \cdots, \beta_p)^\top$, the link function $g(\cdot)$ is unknown and $\perp$ means independence. Equivalently, we have

$$Y \perp \mathbf{X} \mid \mathbf{X}^\top \boldsymbol{\beta}.$$

The above statement is that, given $\mathbf{X}^\top \boldsymbol{\beta}$ which is called index, the response $Y$ is independent of the predictor vector $\mathbf{X}$. Clearly, model (1.1) covers linear models, generalized linear models and also the classical SIM [26], that is, $Y = g_1(\mathbf{X}^\top \boldsymbol{\beta}) + \epsilon$ with $g_1(\cdot)$ being an unknown function. It relaxes restrictive assumptions on parametric models and is flexible enough to capture complex relationship between the response and the predictors.

For high-dimensional SIMs, variable selection has been considered by many authors. Examples include [30], [60], [57], [1], [51], [42], [52], [41] and [43]. Our primary interest is to detect whether a set of predictors contributes to the response $Y$ or not given the other predictors, that is testing the following group inference problem:

$$\mathbb{H}_{0,\mathcal{G}} : \beta_j = 0 \text{ for all } j \in \mathcal{G} \quad \text{versus} \quad \mathbb{H}_{1,\mathcal{G}} : \beta_j \neq 0 \text{ for some } j \in \mathcal{G}, \tag{1.2}$$

where $\mathcal{G}$ is a prespecified subset of $\{1, 2, \ldots, p\}$ with $p_0 = |\mathcal{G}|$. Such a hypothesis naturally arises in the high-dimensional setting. For example, researchers may want to test whether a gene pathway, consisting of multiple genes for the same biological functions, is important for certain clinical outcome The above group inference problem also includes global significance testing as a special case when we set $\mathcal{G} = \{1, 2, \ldots, p\}$ and $p_0 = p$. The group inference problem is more difficult than purely global significance testing. Actually, for group inference problem, we are dealing with a high-dimensional interested parametric vector with a high-dimensional nuisance parameter. Such a group inference problem was considered by [59] for high-dimensional linear models. See also [16], and [23]. All of [59], [16], and [23] did not establish asymptotical honesty of their procedures, which is critical for reliable inference in high-dimensional setting.

For SIM, due to the existence of the unknown link function $g(\cdot)$, the group inference problem is more difficult. Actually, the parameter of interest and the nonparametric function are bundled together, which makes the inference being challenging. For the model (1.1), [17] developed a debiased LASSO procedure for individual coefficient, while [53] considered simultaneous confidence interval for optimal treatment regimes under the classical SIM [26]. Their procedures, as well as many other existing works, rely heavily on the sub-Gaussian assumption about the error term. In practice, data with heavy-tailed distribution or in the presence of outliers are very common [18], and the efficiency of those procedures developed under sub-Gaussian assumption would be largely deteriorated. Recently [25] investigated the robust inference problem of high-dimensional SIM by adopting the Huber loss with original response. However, their procedure still requires bounded first moment condition of the error term and needs to select a suitable robustification parameter. In this paper, we aim to develop robust group inference methods for high-dimensional SIM without any moment condition on the error term and without choosing robustification parameter.

Interestingly, under a mild condition on the predictors, which is called linearity condition in the literature of sufficient dimension reduction [31], we find that the SIM can be recast into a pseudo-linear model with transformed response, allowing us to make robust inference in a simple fashion. With a specific "response-distribution" transformation, the sub-Gaussian response assumption required in existing methods is totally avoided, and accordingly our proposed procedures are robust to outliers or heavy-tailed error distributions.

Besides group inference of a prespecified subset of predictors, we are also interested in identifying relevant predictors. To this aim, large-scale simultaneous hypotheses are considered

$$\mathbb{H}_{0j} : \beta_j = 0 \qquad \text{versus} \qquad \mathbb{H}_{1j} : \beta_j \neq 0, \quad 1 \leq j \leq p.$$

Apart from identifying as many nonzero $\beta_j$ as possible, to obtain results with uncertainty quantification, we would like to control the false discovery rate (FDR) which is an extremely popular tool to maintain the ability to reliably detect true alternatives without excessive false positive results when large-scale hypotheses are simultaneously tested [4]. [28], [20], [36] and [9] considered FDR control in high-dimensional regression models. We follow this line and develop a FDR control procedure for high-dimensional SIM.

Our major contributions are listed from the following three aspects.

- We extend the rank-LASSO procedure in [43] to include both convex and non-convex penalties and establish error bound of any local optimum of the empirical objective. These theoretical results are summarized in subsection 3.1.

- In Section 2, we provide asymptotically honest group inference procedures based on the idea of orthogonalization for testing the joint effect of many predictors, which enjoys the feature that it does not require the zero and nonzero coefficients to be well-separated. We demonstrate the superiority of our test procedures both theoretically and empirically. Please see Section 3 for theoretical justification and Section 4 for numerical studies respectively.

- We develop a multiple testing procedure for determining if the individual coefficients are relevant simultaneously, and show that it is able to control the FDR asymptotically. To this end, we develop suitable multiple testing procedures and show that the proposed methods can control the false discovery rate (FDR, [4]) both theoretically and empirically. We refer the readers to Appendix A.3 and A.6 for details.

**Notation.** For a $d$-dimension vector $\mathbf{U}$, we write $\|\mathbf{U}\|_r = (\sum_{k=1}^{d} U_k^r)^{1/r}$ and $\|\mathbf{U}\|_\infty = \max_{1 \leq k \leq d} |U_k|$ to denote $l_r$ and $l_\infty$ norms of $\mathbf{U}$. Further we define $\|\mathbf{U}\|_0 = \#\{k : U_k \neq 0\}$. A random variable $X$ is *sub-Gaussian* if the moment generating function (MGF) of $X^2$ is bounded at some point, namely $\mathbb{E} \exp(X^2/K^2) \leq 2$, where $K$ is a positive constant. A random variable $Y$ is *sub-Exponential* if the MGF of $|Y|$ is bounded at some point, namely $\mathbb{E} \exp(|Y|/K') \leq 2$, where $K'$ is a positive constant. For $a, b \in \mathbb{R}$, we write $a \vee b = \max\{a, b\}$.

## 2 Group inference based on distribution transformation

Without loss of generality, assume that $\mathbb{E}(\mathbf{X}) = \mathbf{0}$, and $\boldsymbol{\Sigma} = \text{Var}(\mathbf{X}) > 0$. Now let $\boldsymbol{\sigma}_h = \text{Cov}\{\mathbf{X}, h(Y)\}$ for a given transformation function $h(\cdot)$ of the response. Define $\boldsymbol{\beta}_h = \boldsymbol{\Sigma}^{-1} \boldsymbol{\sigma}_h$. We then have the following result.

**Proposition 2.1.** *Assume that $\mathbb{E}(\mathbf{X} \mid \mathbf{X}^\top \boldsymbol{\beta})$ is a linear function of $\mathbf{X}^\top \boldsymbol{\beta}$. Then $\boldsymbol{\beta}_h$ is proportional to $\boldsymbol{\beta}$, that is, $\boldsymbol{\beta}_h = \kappa_h \times \boldsymbol{\beta}$ for some constant $\kappa_h$.*

The above proposition follows directly from Theorem 2.1 in [32]. The assumption in above proposition is known as linearity condition (LC) for predictors. It is satisfied when $\mathbf{X}$ has an elliptical distribution and widely assumed in the sufficient dimension reduction literature [31, 14]. [24] showed that in high-dimensional setting, the LC holds to a reasonable approximation. Throughout of the paper, we assume that $\kappa_h \neq 0$. This assumption is mild. In fact, when $h(\cdot)$ is monotone and $g(\cdot, \cdot)$ is monotone with respect to the first argument, this assumption is satisfied.

From the definition of $\boldsymbol{\beta}_h$, we can write

$$h(Y) = \boldsymbol{\beta}_h^\top \mathbf{X} + e, \tag{2.3}$$

where the error term $e$ must satisfy $\mathbb{E}(e\mathbf{X}) = 0$. Different from the existing literature which usually imposes independence between the regression error and the predictors, the regression error $e$ in the transformed model is only uncorrelated with the predictors. This implies that under LC, we can recast the general SIM into a pseudo-linear model with transformed response $h(Y)$. Therefore, by Proposition 2.1, testing $\mathbb{H}_{0,\mathcal{G}}$ is equivalent to

$$\mathbb{H}'_{0,\mathcal{G}} : \beta_{hj} = 0 \text{ for all } j \in \mathcal{G} \quad \text{versus} \quad \mathbb{H}'_{1,\mathcal{G}} : \beta_{hj} \neq 0 \text{ for some } j \in \mathcal{G}.$$

This reformulation is important, allowing us to circumvent the issue of estimating the unknown link function $g(\cdot, \cdot)$ and to make inference of $\beta_{hj}, j \in \mathcal{G}$ in a linear regression model with transformed response instead of $\beta_j$ in SIM. As we show later, this reformulation greatly facilitates the construction of our test statistic and simplifies the computation as well.

In practice, we need to choose a suitable transformation function $h(\cdot)$. We note that [17]'s procedure also relies on the above proposition and they essentially work with $h(Y) = Y$. For robustness consideration, throughout this paper, we consider the distribution function of $Y$, denoted by $F(Y)$ as the transformation function. Actually with the equation (2.3), given the widely imposed subgaussian assumption on the predictors, any bounded transformation function $h(Y)$ would lead the transformed error term $e$ being subgaussian, even if the original error term $\epsilon$ in the single index model $Y = g(\boldsymbol{X}^\top \boldsymbol{\beta}, \epsilon)$ comes from Cauchy distribution. Further as noted by [43], in the empirical distribution function, the term $\sum_{j=1}^n I(Y_j \leq Y_i)$ is the rank of $Y_i$. Since statistics with ranks such as Wilcoxon test and the Kruskall-Wallis ANOVA test, are well-known to be robust, this then intuitively explains why our procedures with response-distribution transformation are robust with respect to outliers in response. Moreover the distribution function is very easy to estimate and thus our approach is straightforward to implement and understand.

Our test statistic for group inference of $\mathbb{H}'_{0,\mathcal{G}}$ relies on individual inference of $\mathbb{H}'_{0j} : \beta_{hj} = 0$. Now we first consider individual hypothesis $\mathbb{H}'_{0j}$. Let $\mathbf{Z}_j$ be the subvector of $\mathbf{X}$ without $X_j$, and $\boldsymbol{\gamma}_j$ be the subvector of $\boldsymbol{\beta}_h$ without $\beta_{hj}$. Suppose that $\{\mathbf{X}_i, Y_i\}_{i=1}^n$ is a random sample from the population $(\mathbf{X}, Y)$. Similarly, we denote $\mathbf{Z}_{ij}$ as the sample of $\mathbf{Z}_j$. Define

$$\boldsymbol{\beta}_h = \mathbb{E}(\mathbf{X}\mathbf{X}^\top)^{-1}\mathbb{E}[\mathbf{X}\{F(Y) - 1/2\}] \quad \text{and} \quad \boldsymbol{\theta}_j = \mathbb{E}(\mathbf{Z}_j\mathbf{Z}_j^\top)^{-1}\mathbb{E}(\mathbf{Z}_j X_j),$$

where $\boldsymbol{\beta}_h$ is the regression coefficient of the model (2.3) with $h = F$.

Our approach is based on the idea of orthogonalization. The main idea of orthogonalization is to construct a statistic for target parameter which is locally insensitive to the nuisance parameters. Dealing with high-dimensional models, it plays an important role to make the statistic of target parameter immune to the bias from the estimators of high-dimensional nuisance parameters, which in turn enables the statistical inference of parameter of interest. For relevant references, see for example, [3], [40] and [2].

As pointed by [53], the adoption of orthogonalization is nontrivial for high-dimensional semiparametric setting, particularly for index model, where the challenge of bundled parameter arises. Fortunately, with the Proposition 2.1, the SIM can be recast into a pseudo-linear model with transformed response (2.3). Note that under the null hypothesis $\mathbb{H}_{0j}$

$$\mathbb{E}\left[\left\{F(Y) - 1/2 - \mathbf{Z}_j^\top \boldsymbol{\gamma}_j\right\}\left(X_j - \mathbf{Z}_j^\top \boldsymbol{\theta}_j\right)\right] = 0.$$

Further the above equation has the orthogonality property

$$\frac{\partial}{\partial \boldsymbol{\gamma}_j}\mathbb{E}\left[\left\{F(Y) - 1/2 - \mathbf{Z}_j^\top \boldsymbol{\gamma}_j\right\}\left(X_j - \mathbf{Z}_j^\top \boldsymbol{\theta}_j\right)\right] = \mathbb{E}\left[\mathbf{Z}_j\left(X_j - \mathbf{Z}_j^\top \boldsymbol{\theta}_j\right)\right] = 0.$$

This then motivates us to consider the following quantity

$$T_{nj}^* = \frac{1}{\sqrt{n}}\sum_{i=1}^n \left\{F(Y_i) - 1/2 - \mathbf{Z}_{ij}^\top \boldsymbol{\gamma}_j\right\}\left(X_{ij} - \mathbf{Z}_{ij}^\top \boldsymbol{\theta}_j\right). \tag{2.4}$$

In practice, we need to use suitable estimates of $F$, $\boldsymbol{\gamma}_j$ and $\boldsymbol{\theta}_j$ in (2.4) as those quantities are unknown. Naturally, $F_n(y) = n^{-1}\sum_{i=1}^n I(Y_i \leq y)$, the empirical distribution of $Y_1, \ldots, Y_n$, can be used to estimate $F(Y)$. For $\boldsymbol{\theta}_j$, we estimate it by the penalized least-squares method

$$\hat{\boldsymbol{\theta}}_j = \arg\min_{\boldsymbol{\theta}_j \in \mathbb{R}^{p-1}} \frac{1}{2n}\sum_{i=1}^n \left(X_{ij} - \mathbf{Z}_{ij}^\top \boldsymbol{\theta}_j\right)^2 + \sum_{l=1}^{p-1} p_{\lambda_X}(|\theta_{jl}|), \tag{2.5}$$

where $p_{\lambda_X}(\cdot)$ is a penalty function with a tuning parameter $\lambda_X$. Similarly, for $\boldsymbol{\gamma}_j$, we adopt the following penalized least-squares

$$\hat{\boldsymbol{\beta}}_h = \arg\min_{\boldsymbol{\beta}_h \in \mathbb{R}^p} \mathcal{L}_n(\boldsymbol{\beta}_h) + \sum_{l=1}^p p_{\lambda_Y}(|\beta_{hl}|), \tag{2.6}$$

where $\mathcal{L}_n(\boldsymbol{\beta}_h) = (2n)^{-1} \sum_{i=1}^n \{F_n(Y_i) - 1/2 - \mathbf{X}_i^\top \boldsymbol{\beta}_h\}^2$ and $p_{\lambda_Y}(\cdot)$ is a penalty function with a tuning parameter $\lambda_Y$. The $\hat{\boldsymbol{\gamma}}_j$ is then set as the subvector of $\hat{\boldsymbol{\beta}}_h$ without $\hat{\beta}_{hj}$. Accordingly, for each individual hypothesis $\mathbb{H}'_{0j} : \beta_{hj} = 0$ we define the standardized test statistic

$$\widetilde{T}_{nj} = \frac{1}{\hat{\sigma}_j \sqrt{n}} \sum_{i=1}^n \left\{ F_n(Y_i) - 1/2 - \mathbf{Z}_{ij}^\top \hat{\boldsymbol{\gamma}}_j \right\} (X_{ij} - \mathbf{Z}_{ij}^\top \hat{\boldsymbol{\theta}}_j), \tag{2.7}$$

where

$$\hat{\sigma}_j^2 = \frac{1}{n} \sum_{i=1}^n \left\{ (X_{ij} - \mathbf{Z}_{ij}^\top \hat{\boldsymbol{\theta}}_j) \hat{e}_{ij} + \hat{m}_j(Y_i) \right\}^2, \tag{2.8}$$

$\hat{e}_{ij} = F_n(Y_i) - 1/2 - \mathbf{Z}_{ij}^\top \hat{\boldsymbol{\gamma}}_j$ and $\hat{m}_j(y) = n^{-1} \sum_{i=1}^n (X_{ij} - \mathbf{Z}_{ij}^\top \hat{\boldsymbol{\theta}}_j) \{I(Y_i \geq y) - F_n(Y_i)\}$. Note that $F_n(Y_i) = n^{-1} \sum_{j=1}^n I(Y_j \leq Y_i)$. Then given predictors $\mathbf{X}_i$'s being fixed, perturbations in the responses would not make the value of $\widetilde{T}_{nj}$ change as long as the ranks of $Y_i$'s remain unchanged.

Denote $\widetilde{\boldsymbol{T}}_{n,\mathcal{G}} = (\widetilde{T}_{nj})_{j \in \mathcal{G}}$. To test the null hypothesis $\mathbb{H}_{0,\mathcal{G}}$, we consider test statistic based on the max norm of $\widetilde{\boldsymbol{T}}_{n,\mathcal{G}}$. That is,

$$M_{n,\mathcal{G}} = \max_{j \in \mathcal{G}} \widetilde{T}_{nj}^2. \tag{2.9}$$

Based on the limiting null distribution obtained in subsection 3.2, we can reject null hypothesis $\mathbb{H}_{0,\mathcal{G}}$ at the significant level $\alpha$ if and only if $M_{n,\mathcal{G}} \geq c_{\mathcal{G}}(\alpha)$, where $c_{\mathcal{G}}(\alpha) = 2 \log p_0 - \log \log p_0 + q_\alpha$ and $q_\alpha$ is the $1 - \alpha$ quantile of the Gumbel distribution with the cumulative distribution function $\exp\{-\frac{1}{\sqrt{\pi}} \exp(-x/2)\}$, that is,

$$q_\alpha = -\log(\pi) - 2 \log \log(1 - \alpha)^{-1}. \tag{2.10}$$

Our inference procedure is summrized in the Algothrim 1 as follows.

---

**Algorithm 1:** Group inference based on distribution transformation

---

**Input:** Covariates data $\{\mathbf{X}_i\}_{i=1}^n$, response data $\{Y_i\}_{i=1}^n$.
**Output:** Testing results for group inference problem (1.2).

1   Compute the penalized least-square estimator $\hat{\boldsymbol{\beta}}_h$ defined in (2.6);
2   **for** $j \in \mathcal{G}$ **do**
3      compute estimator $\hat{\boldsymbol{\gamma}}_j$ and $\hat{\boldsymbol{\theta}}_j$ defined in (2.5);
4      calculate the standard statistic $\widetilde{T}_{nj}$ defined in (2.7);
5   calculate the test statistic $M_{n,\mathcal{G}}$ defined in (2.9) ;
6   if $M_{n,\mathcal{G}} \geq c_{\mathcal{G}}(\alpha)$, reject $\mathbb{H}_{0,\mathcal{G}}$; otherwise accept $\mathbb{H}_{0,\mathcal{G}}$.

---

**Remark 2.1.** *The use of quantiles of limiting null distribution in $c_{\mathcal{G}}(\alpha)$ is attractive from a computational point of view. On the other hand, the validity of limiting null distribution requires additional assumptions regarding the dependence structure of the components of the covariates $\mathbf{X}$. See Assumption A.5 in Appendix A.8 for details. Besides, it is well known that this weak convergence is typically slow. To solve these problems, we propose a multiplier bootstrap approach and show its validity in theory, please see Appendix A.1 for details.*

## 3 Theoretical properties

### 3.1 Estimation error bound

The theoretical analysis of $\hat{\boldsymbol{\beta}}_h$ requires a substantial modification of the proof technique as compared to existing works on high-dimensional inference. It is related to the fact that empirical distribution $F_n(Y_i)$ are dependent, and thus $\mathcal{L}_n(\boldsymbol{\beta}_h)$ is a sum of dependent random variables. Note that [43] considered a similar penalized procedure with LASSO penalty. While [60] and [51] considered non-convex penalties such as SCAD and MCP to reduce estimation bias incurred by LASSO. However,

their results only allow the dimension to be polynomial order of the sample size and thus cannot work in ultrahigh-dimensional setting. Further their results only provide guarantees for global optima. In this paper, we consider both convex and non-convex penalties and establish error bound of any local optimum of the empirical objective.

To be specific, assume that $\hat{\boldsymbol{\beta}}_h$ satisfies the first-order necessary condition to be a local minimum of 2.6, that is,

$$\left\langle \nabla \mathcal{L}_n(\hat{\boldsymbol{\beta}}_h) + \nabla p_{\lambda_Y}(\hat{\boldsymbol{\beta}}_h), \boldsymbol{\beta} - \hat{\boldsymbol{\beta}}_h \right\rangle \geq 0, \quad \text{for all feasible } \boldsymbol{\beta} \in \mathbb{R}^p. \tag{3.11}$$

Let $s_Y$ be the sparsity level for $\boldsymbol{\beta}_h$, i.e., $s_Y = \|\boldsymbol{\beta}_h\|_0$. For any vector $\hat{\boldsymbol{\beta}}_h$ satisfying the condition (3.11), we have the following result.

**Theorem 3.1.** *Under Assumptions A.1 and A.2 in Appendix A.8, the $\hat{\boldsymbol{\beta}}_h$ defined in* (3.11) *satisfies*

$$\|\hat{\boldsymbol{\beta}}_h - \boldsymbol{\beta}_h\|_2 \leq c\lambda_Y \sqrt{s_Y} \quad and \quad \|\hat{\boldsymbol{\beta}}_h - \boldsymbol{\beta}_h\|_1 \leq c'\lambda_Y s_Y$$

*with probability at least* $1 - c_1 \exp(-c_2 \log p)$. *Here,* $(c, c', c_1, c_2)$ *are universal constants and* $\lambda_X, \lambda_Y \asymp \sqrt{\log p/n}$.

In Theorem 3.1 we establish error bound of penalized least-squares estimators with empirical distribution function of the response. For consistency in $L_2$-loss, we require the sparsity level $s_Y$ satisfy that $s_Y = o(n/\log p)$. While in terms of $L_1$-loss, the limitation becomes $s_Y = o(\sqrt{n/\log p})$. Clearly the sparsity level is allowed to be diverging. Our results unify both convex [43] and non-convex penalties [60, 51]. Compared with the results in [60] and [51], the dimension can be exponential order of the sample size, and no minimal signal condition is imposed to obtain those error bounds. In the proof, we modify Hoeffding's inequality by a probability inequality for $U$-statistic to handle tail probability in dependent case, which may be interesting in their own rights.

## 3.2 Asymptotic null distribution

In this subsection, we derive the asymptotic null distribution for statistic defined in (2.9). Denote $\Omega = \{\boldsymbol{\beta}_h \in \mathbb{R}^p : \|\boldsymbol{\beta}_h\|_0 \vee \max_{1 \leq j \leq p}\|\boldsymbol{\theta}_j\|_0 \leq s\}$ and $\boldsymbol{\beta}_{h\mathcal{G}} = (\beta_{hj})_{j \in \mathcal{G}}$. We consider the following parameter space for $\mathbb{H}_{0,\mathcal{G}}$

$$\Omega_{\mathcal{G}}^0 = \{\boldsymbol{\beta}_h \in \mathbb{R}^p : \boldsymbol{\beta}_{h\mathcal{G}} = \mathbf{0}\} \cap \Omega.$$

**Theorem 3.2.** *Suppose that Assumptions A.1-A.6 in Appendix A.8 and LC condition hold. If* $s = o(\sqrt{n}/(\log p \log p_0))$, *then for given* $t \in \mathbb{R}$ *we have*

$$\lim_{(n,p_0)\to\infty} \sup_{\boldsymbol{\beta}_h \in \Omega_{\mathcal{G}}^0} \left| \Pr\left(M_{n,\mathcal{G}} - 2\log p_0 + \log\log p_0 \leq t\right) - \exp\left\{-\frac{1}{\sqrt{\pi}}\exp(-t/2)\right\} \right| = 0.$$

Theorem 3.2 implies that the type I error of the proposed test statistic $M_{n,\mathcal{G}}$ converges to any pre-specified significance level uniformly over $\boldsymbol{\beta}_h \in \Omega_{\mathcal{G}}^0$. The hypothesis test with such uniform convergence property is called *honest* test. The advantage of honest test is that the limiting distribution of our procedure is uniformly valid over $s$-sparse high-dimensional models despite the possible imperfect model selection via estimator $\hat{\boldsymbol{\beta}}_h$. An immediate implication is that it relaxes the assumption on signal strength and does not require the zero and nonzero effects to be well-separated. In particular, this procedure does not require the initial estimator to select zero and nonzero signals perfectly, which is nearly impossible in practice. Due to these excellent statistical properties, honest test has recently drawn lots of attention. See [47], [29], [3], [2], [11], [53] for further examples.

On the basis of Theorem 3.2 , we can reject null hypothesis $\mathbb{H}_{0,\mathcal{G}}$ at the significant level $\alpha$ if and only if $M_{n,\mathcal{G}} \geq c_{\mathcal{G}}(\alpha)$, where $c_{\mathcal{G}}(\alpha) = 2\log p_0 - \log\log p_0 + q_\alpha$ and $q_\alpha$ is defined in (2.10).

## 3.3 Power analysis

We next to consider the asymptotic power analysis of the $M_{n,\mathcal{G}}$. In this section, we show that our test statistic is powerful in the sense that the separation rate is of order $\sqrt{(2+\epsilon_0)\log p_0/n}$ for some $\epsilon_0 > 0$ when $p_0 \to \infty$. At the beginning we define the following parameter space for $\mathbb{H}_{1,\mathcal{G}}$

$$\Omega_{\mathcal{G}}^1(c_0) = \left\{ \boldsymbol{\beta}_h \in \mathbb{R}^p : \max_{j \in \mathcal{G}} \left| \frac{\beta_{hj}\delta_j}{\sigma_j} \right| \geq \sqrt{c_0 \frac{\log p_0}{n}} \right\} \cap \Omega,$$

where $c_0$ is a positive constant and $\delta_j = \mathbb{E}(X_j^2) - \mathbb{E}(X_j \mathbf{Z}_j^\top)\mathbb{E}(\mathbf{Z}_j \mathbf{Z}_j^\top)^{-1}\mathbb{E}(\mathbf{Z}_j X_j)$.

**Theorem 3.3.** *Suppose that conditions in Theorem 3.2 are satisfied, we have for some $\epsilon_0 > 0$,*

$$\lim_{(n,p_0)\to\infty} \inf_{\boldsymbol{\beta}_h \in \Omega_{\mathcal{G}}^1(2+\epsilon_0)} \Pr\big(M_{n,\mathcal{G}} \geq c_{\mathcal{G}}(\alpha)\big) = 1. \tag{3.12}$$

Theorem 3.3 implies that our test procedure can still be powerful even when there exists few components of $\boldsymbol{\beta}_{\mathcal{G}}$ with a magnitude being larger than $\sqrt{(2+\epsilon_0)\log p_0/n}$. Therefore our testing procedure is powerful against "sparse" alternative aforementioned. This separation rate is widely discussed in the literature, such as [8], [59] and [36]. Theorem 3.3 shows our tests based on distribution transformation can also achieve this lower bound.

**Remark 3.1.** *In the proof of Theorem 3.3, we show that*

$$M_{n,\mathcal{G}} \asymp M_{\mathcal{G}}^0 + \max_{j\in\mathcal{G}}\left(\frac{\beta_{hj}\delta_j}{\sigma_j}\right)^2,$$

*where the $M_{\mathcal{G}}^0$ has the type I extreme limiting distribution, which is mentioned in Theorem 3.2. Then the power is largely determined by the second term*

$$\mathrm{SNR}(M_{n,\mathcal{G}}) = \max_{j\in\mathcal{G}}\left(\frac{\beta_{hj}\delta_j}{\sigma_j}\right)^2, \tag{3.13}$$

*and it can be explained as the maximum of the signal-to-noise ratio for $j \in \mathcal{G}$. The power of the test based on $M_{n,\mathcal{G}}$ increases with the growth of $\mathrm{SNR}(M_{n,\mathcal{G}})$. When $\mathrm{SNR}(M_{n,\mathcal{G}}) \geq (2+\epsilon_0)\log p_0/n$, the asymptotic power is tending to 1.*

In this paper, we focus on the distribution transformation of the response. In the following we make some comparisons between this choice and another natural one. Actually, same to [17], one may consider use $h(Y) = Y$ to conduct the following test statistic

$$\widetilde{T}_{nj}^+ = \frac{1}{\hat{\sigma}_j^+ \sqrt{n}} \sum_{i=1}^n (Y_i - \mathbf{Z}_{ij}^\top \hat{\boldsymbol{\gamma}}_j^+)(X_{ij} - \mathbf{Z}_{ij}^\top \hat{\boldsymbol{\theta}}_j). \tag{3.14}$$

Here $\hat{\boldsymbol{\gamma}}_j^+$ is the subvector of $\hat{\boldsymbol{\beta}}_h^+$ without $j$-th component, and $\hat{\boldsymbol{\beta}}_h^+$ is a penalized least-squares estimator of $\boldsymbol{\beta}_h^+ = \mathbb{E}(\mathbf{X}\mathbf{X}^\top)^{-1}\mathbb{E}(\mathbf{X}Y)$ as follows:

$$\hat{\boldsymbol{\beta}}_h^+ = \arg\min_{\boldsymbol{\beta}_h^+ \in \mathbb{R}^p} \frac{1}{2n}\sum_{i=1}^n (Y_i - \mathbf{X}_i^\top \boldsymbol{\beta}_h^+)^2 + \sum_{l=1}^p p_{\lambda_Y}(|\beta_{hl}^+|).$$

$\hat{\boldsymbol{\theta}}_j$ is estimated by (2.5). And $\hat{\sigma}_j^{+2}$ is an appropriate estimator of $\sigma_j^{+2} := \mathbb{E}\{(X_j - \mathbf{Z}_j^\top \boldsymbol{\theta}_j)^2 (Y - \mathbf{Z}_j^\top \boldsymbol{\gamma}_j^+)^2\}$. For the group inference problem (1.2), it's natural to consider the following testing statistic:

$$M_{n,\mathcal{G}}^+ = \max_{j\in\mathcal{G}}(\widetilde{T}_{nj}^+)^2,$$

Now we discuss the power properties of $M_{n,\mathcal{G}}^+$. By similar arguments as Theorem 3.3, the power of the test based on $M_{n,\mathcal{G}}^+$ is determined by

$$\mathrm{SNR}(M_{n,\mathcal{G}}^+) = \max_{j\in\mathcal{G}}\left(\frac{\beta_{hj}^+\delta_j}{\sigma_j^+}\right)^2. \tag{3.15}$$

For the convenience of illustration, we consider the classical SIM, that is, $Y = g_1(\mathbf{X}^\top\boldsymbol{\beta}) + \epsilon$ and $\epsilon$ is independent of $\mathbf{X}$. To compare the theoretical power of $M_{n,\mathcal{G}}$ and $M_{n,\mathcal{G}}^+$, it suffices to compare the SNRs aforementioned. In the classical SIM case, the ratio of SNR can be simplified as

$$\frac{\mathrm{SNR}(M_{n,\mathcal{G}})}{\mathrm{SNR}(M_{n,\mathcal{G}}^+)} \geq c_3 \mathrm{Var}(\epsilon),$$

where $c_3$ is a constant depending on $(\mathbf{X}, Y)$. The detailed form of $c_3$ is presented in the Appendix A.2. When $\mathrm{Var}(\epsilon) > c_3^{-1}$, $M_{n,\mathcal{G}}$ is more powerful than $M_{n,\mathcal{G}}^+$. Actually when $\epsilon$ follows a heavy-tail distribution, $\mathrm{Var}(\epsilon)$ can be very large even be infinite.

The above results illustrate the robustness of our test statistic $M_{n,\mathcal{G}}$. In fact, with distribution transformation, no moment assumption on $\epsilon$ is required. Even if the $\epsilon$ comes from Cauchy distribution, our test procedure still works well but the $M_{n,\mathcal{G}}^+$ requires sub-Gaussian assumption on the error term and thus in this situation would fail.

## 4 Numerical studies

In this section, extensive simulation studies are carried out to evaluate the numerical performance of the proposed methods for the global inference problem described in (1.2). We consider the high-dimensional SIM in equation (1.1) and generate the data from the following models:

**Model 1**: Linear model: $Y = \mathbf{X}^\top \boldsymbol{\beta} + \epsilon$.

**Model 2**: Non-linear model: $Y = \exp(\mathbf{X}^\top \boldsymbol{\beta} + \epsilon)$.

Here, the predictors $\mathbf{X}_i, i = 1, \cdots, n$ are generated from multivariate normal distribution $N_p(\mathbf{0}_p, \boldsymbol{\Sigma})$. The covariance matrix $\boldsymbol{\Sigma} \in \mathbb{R}^{p \times p}$ is block diagonal, that is, $\boldsymbol{\Sigma} = \text{diag}(\boldsymbol{\Sigma}_1, \cdots, \boldsymbol{\Sigma}_{10})$, where $\boldsymbol{\Sigma}_1$ is $\frac{p}{10} \times \frac{p}{10}$ dimensional identity matrix, and each $\boldsymbol{\Sigma}_k$ is $\frac{p}{10} \times \frac{p}{10}$ dimensional with an AR(1) correlation structure, that is $(\boldsymbol{\Sigma}_k)_{ij} = (0.1k - 0.1)^{|i-j|}, k = 2, \ldots, 10; i, j = 1, \cdots, p/10$. We consider two different error distributions for $\epsilon$ which is independent of $\mathbf{X}$: (1) standard normal distribution $N(0, 1)$; (2) Cauchy distribution or equivalently Student's $t$ distribution with 1 degree of freedom, $t(1)$. Without loss of generality, we set the active set be $\{1, 2, \ldots, 6\}$. The regression coefficients $\boldsymbol{\beta}$ are generated from an arithmetic sequence from 0.1 to 2, that is $\beta_j = 0.2 + 0.038(j - 1)$ for $1 \leq j \leq 6$ and $\beta_j = 0$ otherwise. We consider $n = 200, 500$ and $p = 800$. We use Intel(R) Xeon(R) Silver 4208 CPU @ 2.10GHz.

In order to explore the robustness behaviors of our proposed statistics, we add outliers to pollute the observations: $p_{\text{out}}$ of the responses are picked at random and increased by $m_{\text{out}}$-times maximum of original responses, shorted as $p_{\text{out}} + m_{\text{out}} \cdot \max(\text{responses})$. Here $p_{\text{out}}$ is the proportion of outliers and $m_{\text{out}}$ is a pre-set constant standing for outlier strength. Throughout the simulation study, we analyze the results with both the original data (shorted as Ori.) and the data with outliers (shorted as Out.).

We assess the empirical type I error and the empirical power of group inference with the nominal level $\alpha = 0.05$ based on 500 simulation runs. We consider the following five different choices for $\mathcal{G}$ to test the hypothesis (1.2):

$$\mathcal{G}_1 = \{10, 11, p - 4, p - 3, p - 2, p - 1, p\},$$
$$\mathcal{G}_2 = \{3, \ldots, 6\} \cup \mathcal{G}_1,$$
$$\mathcal{G}_3 = \{10, \ldots, 59, p - 149, \ldots, p\},$$
$$\mathcal{G}_4 = \{3, \ldots, 6\} \cup \mathcal{G}_3.$$

Note that $\mathcal{G}_1$ and $\mathcal{G}_2$ are small groups, $\mathcal{G}_3$ and $\mathcal{G}_4$ are large groups. $\mathcal{G}_1$ and $\mathcal{G}_3$ consist of only zero coefficients, while $\mathcal{G}_2$ and $\mathcal{G}_4$ includes nonzero elements.

Then we report the numerical results of empirical rejection rate (ERR) for different scenarios, where ERR is the proportion of rejected hypotheses among the total 500 simulations. For $\mathcal{G}_1$ and $\mathcal{G}_3$, the ERR is the empirical type I error; For $\mathcal{G}_2$ and $\mathcal{G}_4$, the ERR is the empirical power.

From the numerical results for all scenarios displayed in table 1, we have the following findings. Firstly, when there are outliers or heavy errors, our methods still have very high powers and control empirical sizes well. Secondly, for both linear model and nonlinear model, small groups and large groups, our procedures have similar satisfactory performance in this experiment. Thirdly, we find that the dimensionality of all predictors $p$ does not effect the empirical performances. Our test procedures with LASSO penalty perform similarly to those with SCAD or MCP. Therefore, we only present the results of test statistics with LASSO penalty. We use the R-package `ncvreg` [6].

Moreover, we compare the proposed methods with the three-step testing procedure based on the studentized statistics in [59], denoted as ST. Here, we use R package `SILM` [59] to implement the three-step testing procedure, set the number of bootstrap replications as 500 and choose the splitting proportion of 30% for screening.

Table 2 summarizes the results of empirical type I errors and powers with $\alpha = 0.05$ for different methods and models based on 500 simulation runs (due to the computation limit, we only run 200 times for ST). Here we consider $(n, p) = (200, 400)$, and other settings are same as before. It is obvious that our methods outperform the ST. Firstly, when there are no outliers and heavy errors, all methods have empirical power 1 for linear model. But for non-linear model, our test procedures $T_\alpha^{(1)}$ and $T_\alpha^{(2)}$ are more powerful than ST. For instance, under non-linear model with original data and $\epsilon \sim N(0, 1)$, the empirical powers of ST for $\mathcal{G}_2$ and $\mathcal{G}_4$ are 0.785 and 0.430, while the corresponding

Table 1: Simulation results for the group inference problems

| | $\epsilon \sim N(0,1)$ | | | | | | | | $\epsilon \sim t(1)$ | | | |
| | $\mathcal{G}_1$ | | $\mathcal{G}_2$ | | $\mathcal{G}_3$ | | $\mathcal{G}_4$ | | $\mathcal{G}_1$ | $\mathcal{G}_2$ | $\mathcal{G}_3$ | $\mathcal{G}_4$ |
| $n$ | Ori. | Out. | Ori. | Out. | Ori. | Out. | Ori. | Out. | Ori. | Ori. | Ori. | Ori. |
| --- | --- | --- | --- | --- | --- | --- | --- | --- | --- | --- | --- | --- |
| | | | | | $Y = \mathbf{X}^\top \boldsymbol{\beta} + \epsilon$ | | | | | | | |
| 200 | 0.028 | 0.034 | 1.000 | 1.000 | 0.026 | 0.046 | 1.000 | 1.000 | 0.026 | 1.000 | 0.058 | 1.000 |
| 500 | 0.018 | 0.028 | 1.000 | 1.000 | 0.024 | 0.040 | 1.000 | 1.000 | 0.032 | 1.000 | 0.054 | 1.000 |
| | | | | | $Y = \exp(\mathbf{X}^\top \boldsymbol{\beta} + \epsilon)$ | | | | | | | |
| 200 | 0.018 | 0.024 | 1.000 | 1.000 | 0.030 | 0.052 | 1.000 | 1.000 | 0.044 | 1.000 | 0.084 | 1.000 |
| 500 | 0.034 | 0.040 | 1.000 | 1.000 | 0.026 | 0.040 | 1.000 | 1.000 | 0.030 | 1.000 | 0.028 | 1.000 |

empirical powers of our methods are all 1. Secondly, the outliers and heavy errors have dramatic effect on the power performances of ST, which has very low powers even as low as the nominal level. For example, under linear model with outliers, the empirical power of ST for $\mathcal{G}_4$ is 0.050 and that of our proposal is 1; under non-liner model with Cauchy error, the corresponding empirical powers of $ST$ and our proposal are 0.040 and 1, respectively. Thirdly, all methods can control type I errors reasonably. Overall our methods have better powers and are robust with respect to outliers and heavy errors, whereas ST in [59] can completely fail when outliers exist.

Table 2: Simulation results for our method and ST.

| | $\epsilon \sim N(0,1)$ | | | | | | | | $\epsilon \sim t(1)$ | | | |
| | $\mathcal{G}_1$ | | $\mathcal{G}_2$ | | $\mathcal{G}_3$ | | $\mathcal{G}_4$ | | $\mathcal{G}_1$ | $\mathcal{G}_2$ | $\mathcal{G}_3$ | $\mathcal{G}_4$ |
| Method | Ori. | Out. | Ori. | Out. | Ori. | Out. | Ori. | Out. | Ori. | Ori. | Ori. | Ori. |
| --- | --- | --- | --- | --- | --- | --- | --- | --- | --- | --- | --- | --- |
| | | | | | $Y = \mathbf{X}^\top \boldsymbol{\beta} + \epsilon$ | | | | | | | |
| Our | 0.028 | 0.030 | 1.000 | 1.000 | 0.038 | 0.044 | 1.000 | 1.000 | 0.012 | 1.000 | 0.050 | 1.000 |
| ST | 0.045 | 0.040 | 1.000 | 0.115 | 0.015 | 0.040 | 1.000 | 0.050 | 0.055 | 0.370 | 0.055 | 0.250 |
| | | | | | $Y = \exp(\mathbf{X}^\top \boldsymbol{\beta} + \epsilon)$ | | | | | | | |
| Our | 0.032 | 0.054 | 1.000 | 1.000 | 0.034 | 0.070 | 1.000 | 1.000 | 0.042 | 1.000 | 0.054 | 1.000 |
| ST | 0.025 | 0.040 | 0.785 | 0.045 | 0.060 | 0.075 | 0.430 | 0.080 | 0.045 | 0.085 | 0.035 | 0.040 |

## 5 Discussions

In this paper, we investigate robust inference for high-dimensional single index model (SIM). Under the linearity condition about the predictors, we recast the SIM into pseudo-linear model with transformed response. A response-distribution transformation is considered. This transformation choice avoids the sub-Gaussian assumption for the response. Our introduced procedures are thus robust with respect to outliers or heavy-tailed errors. We develop asymptotically honest group inference procedures. Asymptotic distribution and bootstrap implementation are both established. For testing the individual coefficients simultaneously, multiple testing procedures are proposed and shown to control the false discovery rate asymptotically. Our numerical studies illustrate the robustness of our procedures against outliers or heavy errors.

In this paper, the linearity condition about the predictors plays a very important role in the methodology development. How to make inference for high-dimensional SIM without this condition is of great importance and interest. The second-order Stein's method with score function-based corrections investigated in [56] and [55] would be a powerful alternative. The condition $\kappa_h \neq 0$ excludes even link functions, and in particular the problem of sparse phase retrieval. [39] introduced a novel procedure to deal with the problem of sparse phase retrieval when $\kappa_h = 0$. It would be of interest to extend their approach to make robust group inference. Further our paper doesn't provide quantitative bounds on

robustness. It is also of interest to make inference for partially linear single-index regression model [15]. We will pursue these challenging problems in near future.

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

# A   Appendix / supplemental material

**Notation.** For functions $f(n)$ and $g(n)$, we write $f(n) \lesssim g(n)$ to mean that $f(n) \leq cg(n)$ for some universal constant $c \in (0, \infty)$, and similarly, $f(n) \gtrsim g(n)$ when $f(n) \geq c'g(n)$ for some universal constant $c' \in (0, \infty)$. We write $f(n) \asymp g(n)$ when $f(n) \lesssim g(n)$ and $f(n) \gtrsim g(n)$ hold simultaneously. For a function $h : \mathbb{R}^p \to \mathbb{R}$, we write $\nabla h$ and $\nabla_k h$ to denote a gradient or subgradient and its $k$-th component, if they exist. The *sub-Gaussian norm* of a *sub-Gaussian* random variable $X$ is defined as $\|X\|_{\psi_2} := \inf\{t > 0 : \mathbb{E} \exp(X^2/t^2) \leq 2\}$. Similarly, the *sub-Exponential norm* of a *sub-Exponential* random variable $Y$ is defined as $\|Y\|_{\psi_1} := \inf\{t > 0 : \mathbb{E} \exp(|Y|/t) \leq 2\}$.

## A.1   A multiplier bootstrap test procedure

As discussed in Remark 2.1, the critical value obtained from the asymptotic distribution may not work well in practice since this weak convergence is typically slow and assumptions for dependence structure of the covariates. We next describe a simple gaussian multiplier bootstrap method to obtain an accurate critical value. Let $w_1, \ldots, w_n$ be i.i.d. $N(0, 1)$ random variables that are independent of $\{\mathbf{X}_i, Y_i\}_{i=1}^n$ and define the gaussian multiplier bootstrap statistic:

$$M_{n,\mathcal{G}}^{\sharp} = \max_{j \in \mathcal{G}} \left[ \frac{2}{\hat{\sigma}_j \sqrt{n}} \sum_{i=1}^{n} \left\{ \frac{1}{n-1} \sum_{\tilde{i} \neq i} \hat{h}_j^{sym}(Y_i, \mathbf{X}_i; Y_{\tilde{i}}, \mathbf{X}_{\tilde{i}}) - \frac{1}{\sqrt{n}} T_{nj} \right\} w_i \right]^2,$$

where

$$\hat{h}_j^{sym}(Y_i, \mathbf{X}_i; Y_{\tilde{i}}, \mathbf{X}_{\tilde{i}}) = \frac{1}{2} \left\{ \hat{h}_j(Y_i, \mathbf{X}_i; Y_{\tilde{i}}, \mathbf{X}_{\tilde{i}}) + \hat{h}_j(Y_{\tilde{i}}, \mathbf{X}_{\tilde{i}}; Y_i, \mathbf{X}_i) \right\},$$

and

$$\hat{h}_j(Y_i, \mathbf{X}_i; Y_{\tilde{i}}, \mathbf{X}_{\tilde{i}}) = \left\{ I(Y_{\tilde{i}} \leq Y_i) - \frac{1}{2} - \mathbf{Z}_{ij}^{\top} \hat{\boldsymbol{\gamma}}_j \right\} \left( X_{ij} - \mathbf{Z}_{ij}^{\top} \hat{\boldsymbol{\theta}}_j \right)$$

is the kernel funtion in $T_{nj}$. The bootstrap critical value is given by $c_{\mathcal{G}}^{(2)}(\alpha) = \inf\{t \in \mathbb{R} : \Pr_w\big(M_{n,\mathcal{G}}^{\sharp} \leq t\big) \geq 1 - \alpha\}$, where $\Pr_w$ is the probability measure induced by the multiplier variables $\{w_i\}_{i=1}^n$ holding $\{\mathbf{X}_i, Y_i\}_{i=1}^n$ fixed. We can reject null hypothesis $\mathbb{H}_{0,\mathcal{G}}$ at the significant level $\alpha$ if and only if $M_{n,\mathcal{G}} \geq c_{\mathcal{G}}^{(2)}(\alpha)$,

**Theorem A.1.** *Suppose that Assumptions A.1, A.2, A.7, A.8 and LC condition hold. If $s = o\big(\sqrt{n}/(\log p(\log np_0)^{3/2})\big)$, we have*

$$\lim_{n \to \infty} \sup_{\boldsymbol{\beta}_h \in \Omega_{\mathcal{G}}^0} \sup_{\alpha \in (0,1)} \left| P\big(M_{n,\mathcal{G}} > c_{\mathcal{G}}^{(2)}(\alpha)\big) - \alpha \right| = 0. \tag{1.16}$$

**Remark A.1.** *Here we assume $s = o\big(\sqrt{n}/(\log p (\log n p_0)^{3/2})\big)$ which is slightly stricter than $s = o(\sqrt{n}/(\log p \log p_0))$ in Theorem 3.2. Suppose $\zeta_1$ is a sequence satisfying $\zeta_1 \sqrt{1 \vee \log(p_0/\zeta_1)} \to 0$ as $n \to \infty$. This assumption has two purposes. Firstly, making the difference between $\|\widetilde{T}_{n,\mathcal{G}}\|_\infty$ and $\|\widetilde{S}_{n,\mathcal{G}}\|_\infty$ bounded by $\zeta_1$. Secondly, making $\mathrm{Pr}_w\big(\big|\sqrt{M_{n,\mathcal{G}}^\sharp} - \sqrt{L_{n,\mathcal{G}}^\sharp}\big| \geq \zeta_1\big) \to 0$ in probability, where $L_{n,\mathcal{G}}^\sharp$ is the ideal version of $M_{n,\mathcal{G}}^\sharp$ and it is defined in (1.27) in the Appendix A.9.*

**Theorem A.2.** *Suppose that conditions in Theorem A.1 are satisfied, we have for some $\epsilon_0 > 0$,*

$$\lim_{(n,p_0)\to\infty} \inf_{\boldsymbol{\beta}_h \in \Omega_{\mathcal{G}}^1(2+\epsilon_0)} \mathrm{Pr}\big(M_{n,\mathcal{G}} \geq c_{\mathcal{G}}(\alpha)\big) = 1. \tag{1.17}$$

*Where $\Omega_{\mathcal{G}}^1(\cdot)$ is defined in subsection 3.3.*

The theoretical guarantee of the bootstrap procedure is based on the gaussian approximation theory for nondegenerate $U$-statistic. Symmetrization of the kernel function $\hat{h}_j(\cdot)$ plays an important role in theory. Thus $\hat{h}_j^{sym}(\cdot)$ is necessary in our gaussian multiplier bootstrap procedure. Meanwhile, Theorem A.1 holds whether $p_0$ is divergent or not. While the limiting null distribution established in Theorem 3.2 asks $p_0$ be divergent.

## A.2 The detailed form of $c_3$

By the definition of SNR and Lemma A.8 in the Supplementary Material, for a given $\boldsymbol{\beta} \in \mathbb{R}^p$ we have

$$\mathrm{SNR}(M_{n,\mathcal{G}}) = \frac{Cov^2(F(Y), \mathbf{X}^\top \boldsymbol{\beta})}{(\boldsymbol{\beta}^\top \Sigma \boldsymbol{\beta})^2} \max_{j \in \mathcal{G}} \left( \frac{\beta_j \delta_j}{\sigma_j} \right)^2.$$

Similarly, we have

$$\mathrm{SNR}(M_{n,\mathcal{G}}^+) = \frac{Cov^2(Y, \mathbf{X}^\top \boldsymbol{\beta})}{(\boldsymbol{\beta}^\top \Sigma \boldsymbol{\beta})^2} \max_{j \in \mathcal{G}} \left( \frac{\beta_j \delta_j}{\sigma_j^+} \right)^2.$$

Thus the ratio of $\mathrm{SNR}(M_{n,\mathcal{G}})$ and $\mathrm{SNR}(M_{n,\mathcal{G}}^+)$ can be simplified as

$$\frac{\mathrm{SNR}(M_{n,\mathcal{G}})}{\mathrm{SNR}(M_{n,\mathcal{G}}^+)} \geq \frac{Cov^2(F(Y), \mathbf{X}^\top \boldsymbol{\beta})}{Cov^2(Y, \mathbf{X}^\top \boldsymbol{\beta})} \cdot \frac{(\sigma_{j'}^+)^2}{\sigma_{j'}^2},$$

where $j' = \arg_{j \in \mathcal{G}} \max\left( \frac{\beta_j \delta_j}{\sigma_j^+} \right)^2$. In the case of classical SIM, that is, $Y = g_1(\mathbf{X}^\top \boldsymbol{\beta}) + \epsilon$ and $\epsilon$ is independent of $\mathbf{X}$. We can derive

$$(\sigma_{j'}^+)^2 = E\{(X_{j'} - \mathbf{Z}_{j'}^\top \boldsymbol{\theta}_{j'})^2 (Y - \mathbf{Z}_{j'}^\top \boldsymbol{\gamma}_{j'})^2\} \geq E(X_{j'} - \mathbf{Z}_{j'}^\top \boldsymbol{\theta}_{j'})^2 \mathrm{Var}(\epsilon).$$

Thus we have

$$\frac{\mathrm{SNR}(M_{n,\mathcal{G}})}{\mathrm{SNR}(M_{n,\mathcal{G}}^+)} \geq \frac{Cov^2(F(Y), \boldsymbol{\beta}^\top \mathbf{X}) E(X_{j'} - \mathbf{Z}_{j'}^\top \boldsymbol{\theta}_{j'})^2}{Cov^2(Y, \boldsymbol{\beta}^\top \mathbf{X}) \sigma_{j'}^2} \mathrm{Var}(\epsilon),$$

and

$$c_3 = \frac{Cov^2(F(Y), \boldsymbol{\beta}^\top \mathbf{X}) E(X_{j'} - \mathbf{Z}_{j'}^\top \boldsymbol{\theta}_{j'})^2}{Cov^2(Y, \boldsymbol{\beta}^\top \mathbf{X}) \sigma_{j'}^2}.$$

## A.3 False discovery rate control

In the previous section, we consider joint significance testing of a prespecified subset of predictors. In applications, however, the parameter of interest may not be specified in advance. Denote $\mathcal{H}_0 = \{j : \beta_j = 0, j = 1, \ldots, p\}$, and $\mathcal{H}_1 = \{j : \beta_j \neq 0, j = 1, \ldots, p\}$. It is of interest to identify the elements in $\mathcal{H}_1$. To this aim, we consider simultaneous testing of the following hypotheses

$$\mathbb{H}_{0j} : \beta_j = 0 \qquad \text{versus} \qquad \mathbb{H}_{1j} : \beta_j \neq 0, \quad 1 \leq j \leq p.$$

In this section, we aim to develop data-driven procedures to control the FDR.

Recall that in Section 2, for each individual hypothesis $\mathbb{H}_{0j} : \beta_j = 0$ we define the standardized statistic $\widetilde{T}_{nj} = \hat{\sigma}_j^{-1} T_{nj}$ defined in (2.7). At a given threshold level $t > 0$, $\mathbb{H}_{0j}$ is rejected if $|\widetilde{T}_{nj}| \geq t$. For each $t$, let $R_0(t) = \sum_{j \in \mathcal{H}_0} I(|\widetilde{T}_{nj}| \geq t)$, and $R(t) = \sum_{j=1}^p I(|\widetilde{T}_{nj}| \geq t)$ be the total number of false discoveries and the total number of discoveries, respectively. Accordingly, the false discovery proportion (FDP) and FDR are

$$\text{FDP}(t) = \frac{\sum_{j \in \mathcal{H}_0} I(|\widetilde{T}_{nj}| \geq t)}{\max\{\sum_{j=1}^p I(|\widetilde{T}_{nj}| \geq t), 1\}}, \qquad \text{FDR}(t) = \mathbb{E}\{\text{FDP}(t)\}.$$

Let $G_0(t)$ be the proportion of the nulls falsely rejected by the procedure among all the true nulls at the threshold level $t$, namely, $G_0(t) = q_0^{-1} \sum_{j \in \mathcal{H}_0} I(|\widetilde{T}_{nj}| \geq t)$, where $q_0 = |\mathcal{H}_0|$. In practice, it is reasonable to assume that the true alternatives are sparse, that is, $p - q_0 =: q_1 = o(p)$. If the sample size is large, we can use the tails of normal distribution $G(t) = 2 - 2\Phi(t)$ to approximate $G_0(t)$. In fact, it will be shown that, for $b_p = \sqrt{2 \log p - \log \log p}$,

$$\sup_{0 \leq t \leq b_p} \left| \frac{G_0(t)}{G(t)} - 1 \right| \to 0$$

in probability as $(n, p) \to \infty$. To summarize, we have the following multiple testing procedure controlling the FDR and FDP at a pre-specified level $0 < \alpha < 1$, which is summarized in Algorithm A.1.

---

**Algorithm 2:** multiple testing procedure

---

1 Let $0 < \alpha < 1$, $b_p = \sqrt{2 \log p - \log \log p}$ and define

$$\hat{t} = \inf\left\{ 0 \leq t \leq b_p : \frac{pG(t)}{\max\{\sum_{j=1}^p I(|\widetilde{T}_{nj}| \geq t), 1\}} \leq \alpha \right\}. \qquad (1.18)$$

If $\hat{t}$ in (1.18) does not exist, then let $\hat{t} = \sqrt{2 \log p}$. We reject $\mathbb{H}_{0j}$ whenever $|\widetilde{T}_j| \geq \hat{t}$.

---

Next we show that under mild conditions, our proposed multiple testing procedure control the FDR asymptotically. Recall that $\Omega = \left\{ \beta_h \in \mathbb{R}^p : \|\beta_h\|_0 \vee \max_{1 \leq j \leq p} \|\theta_j\|_0 \leq s \right\}$. The following theorem shows that $\widetilde{T}_{nj}$ is uniformly asymptotically normal distributed and $G_0(t)$ is well approximated by $G(t)$.

**Theorem A.3.** *Suppose that Assumptions A.1, A.2, A.8 in Appendix A.8 and LC condition hold. If $p = O(n^c)$ for some constant $c > 0$ and $s = o(\sqrt{n}/(\log p)^2)$. Then as $(n, p) \to \infty$, for any $\beta_h \in \Omega$ we have*

$$\sup_{j \in \mathcal{H}_0} \sup_{0 \leq t \leq \sqrt{2 \log p}} \left| \frac{\Pr(|\widetilde{T}_{nj}| \geq t)}{2 - 2\Phi(t)} - 1 \right| \to 0. \qquad (1.19)$$

*If in addition Assumption A.9 holds, then*

$$\sup_{0 \leq t \leq b_p} \left| \frac{G_0(t)}{G(t)} - 1 \right| \to 0 \qquad (1.20)$$

*in probability.*

In the results given in Section 2, we could allow $p$ to grow exponentially fast in $n$. However, in order to control FDR, we only allow $p = O(n^c)$ to apply the moderate derivation result in [34]. Further we assume $s = o(\sqrt{n}/(\log p)^2)$ which is slightly stricter than $s = o(\sqrt{n}/(\log p \log p_0))$ to make the difference between $\widetilde{T}_{nj}$ and its ideal version $\widetilde{S}_{nj}$ uniformly bounded with the rate $o(1/\sqrt{\log p})$. Note that [28] imposed a similar condition. The following theorem provides the asymptotic FDR control of our procedure.

**Theorem A.4.** *Under the assumptions in Theorem A.3, for $\hat{t}$ defined in our multiple testing procedure, we have*

$$\limsup_{(n,p) \to \infty} \text{FDR}(\hat{t}) \leq \alpha. \qquad (1.21)$$

## A.4 Discussions of robustness

In this subsection, we carefully discuss the robustness of our inference procedure based on the efficient influence function. We find that our statistics are robust with respect to perturbations in the responses. The details are given below. Recall that our test procedure is inspired by the quantity

$$I := \mathbb{E}[\{F(Y) - 1/2 - Z_j^\top \gamma_j\}(X_j - Z_j^\top \theta_j)].$$

Next we derive the efficient influence function (EIF) of $I$. We rewrite $I$ as

$$I = \Psi(\mathcal{P}) = \mathbb{E}_{\mathcal{P}}[\{F_{\mathcal{P}}(Y) - 1/2 - Z_j^\top \gamma_j\}(X_j - Z_j^\top \theta_j)],$$

where $\mathcal{P}$ is distribution of $(X_j, Z_j^\top, Y)^\top$. Consider the following parametric submodel indexed by $t$, i.e.

$$\mathcal{P}_t = t\tilde{\mathcal{P}} + (1 - t)\mathcal{P},$$

where $t \in [0, 1]$, and $\tilde{\mathcal{P}}$ is a point mass at a single observation $\tilde{o} = (\tilde{x}_j, \tilde{z}_j^\top, \tilde{y})^\top$. As mentioned in [27], the EIF for $I$ at observation $\tilde{o}$ is

$$\phi(\tilde{o}, \mathcal{P}) = \frac{d\Psi(\mathcal{P}_t)}{dt}\Big|_{t=0},$$

where $\Psi(\mathcal{P}_t) = \mathbb{E}_{\mathcal{P}_t}[\{F_{\mathcal{P}_t}(Y) - 1/2 - Z_j^\top \gamma_j\}(X_j - Z_j^\top \theta_j)]$. Denote $o = (x_j, z_j^\top, y)^\top$ and $O = (X_j, Z_j^\top, Y)^\top$. Let $m_t(o) = \{F_{\mathcal{P}_t}(y) - 1/2 - z_j^\top \gamma_j\}(x_j - z_j^\top \theta_j)$ and $\Psi(\mathcal{P}_t) = \mathbb{E}_{\mathcal{P}_t}[m_t(O)]$. Further, the operator, $\partial_t$, applied to an arbitrary function $g(t)$, is defined as

$$\partial_t g(t) = \frac{dg(t)}{dt}\Big|_{t=0}.$$

Some calculation entails that

$$\partial_t \Psi(\mathcal{P}_t) = \int \partial_t m_t(O) d\mathcal{P}_t|_{t=0} + \int m_t(O)|_{t=0} d\partial_t \mathcal{P}_t$$
$$= \mathbb{E}_{\mathcal{P}}[\partial_t m_t(O)] + m_0(\tilde{o}) - \Psi(\mathcal{P}).$$

Note that

$$m_t(O) = \{t\mathbb{I}(Y \geq \tilde{y}) + (1 - t)F_{\mathcal{P}}(Y) - 1/2 - Z_j^\top \gamma_j\}(X_j - Z_j^\top \theta_j).$$

Thus we derive that

$$\partial_t m_t(O) = \{\mathbb{I}(Y \geq \tilde{y}) - F_{\mathcal{P}}(Y)\}(X_j - Z_j^\top \theta_j),$$

and

$$\mathbb{E}_{\mathcal{P}}[\partial_t m_t(O)] = \mathbb{E}_{\mathcal{P}}[\{\mathbb{I}(Y \geq \tilde{y}) - F_{\mathcal{P}}(Y)\}(X_j - Z_j^\top \theta_j)].$$

As $m_0(\tilde{o}) = \{F_{\mathcal{P}}(\tilde{y}) - 1/2 - \tilde{z}_j^\top \gamma_j\}(\tilde{x}_j - \tilde{z}_j^\top \theta_j)$, by some calculations we derive

$$\phi(\tilde{o}, \mathcal{P}) = \{F_{\mathcal{P}}(\tilde{y}) - 1/2 - \tilde{z}_j^\top \gamma_j\}(\tilde{x}_j - \tilde{z}_j^\top \theta_j) + \mathbb{E}_{\mathcal{P}}[\{\mathbb{I}(Y \geq \tilde{y}) - F_{\mathcal{P}}(Y)\}(X_j - Z_j^\top \theta_j)] - \Psi(\mathcal{P}).$$

Recall that

$$T_{nj} = \frac{1}{\sqrt{n}} \sum_{i=1}^{n} \left[ \left\{ F_n(Y_i) - \frac{1}{2} - Z_{ij}^\top \hat{\gamma}_j \right\} \left( X_{ij} - Z_{ij}^\top \hat{\theta}_j \right) \right],$$

$$S_{nj} = \frac{1}{\sqrt{n}} \sum_{i=1}^{n} \left[ \left\{ F(Y_i) - \frac{1}{2} - Z_{ij}^\top \gamma_j \right\} \left( X_{ij} - Z_{ij}^\top \theta_j \right) + m_j(Y_i) \right],$$

where $m_j(y) = \mathbb{E}[(X_j - Z_j^\top \theta_j)\{I(Y \geq y) - F(Y)\}]$. Here $T_{nj}$ is an individual test statistic corresponding to the parameter $I$. While $S_{nj}$ is an individual quantity corresponding to the EIF of $I$. From our proof, we know that $T_{nj} = S_{nj} + o_p(1)$.

By the formula of $\phi(\tilde{o}, \mathcal{P})$, given $(x_j, z_j^\top)^\top$, it can shown that $\phi(\tilde{o}, \mathcal{P})$ is bounded for any $\tilde{y} \in \mathbb{R}$. Thus our test statistics are robust with respect to the perturbations in the responses.

## A.5 Additional simulation results for group inference

In this subsection, we conduct simulation studies to compare our procedure with other methods based on different transformation functions. The simulation settings are summarized as follows. We consider the following two models:

- Model 1: Linear model: $Y = X^\top \beta + \epsilon$.
- Model 2: Non-linear model: $Y = \exp(X^\top \beta + \epsilon)$.

The regression coefficients $\beta = (\beta_1, \beta_2, \ldots, \beta_p)^\top$ are generated as $\beta_j = \delta$ for $j = 1, \ldots, 6$ and $\beta_j = 0$ otherwise, where $\delta$ can be regarded as a signal strength parameter. We generate the error term from the standard normal distribution. We add outliers to pollute the observations: $p_{out}$ of the responses are picked at random and increased by $m_{out}$-times maximum of original responses. Specifically, the detailed settings for the above parameters are as follows. Firstly, we consider the sample size $n = 500$ and the dimension $p = 800$. Secondly, we set the signal strength parameter $\delta$ to vary from $\{0.1, 0.3, 0.5\}$. Thirdly, we fix $m_{out}$ to 10. Lastly, we vary $p_{out}$ from 0 to 0.5 in increments of 0.1. For more complete comparisons, we consider three transformation procedures: (1) $h(Y) = F(Y)$ (Our method); (2) $h(Y) = Y$; (3) $h(Y) = \text{sigmoid}(Y) = 1/\{1 + \exp(-Y)\}$. Other simulation settings are the same as described in section 4 of the main text.

Figure 1 summarizes the results of empirical type I error and empirical power with the significant level of $\alpha = 0.05$ for different methods when the error term follows the standard normal distribution. Firstly, under the null hypothesis $(G_1)$, $h(Y) = \text{sigmoid}(Y)$ cannot control the type I error for Model 2, while other procedures perform well. Secondly, under the alternative hypothesis $(G_2)$, the empirical powers of other procedures decrease rapidly with the increase of $p_{out}$, while the powers of our procedure remain stable, which is particularly noticeable when $\delta = 0.5$. This finding indicates that our method has strong robustness when the responses are polluted. Thirdly, our method performs well for both Model 1 and Model 2, indicating that our method is robust across different single-index models.

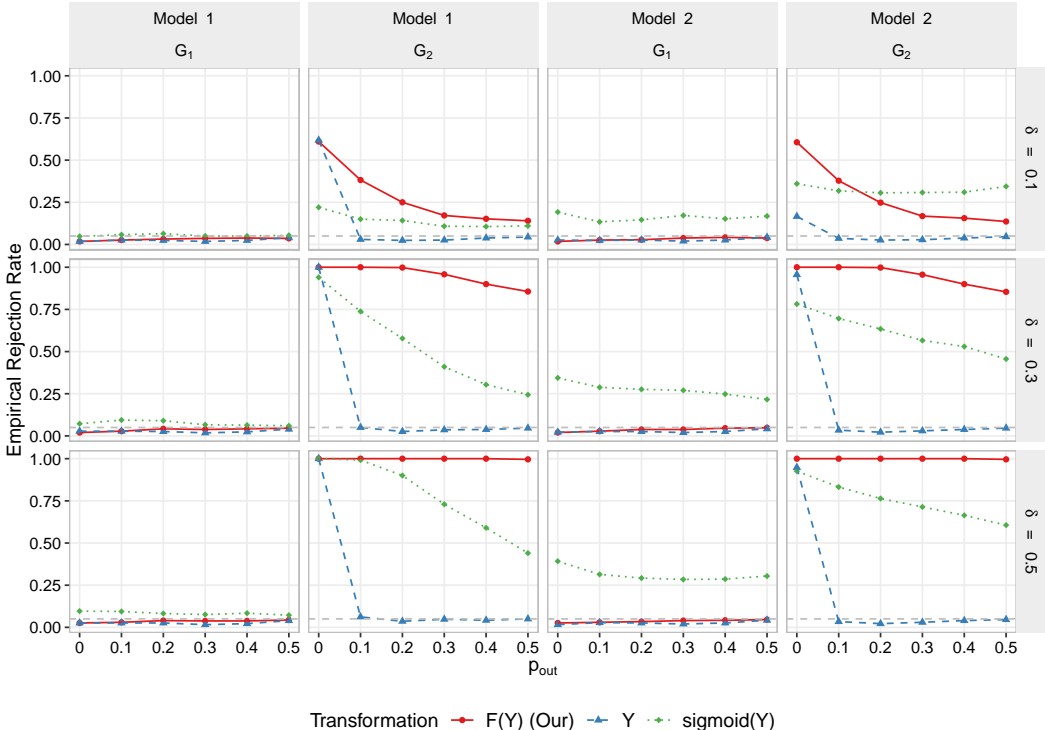

Figure 1: Under different settings of the generated model, the testing group and the signal strength, simulated results of the proposed method for the proportion of outliers $p_{out}$ from 0 to 0.5 in increments of 0.1 when the error term follows the standard normal distribution.

## A.6 Additional simulation results for false discovery rate control

In this subsection, we examine the empirical FDR results of our proposed procedure. We also make comparisons with two alternatives. Let BH-$V_n$ denote the BH procedure [4] with $p$-values obtained from $V_n$. BH-$W_n$ is similarly defined. Here $V_n$ and $W_n$ denote the debiased lasso procedure introduced by [17] and the decorrelated score method proposed by [40], respectively. Here, we use the R function `p.adjust(...,method="BH")` to implement the BH procedure. We simultaneously test $\mathbb{H}_{0j} : j \in \mathcal{H}_0$ for all $j = 1, \cdots, p$. We set the dimension $p = 1000$ and sample size $n = 1000, 1500$. The sparsity level $|\mathcal{H}_1| = q_1$ is set as 4 or 8. The data is generated from the following linear and nonlinear models:

$$Y = \mathbf{X}^\top \boldsymbol{\beta} + \epsilon,$$
$$Y = \sin(0.5\mathbf{X}^\top \boldsymbol{\beta}) \cdot \exp(\mathbf{X}^\top \boldsymbol{\beta} + 1) + \epsilon,$$

where $\boldsymbol{\beta} = \boldsymbol{\beta}_1$ with $\delta_0 = 1$, that is $\boldsymbol{\beta} = (\overbrace{1, \cdots, 1}^{s_0}, 0, \cdots, 0)^\top$ and $\epsilon$ comes from $N(0,1)$ or $t(1)$.

We evaluate the empirical FDRs and the empirical powers of our test statistic $T_n$ and two existing high-dimensional inference procedures BH-$V_n$ and BH-$W_n$. Here the power is defined as the number of correctly discovered variables divided by the number of truly active variables. We set the desired FDR level as $\alpha = 0.2$ and the results are calculated based on 500 replications.

We illustrate our proposed procedure under different models, error distributions, sample sizes, and sparsity levels. Table 3 shows the simulation results. We have the following important observations. Firstly our introduced procedure controls the FDR well, even when the random error comes from Cauchy distribution. Secondly, our method has power 1, indicating that the proposed method can detect all important predictors. Thirdly, the other two methods lose their powers when random error follows Cauchy distribution and sometimes fail to control FDR.

Table 3: Empirical FDRs and powers of our proposed method, BH-$V_n$ and BH-$W_n$. Here the dimension $p = 1000$ and nominal FDR level $\alpha = 0.2$.

| | | Linear model | | | | Non-linear model | | | |
| --- | --- | --- | --- | --- | --- | --- | --- | --- | --- |
| | | $n = 1000$ | | $n = 1500$ | | $n = 1000$ | | $n = 1500$ | |
| Sparsity | Method | FDR | Power | FDR | Power | FDR | Power | FDR | Power |
| | | $\epsilon \sim N(0,1)$ | | | | | | | |
| 4 | Proposed | 0.188 | 1.000 | 0.187 | 1.000 | 0.197 | 1.000 | 0.203 | 1.000 |
| | BH-$V_n$ | 0.245 | 1.000 | 0.235 | 1.000 | 0.198 | 0.773 | 0.206 | 0.730 |
| | BH-$W_n$ | 0.133 | 1.000 | 0.122 | 1.000 | 0.125 | 0.851 | 0.125 | 0.879 |
| 8 | Proposed | 0.194 | 1.000 | 0.198 | 1.000 | 0.211 | 1.000 | 0.211 | 1.000 |
| | BH-$V_n$ | 0.243 | 1.000 | 0.230 | 1.000 | 0.458 | 0.011 | 0.248 | 0.005 |
| | BH-$W_n$ | 0.138 | 1.000 | 0.142 | 1.000 | 0.097 | 0.028 | 0.122 | 0.041 |
| | | $\epsilon \sim t(1)$ | | | | | | | |
| 4 | Proposed | 0.191 | 1.000 | 0.196 | 1.000 | 0.196 | 1.000 | 0.213 | 1.000 |
| | BH-$V_n$ | 0.328 | 0.017 | 0.283 | 0.013 | 0.249 | 0.448 | 0.204 | 0.446 |
| | BH-$W_n$ | 0.121 | 0.028 | 0.104 | 0.032 | 0.127 | 0.628 | 0.134 | 0.664 |
| 8 | Proposed | 0.204 | 1.000 | 0.206 | 1.000 | 0.212 | 1.000 | 0.202 | 1.000 |
| | BH-$V_n$ | 0.299 | 0.019 | 0.244 | 0.013 | 0.460 | 0.014 | 0.245 | 0.006 |
| | BH-$W_n$ | 0.087 | 0.022 | 0.113 | 0.031 | 0.114 | 0.024 | 0.093 | 0.050 |

## A.7 Real data analysis

In this subsection, we illustrate the practical applications of our procedures by a real data analysis. The group inference is helpful to decide whether a group of predictors are important or not for the response. If we find a group of predictors are important, we would like to know which specific predictors in the group are significant. For this aim, our developed multiple testing procedure is useful. For instance, researchers may aim to test whether a gene pathway, consisting of high-dimensional

genes for the same biological function, is important for a certain clinical outcome, given the other high-dimensional genes. When determining that a certain gene pathway is important, researchers need to further identify specific genes within the pathway which are important for a certain clinical outcome.

We apply our methods on a dataset about riboflavin (vitamin B2) production rate with Bacillus Subtilis. This dataset is made publicly by [7] and has been analyzed by many authors, for instance [37], [47], [29], and [22]. The dataset riboflavin can be obtained from the R package `hdi`. It consists of $n = 71$ observations of strains of Bacillus Subtilis and $p = 4088$ covariates, measuring the log-expression levels of 4088 genes. The response variable is the logarithm of the riboflavin production rate.

Our goal is to detect which genes are associated with riboflavin production rate. Like most existing studies, we first reduce ultrahigh-dimension to a moderate high-dimension. Here we pick out first 300 genes by distance correlation based screening [33]. We first conduct global testing on these 300 genes. The $p$-value of our group inference procedure is $1.286e - 04$, indicating that the null hypothesis is rejected and the selected 300 genes are influential for riboflavin production rate. Next, we further use FDR control procedure to select the important genes in these 300 genes.

By implementing our proposed FDR control procedure with the FDR level of 0.1, we identify 10 genes that are significantly associated with the response. That is $\mathcal{G}_I = \{$YTGB_at, YCKE_at, YXLE_at, YXLD_at, YJCJ_at, XHLA_at, xepA_at, YCGO_at, RPLP_at, XKDS_at$\}$. If the FDR level is set as 0.2, 5 more genes will be selected. That is $\mathcal{G}_{II} = \mathcal{G}_I \cup \{$SPOIISA_at, YHCB_at, XKDI_at, YJCF_at, XHLB_at$\}$.

We further conduct group inference on the selected subsets $\mathcal{G}_I, \mathcal{G}_{II}$ and their complement sets $\mathcal{G}_I^c, \mathcal{G}_{II}^c$. As expected, our group inference procedure finds again that $\mathcal{G}_I, \mathcal{G}_{II}$ are significant while $\mathcal{G}_I^c, \mathcal{G}_{II}^c$ are not. The corresponding $p$-values of $\mathcal{G}_I, \mathcal{G}_I^c, \mathcal{G}_{II}$ and $\mathcal{G}_{II}^c$ are 6.748e-06, 7.257e-01, 9.334e-06 and 9.370e-01, respectively. These results suggest that the genes selected by the FDR control procedure are really influential.

We compare these selected genes with other methods. For example, the multi-sample-splitting method proposed in [37] identified YXLD_at; [47] did not select any gene using the de-sparsified Lasso; [29] only selected two genes: YXLD_at and YXLE_at and [22] claimed YCKE_at, XHLA_at, YXLD_at, YDAR_at and YCGN_at as significant. From $\mathcal{G}_I$ and $\mathcal{G}_{II}$, we can see clearly that the gene YXLD_at is detected by not only [37], [29], [22], but also by our procedure. Besides, the genes YCKE_at, YXLE_at and XHLA_at which are detected by [29] and [22], are also found by our method. Further, our procedure detects some additional important genes.

## A.8 Technical assumptions

**Assumption A.1.** *We assume that*

    *(i) $X_j$, $\mathbf{X}^\top \boldsymbol{\beta}_h$ and $\mathbf{Z}_j^\top \boldsymbol{\theta}_j$, $j = 1, \ldots, p$ are all sub-Gaussian with uniformly bounded sub-Gaussian norms.*

    *(ii) $c\sqrt{\frac{\log p}{n}} \leq \lambda_X, \lambda_Y \leq C\sqrt{\frac{\log p}{n}}$ for some constants $0 < c \leq C$.*

    *(iii) $\boldsymbol{\Sigma} = \mathrm{Cov}(\mathbf{X}) > 0$.*

The above conditions are mild and widely assumed in the high-dimensional literature. Condition (i) is used to obtain $l_\infty$ norm of $\frac{1}{\sqrt{n}}\sum_{i=1}^n (X_{ij} - \mathbf{Z}_{ij}^\top \boldsymbol{\theta}_j)\mathbf{Z}_{ij}$ in the proof. Sub-Gaussian assumption is a standard condition while working with random predictors in high-dimensional models [49]. However, we should emphasize here that we do not make sub-Gaussian assumption for the error term (or equivalently for the response), which is widely assumed in the literature of high-dimensional inference [40, 17]. Actually we even do not assume the existence of $\mathbb{E}(\epsilon)$.

**Assumption A.2.** *We assume that*

    *(i) The function $p_\lambda$ satisfies $p_\lambda(0) = 0$ and is symmetric around zero (i.e., $p_\lambda(t) = p_\lambda(-t)$, for all $t \in \mathbb{R}$).*

    *(ii) On the nonnegative real line, the function $p_\lambda$ is nondecreasing.*

    *(iii) For $t > 0$, the function $t \to \frac{p_\lambda(t)}{t}$ is nonincreasing in t.*

(iv) *The function $p_\lambda$ is differentiable for all $t \neq 0$ and subdifferentiable at $t = 0$, with $\lim_{t\to 0^+} p'_\lambda(t) = \lambda L$ for some $L > 0$.*

(v) *There exists $\mu > 0$ such that $p^\mu_\lambda(t) := p_\lambda(t) + \frac{\mu}{2}t^2$ is convex.*

Many commonly used penalty functions $p_\lambda$ satisfy all the conditions in Assumption A.2, such as LASSO, SCAD and MCP penalties. Proofs can be found in [35]. By the result in [35], we have the following proposition for $\hat{\boldsymbol{\theta}}_j$.

**Assumption A.3.** *The RSC condition in [35] is*

$$\{\nabla \mathcal{L}_n(\boldsymbol{\beta}_h + \Delta) - \nabla \mathcal{L}_n(\boldsymbol{\beta}_h)\}^\top \Delta \geq \begin{cases} \alpha_1 \|\Delta\|_2^2 - \tau_1 \dfrac{\log p}{n} \|\Delta\|_1^2, & \forall \|\Delta\|_2 \leq 1, \quad (1.22) \\[3mm] \alpha_2 \|\Delta\|_2 - \tau_2 \sqrt{\dfrac{\log p}{n}} \|\Delta\|_1, & \forall \|\Delta\|_2 \geq 1, \quad (1.23) \end{cases}$$

*where the $\alpha_1$, $\alpha_2$ are strictly positive constants, the $\tau_1$, $\tau_2$ are nonnegative constants and the $\Delta \in \mathbb{R}^p$ is a $p$-dimension column vector.*

Here we denote

$$S_{nj} = \frac{1}{\sqrt{n}} \sum_{i=1}^{n} \left[ \left\{ F(Y_i) - \frac{1}{2} - \mathbf{Z}_{ij}^\top \boldsymbol{\gamma}_j \right\} \left( X_{ij} - \mathbf{Z}_{ij}^\top \boldsymbol{\theta}_j \right) + m_j(Y_i) \right].$$

and $\widetilde{S}_{nj} = \sigma_j^{-1} S_{nj}$, where $m_j(y) = \mathbb{E}[(X_j - \mathbf{Z}_j^\top \boldsymbol{\theta}_j)\{I(Y \geq y) - F(Y)\}]$. It could be shown that $\widetilde{S}_{nj} = \widetilde{T}_{nj} + o_p(1)$ and $\mathrm{Var}(\widetilde{S}_{nj}) = 1$. Denote $\widetilde{\boldsymbol{S}}_n = (\widetilde{S}_{nj})_{j=1}^p$ and $\boldsymbol{\Lambda} = \mathrm{Var}(\widetilde{\boldsymbol{S}}_n)$, where $\Lambda_{jk} = \mathrm{Cov}(\widetilde{S}_{nj}, \widetilde{S}_{nk})$ is $(j, k)$ component of $\boldsymbol{\Lambda}$. And denote $\widetilde{\boldsymbol{S}}_{n,\mathcal{G}} = (\widetilde{S}_{nj})_{j\in\mathcal{G}}$ and $\boldsymbol{\Lambda}_\mathcal{G} = \mathrm{Var}(\widetilde{\boldsymbol{S}}_{n,\mathcal{G}})$ correspondingly. Further denote $\Omega = \{\boldsymbol{\beta}_h \in \mathbb{R}^p : \|\boldsymbol{\beta}_h\|_0 \vee \max_{1\leq j\leq p}\|\boldsymbol{\theta}_j\|_0 \leq s\}$ and $\boldsymbol{\beta}_{h\mathcal{G}} = (\beta_{hj})_{j\in\mathcal{G}}$.

**Assumption A.4.** $\{\log(np_0)\}^7/n = o(1)$.

**Assumption A.5.** $C_0^{-1} \leq \lambda_{\min}(\boldsymbol{\Lambda}) \leq \lambda_{\max}(\boldsymbol{\Lambda}) \leq C_0$ *for some constant $C_0 > 0$.*

**Assumption A.6.** $\max_{1\leq j\neq k\leq p} |\Lambda_{jk}| \leq c_0 < 1$ *for some constant $c_0 > 0$.*

Assumptions A.4-A.6 are mild and commonly used in the high-dimensional settings. Assumption A.4 is the technical condition to bound the difference between $\|\widetilde{\boldsymbol{S}}_{n,\mathcal{G}}\|_\infty$ and $\|\mathbf{N}_\mathcal{G}\|_\infty$, where $\mathbf{N}_\mathcal{G} \sim N_{p_0}(\mathbf{0}, \boldsymbol{\Lambda}_\mathcal{G})$. Specially, Assumption A.4 imposes suitable restrictions on the growth rate of $p_0$ and it is commonly used in the group inference literature, such as [59] and [16]. Assumption A.5 and A.6 are used to establish the limiting distribution of $\|\mathbf{N}_\mathcal{G}\|_\infty$. Assumptions A.5 requires that the eigenvalues of $\boldsymbol{\Lambda}_\mathcal{G}$ are bounded. Assumption A.6 is widely used to establish the limiting distribution of the max type statistic in the high-dimensional settings, see [8], [54], and [36] for further examples.

The validity of the above bootstrap method requires the following assumptions.

**Assumption A.7.** $\{\log(np_0)\}^8/n = o(1)$.

**Assumption A.8.** $\min_{1\leq j\leq p} \mathbb{E}(\widetilde{S}_{nj}^2) \geq C_{\min}$, *where $C_{\min}$ is a positive constant.*

Assumption A.7 and A.8 are the technical assumptions needed for the gaussian multiplier bootstrap method. Assumption A.7 is imposed for controlling the estimation error of gaussian multiplier bootstrap statistic $M_{n,\mathcal{G}}^\sharp$. Theorem A.1 below establishes the honest property of the bootstrap procedure for the statistic $M_{n,\mathcal{G}}$.

Denote $\mathcal{B}(a) = \{(j, k) : |\Lambda_{jk}| \geq a, \ j, k \in \mathcal{H}_0, j \neq k\}$ and $\mathcal{A}(b) = \mathcal{B}((\log p)^{-2-b})$.

**Assumption A.9.** *Suppose that for some $b > 0$ and $q > 0$,*

$$\sum_{(j,k)\in\mathcal{A}(b)} p^{\frac{2|\Lambda_{jk}|}{1+|\Lambda_{jk}|}+q} = O\left(p^2/(\log p)^2\right).$$

**Remark A.2.** *Recall $\Lambda_{jk}$ is the covariance between $\widetilde{S}_{nj}$ and $\widetilde{S}_{nk}$. Thus, $\mathcal{A}(b)$ contains all the strongly correlated pairs and Assumption A.9 requires that the number of these pairs cannot be too large. If $|\Lambda_{jk}| \leq a_0$ for some constant $0 < a_0 \leq 1$, then this assumption holds under $|\mathcal{A}(b)| = O\left(p^{\frac{2}{1+a_0}-q}/(\log p)^2\right)$ for some $b > 0$ and $q > 0$. Similar assumptions were considered in [10], [20] and [36].*

## A.9 Proofs of theorems

**The proof of Theorem 3.1:** We denote

$$\tilde{\boldsymbol{\beta}}_h = \arg \min_{\boldsymbol{\beta}_h \in \mathbb{R}^p} \mathbb{E}\mathcal{L}_n(\boldsymbol{\beta}_h)$$

where $\mathcal{L}_n(\boldsymbol{\beta}_h)$ is defined in equation (2.6). By triangle inequality of $l_1$ and $l_2$ norms, it suffices to show that the $l_1$ and $l_2$ norms of $\hat{\boldsymbol{\beta}}_h - \tilde{\boldsymbol{\beta}}_h$ and $\tilde{\boldsymbol{\beta}}_h - \boldsymbol{\beta}_h$ satisfy the requirements in this theorem respectively.

Firstly we calculate $l_1$ and $l_2$ norms of difference between $\hat{\boldsymbol{\beta}}_h$ and $\tilde{\boldsymbol{\beta}}_h$. The results are presented in Lemma A.2.

Secondly, by Lemma A.8 in Supplementary Material and Theorem 1 in [43], we can show that $\tilde{\boldsymbol{\beta}}_h = (n-1)\boldsymbol{\beta}_h/n$, and thus

$$\|\tilde{\boldsymbol{\beta}}_h - \boldsymbol{\beta}_h\|_2 = \frac{1}{n}\|\boldsymbol{\beta}_h\|_2, \quad \text{and} \quad \|\tilde{\boldsymbol{\beta}}_h - \boldsymbol{\beta}_h\|_1 = \frac{1}{n}\|\boldsymbol{\beta}_h\|_1.$$

By the sparse assumption, $\|\boldsymbol{\beta}_h\|_2/n = o(\sqrt{s_Y \log p/n})$ and $\|\boldsymbol{\beta}_h\|_1/n = o(s_Y \sqrt{\log p/n})$. Thus Theorem 3.1 is proved.

**The proof of Theorem 3.2:** By Lemma A.3, the distribution of $\max_{j \in \mathcal{G}} |\widetilde{T}_{nj}|$ can be approximated by $\|\mathbf{N}_{\mathcal{G}}\|_\infty$ with $\mathbf{N}_{\mathcal{G}} \sim N_{p_0}(\mathbf{0}, \boldsymbol{\Lambda}_{\mathcal{G}})$. By Lemma 6 of [8], we obtain

$$\lim_{(n,p_0)\to\infty} \left| \Pr\left(M_{n,\mathcal{G}} - 2\log p_0 + \log\log p_0 \le t\right) - \exp\left\{-\frac{1}{\sqrt{\pi}} \exp(-t/2)\right\} \right| = 0.$$

Note that all the universal constants do not depend on $n$, $p$, $\boldsymbol{\beta}_h$ and $\boldsymbol{\theta}_j$, $j = 1, \ldots, p$. Thus this theorem is proved.

**The proof of Theorem A.1:** For the convenience of the proof, we introduce some notations. Denote $U_{nj}$ as

$$U_{nj} = \frac{1}{n(n-1)} \sum_{1 \le i \ne \tilde{i} \le n} h_j^{sym}(Y_i, \mathbf{X}_i; Y_{\tilde{i}}, \mathbf{X}_{\tilde{i}}), \tag{1.24}$$

where

$$h_j^{sym}(Y_i, \mathbf{X}_i; Y_{\tilde{i}}, \mathbf{X}_{\tilde{i}}) = \frac{1}{2}\left\{ h_j(Y_i, \mathbf{X}_i; Y_{\tilde{i}}, \mathbf{X}_{\tilde{i}}) + h_j(Y_{\tilde{i}}, \mathbf{X}_{\tilde{i}}; Y_i, \mathbf{X}_i) \right\}, \tag{1.25}$$

and

$$h_j(Y_i, \mathbf{X}_i; Y_{\tilde{i}}, \mathbf{X}_{\tilde{i}}) = \left\{ I(Y_{\tilde{i}} \le Y_i) - \frac{1}{2} - \mathbf{Z}_{ij}^\top \boldsymbol{\gamma}_j \right\}\left( X_{ij} - \mathbf{Z}_{ij}^\top \boldsymbol{\theta}_j \right). \tag{1.26}$$

Denote $\widetilde{U}_{nj} =: \sigma_j^{-1} U_{nj}$. $U_{nj}$ plays an important role in the proof of this theorem.

Let $w_1, \ldots, w_n$ be i.i.d. $N(0,1)$ random variables that are independent of $\{\mathbf{X}_i, Y_i\}_{i=1}^n$ and denote $L_{n,\mathcal{G}}^\sharp$ as :

$$L_{n,\mathcal{G}}^\sharp = \max_{j \in \mathcal{G}} \left[ \frac{2}{\sigma_j \sqrt{n}} \sum_{i=1}^n \left\{ \frac{1}{n-1} \sum_{\tilde{i} \ne i} h_j^{sym}(Y_i, \mathbf{X}_i; Y_{\tilde{i}}, \mathbf{X}_{\tilde{i}}) - U_{nj} \right\} w_i \right]^2, \tag{1.27}$$

where $U_{nj}$ and $h_j^{sym}(\cdot)$ are defined in (1.24) and (1.25). The proof follows from the general results for the multiplier bootstrap of $U$-statistic in [12]. Here, we extend these results for approximate high-dimensional $U$-statistic by results in [13] and rewrite it in a suitable form for our analysis. It is now presented in Lemma A.22 in the Supplementary Material. Suppose $\zeta_1$ is a sequence satisfying $\zeta_1 \sqrt{1 \vee \log(p_0/\zeta_1)} \to 0$ as $n \to \infty$. According to Lemma A.22, to prove this theorem, it suffices to prove $\left| \|\widetilde{\boldsymbol{T}}_{n,\mathcal{G}}\|_\infty - \|\widetilde{\boldsymbol{S}}_{n,\mathcal{G}}\|_\infty \right| = o_p(\zeta_1)$ and $\Pr_w\left( \left| \sqrt{M_{n,\mathcal{G}}^\sharp} - \sqrt{L_{n,\mathcal{G}}^\sharp} \right| \ge \zeta_1 \right) \to 0$. We denote

$\zeta_1 = C(\log(np_0))^{-1/2}$, where $C$ is a sufficient large constant. The proofs of these results are presented in Lemmas A.17, A.18 in the Supplementary Material.

With Lemmas A.17, A.18 and A.22 in the Supplementary Material, we obtain

$$\sup_{\alpha \in (0,1)} \left| P\big(M_{n,\mathcal{G}} > c_{\mathcal{G}}^{(2)}(\alpha)\big) - \alpha \right| = o(1)$$

when $\mathbb{H}_{0,\mathcal{G}}$ holds. Note that all the universal constants in the proof do not depend on $n$, $p$, $\boldsymbol{\beta}_h$ and $\boldsymbol{\theta}_j$, $j = 1, \ldots, p$. We thus have

$$\sup_{\boldsymbol{\beta}_h \in \Omega_{\mathcal{G}}^0} \sup_{\alpha \in (0,1)} \left| P\big(M_{n,\mathcal{G}} > c_{\mathcal{G}}^{(2)}(\alpha)\big) - \alpha \right| = o(1).$$

**The proof of Theorem 3.3:** Using the inequality $2a_1a_2 \le \epsilon^{-1}a_1^2 + \epsilon a_2^2$ for any $\epsilon > 0$, we have

$$\max_{j \in \mathcal{G}} \big(\hat{\sigma}_j^{-1} \sqrt{n} \beta_{hj} \delta_j \big)^2 \le (1+\epsilon) M_{n,\mathcal{G}} + (1 + \epsilon^{-1}) \max_{j \in \mathcal{G}} \hat{\sigma}_j^{-2}(T_{nj} - \sqrt{n}\beta_{hj}\delta_j)^2. \qquad (1.28)$$

Denote $\eta_{\mathcal{G}} = |\max_{j \in \mathcal{G}} \hat{\sigma}_j^{-2}(T_{nj} - \sqrt{n}\beta_{hj}\delta_j)^2 - \max_{j \in \mathcal{G}} \sigma_j^{-2}(S_{nj} - \sqrt{n}\beta_{hj}\delta_j)^2|$, and $\eta_{\mathcal{G}} = o_p((\log p_0)^{-1/2})$ by Lemma A.16 in the Supplementary Material. We can derive

$$\lim_{(n,p_0) \to \infty} \inf_{\boldsymbol{\beta}_h \in \Omega_{\mathcal{G}}^1(2+\epsilon_0)} \Pr\big(\max_{j \in \mathcal{G}} \hat{\sigma}_j^{-2}(T_{nj} - \sqrt{n}\beta_{hj}\delta_j)^2 - 2\log p_0 + \log\log p_0 \le t\big)$$

$$\ge \lim_{(n,p_0) \to \infty} \inf_{\boldsymbol{\beta}_h \in \Omega_{\mathcal{G}}^1(2+\epsilon_0)} \Pr\big(\max_{j \in \mathcal{G}} \sigma_j^{-2}(S_{nj} - \sqrt{n}\beta_{hj}\delta_j)^2 - 2\log p_0 + \log\log p_0 \le t - \eta_{\mathcal{G}}\big)$$

$$\ge \lim_{(n,p_0) \to \infty} \inf_{\boldsymbol{\beta}_h \in \Omega_{\mathcal{G}}^1(2+\epsilon_0)} \Pr\big(\max_{j \in \mathcal{G}} (\sigma_j^2 - \beta_{hj}^2\delta_j^2)^{-1}(S_{nj} - \sqrt{n}\beta_{hj}\delta_j)^2 - 2\log p_0 + \log\log p_0 \le t - \eta_{\mathcal{G}}\big)$$

$$= \exp\left\{-\frac{1}{\sqrt{\pi}} \exp(-t/2)\right\}.$$

The last equality holds by Lemma A.19, A.20 in the Supplementary Material and Lemma 6 of [8]. Denote $t = \log\log p_0/2$, it implies that

$$\lim_{(n,p_0) \to \infty} \inf_{\boldsymbol{\beta}_h \in \Omega_{\mathcal{G}}^1(2+\epsilon_0)} \Pr\big(\max_{j \in \mathcal{G}} \hat{\sigma}_j^{-2}(T_{nj} - \sqrt{n}\beta_{hj}\delta_j)^2 \le 2\log p_0 - \log\log p_0/2\big) = 1. \qquad (1.29)$$

For any $\boldsymbol{\beta}_h \in \Omega_{\mathcal{G}}^1(2+\epsilon_0)$, applying Lemma A.14 in the Supplementary Material, we derive

$$\max_{j \in \mathcal{G}} \big(\hat{\sigma}_j^{-1}\sqrt{n}\beta_{hj}\delta_j\big)^2 = (1 + o(1)) \max_{j \in \mathcal{G}} \big(\sigma_j^{-1}\sqrt{n}\beta_{hj}\delta_j\big)^2 > (2+\epsilon_0)\log p_0 \qquad (1.30)$$

holds with probability 1 as $(n, p_0) \to 1$.

Substitute (1.29) and (1.30) into (1.28), we derive that

$$M_{n,\mathcal{G}} \ge \frac{1}{1+\epsilon} \max_{j \in \mathcal{G}} \big(\hat{\sigma}_j^{-1}\sqrt{n}\beta_{hj}\delta_j\big)^2 - \frac{1}{\epsilon}\max_{j \in \mathcal{G}} \hat{\sigma}_j^{-2}(T_{nj} - \sqrt{n}\beta_{hj}\delta_j)^2$$

$$> \frac{(2+\epsilon_0)\log p_0}{1+\epsilon} - \frac{2\log p_0}{\epsilon} + \frac{\log\log p_0}{2\epsilon}.$$

Now we prove the validity of $c_{\mathcal{G}}^{(i)}(\alpha)$, $i = 1, 2$. Note that $c_{\mathcal{G}}^{(1)}(\alpha) \le 2\log p_0$ by the definition of $c_{\mathcal{G}}^{(1)}(\alpha)$. By the properties of Gaussian multiplier bootstrap statistic(Lemma A.22), $c_{\mathcal{G}}^{(2)}(\alpha)$ is equal to the $(1-\alpha)$th quantile of $\max_{j \in \mathcal{G}} \hat{\sigma}_j^{-2}(T_{nj} - \sqrt{n}\beta_{hj}\delta_j)^2$ asymptotically. Thus $\lim_{(n,p_0) \to \infty} \inf_{\boldsymbol{\beta}_h \in \Omega_{\mathcal{G}}^1(2+\epsilon_0)} \Pr\big(2\log p_0 > c_{\mathcal{G}}^{(2)}(\alpha)\big) = 1$ by (1.29). By choosing $(\epsilon_0, \epsilon)$ satisfying $\frac{2+\epsilon_0}{1+\epsilon} - \frac{2}{\epsilon} \ge 2$,

$$\lim_{(n,p_0) \to \infty} \inf_{\boldsymbol{\beta}_h \in \Omega_{\mathcal{G}}^1(2+\epsilon_0)} \Pr\big(M_{n,\mathcal{G}} > c_{\mathcal{G}}^{(i)}(\alpha)\big) = 1, \quad i = 1, 2$$

can be proved.

**The proof of Theorem A.3:** At first we calculate the uniform bound of the difference between constructed statistic $\widetilde{T}_{nj}$ and its ideal version $\widetilde{S}_{nj}$. The result is presented in Lemma A.4.

For (1.19), by Lemma 6.1 in [34], we have

$$\max_{1\leq j\leq p}\sup_{0\leq t\leq 2\sqrt{\log p}}\left|\frac{\Pr\big(|\widetilde{S}_{nj}|\geq t\big)}{G(t)}-1\right|\leq C(\log p)^{-2-\gamma_1} \tag{1.31}$$

for some constant $0<\gamma_1<1/2$. So (1.19) follows from Lemma A.4, and the fact that $G(t+o((\log p)^{-1/2}))/G(t)=1+o(1)$ uniformly in $0\leq t\leq\sqrt{2\log p}$.

For (1.20), it suffices to show that

$$\sup_{0\leq t\leq b_p}\left|\frac{\sum_{j\in\mathcal{H}_0}I(|\widetilde{S}_{nj}|\geq t)}{q_0 G(t)}-1\right|\to 0 \quad\text{in probability.} \tag{1.32}$$

Let $z_0<z_1<\ldots<z_{d_p}\leq 1$ and $t_i=G^{-1}(z_i)$, where $z_0=G(b_p)$, $z_i=c_p/p+c_p^{2/3}e^{i^\nu}/p$ with $c_p=pG(b_p)$, and $d_p=\left\{\log\frac{(p-c_p)}{c_p^{2/3}}\right\}^{1/\nu}$ and $0<\nu<1$, which will be specified later. We have $G(t_i)/G(t_{i+1})=1+o(1)$ uniformly in $i$, and $t_0/\sqrt{2\log(p/c_p)}=1+o(1)$. Note that uniformly for $1\leq j\leq m$, $G(t_i)/G(t_{i-1})\to 1$ as $p\to\infty$. The proof of (1.32) reduces to show that

$$\max_{0\leq i\leq d_p}\left|\frac{\sum_{j\in\mathcal{H}_0}I(|\widetilde{S}_{nj}|\geq t_i)}{q_0 G(t_i)}-1\right|\to 0 \quad\text{in probability.} \tag{1.33}$$

In fact, for each $\epsilon>0$, we have

$$\Pr\left(\max_{0\leq i\leq d_p}\left|\frac{\sum_{j\in\mathcal{H}_0}\{I(|\widetilde{S}_{nj}|\geq t_i)-G(t_i)\}}{q_0 G(t_i)}\right|\geq\epsilon\right)\leq\sum_{j=0}^{d_p}\Pr\left(\left|\frac{\sum_{j\in\mathcal{H}_0}\{I(|\widetilde{S}_{nj}|\geq t_i)-G(t_i)\}}{q_0 G(t_i)}\right|\geq\epsilon/2\right).$$

Set $I(t)=\frac{\sum_{j\in\mathcal{H}_0}\{I(|\widetilde{S}_{nj}|\geq t)-\Pr(|\widetilde{S}_{nj}|\geq t)\}}{q_0 G(t)}$. By Markov's inequality $\Pr(|I(t_i)|\geq\epsilon/2)\leq\frac{\mathbb{E}\{I(t_i)\}^2}{\epsilon^2/4}$, and it suffices to show $\sum_{j=0}^{d_p}\mathbb{E}\{I(t_i)\}^2=o(1)$. To see this, by (1.31),

$$\mathbb{E}I^2(t)=\frac{\sum_{j\in\mathcal{H}_0}\{\Pr(|\widetilde{S}_{nj}|\geq t)-\Pr^2(|\widetilde{S}_{nj}|\geq t)\}}{q_0^2 G^2(t)}$$

$$+\frac{\sum_{j,k\in\mathcal{H}_0,k\neq j}\{\Pr(|\widetilde{S}_{nj}|\geq t,|\widetilde{S}_{nk}|\geq t)-\Pr(|\widetilde{S}_{nj}|\geq t)\Pr(|\widetilde{S}_{nk}|\geq t)\}}{q_0^2 G^2(t)}$$

$$\leq\frac{C}{q_0 G(t)}+\frac{1}{q_0^2}\sum_{(j,k)\in\mathcal{A}(b):j,k\in\mathcal{H}_0}\frac{\Pr(|\widetilde{S}_{nj}|\geq t,|\widetilde{S}_{nk}|\geq t)}{G^2(t)}$$

$$+\frac{1}{q_0^2}\sum_{(j,k)\in\mathcal{A}(b)^c:j,k\in\mathcal{H}_0}\left\{\frac{\Pr(|\widetilde{S}_{nj}|\geq t,|\widetilde{S}_{nk}|\geq t)}{G^2(t)}-1\right\}$$

$$=\frac{C}{q_0 G(t)}+I_{11}(t)+I_{12}(t).$$

For $(j,k)\in\mathcal{A}(b)^c$ with $j,k\in\mathcal{H}_0$, applying Lemma 6.1 in [34], we have $I_{12}(t)\leq C(\log p)^{-1-\xi}$ for some $\xi>0$ uniformly in $0<t<\sqrt{2\log p}$. By Lemma 6.2 in [34], for $(j,k)\in\mathcal{A}(b)$ with $j,k\in\mathcal{H}_0$, we have

$$\Pr(|\widetilde{S}_{nj}|\geq t,|\widetilde{S}_{nk}|\geq t)\leq C(t+1)^{-2}\exp\left(-\frac{t^2}{1+|\Lambda_{jk}|}\right).$$

So that

$$I_{11}(t)\leq C\frac{1}{q_0^2}\sum_{(j,k)\in\mathcal{A}(b):j,k\in\mathcal{H}_0}(t+1)^{-2}\exp\left(-\frac{t^2}{1+|\Lambda_{jk}|}\right)G^{-2}(t)\leq C\frac{1}{q_0^2}\sum_{(j,k)\in\mathcal{A}(b):j,k\in\mathcal{H}_0}G(t)^{-\frac{2|\Lambda_{jk}|}{1+|\Lambda_{jk}|}}.$$

Note that for $0 \leq t \leq b_p$, we have $G(t) \geq G(b_p) = c_p/p$, so that by assumption A.9 it follows that for some $b, q > 0$,

$$I_{11}(t) \leq C \sum_{(j,k) \in \mathcal{A}(b): j, k \in \mathcal{H}_0} p^{\frac{2|\Lambda_{jk}|}{1+|\Lambda_{jk}|}+q-2} = O\big(1/(\log p)^2\big).$$

By the above inequalities, we can prove (1.33) by choosing $0 < \nu < 1$ so that

$$\sum_{i=0}^{d_p} \mathbb{E}\{I(t_i)\}^2 \leq C \sum_{i=0}^{d_p}\{pG(t_i)\}^{-1} + Cd_p\{(\log p)^{-1-\nu} + (\log p)^{-2}\}$$

$$\leq C \sum_{i=0}^{d_p} \frac{1}{c_p + c_p^{2/3}e^{i\nu}} + o(1)$$

$$= o(1).$$

**The proof of Theorem A.4:** We first consider the case when $\hat{t}$, given by (1.18), doesn't exist. In this case, $\hat{t} = \sqrt{2 \log p}$ and we consider the event $\mathcal{F}_0 = \{\sum_{j \in \mathcal{H}_0} I(|\tilde{T}_{nj}| \geq \sqrt{2 \log p}) \geq 1\}$, which mean at least one false positive. In order to show the FDR/FDP can be controlled in this case, we show that

$$\Pr(\mathcal{F}_0) \to 0, \qquad as\ (n, p) \to \infty. \tag{1.34}$$

And we have

$$\Pr(\mathcal{F}_0) \leq \Pr\bigg(\sum_{j \in \mathcal{H}_0} I(\tilde{T}_{nj} \geq \sqrt{2 \log p}) \geq 1\bigg) + \Pr\bigg(\sum_{j \in \mathcal{H}_0} I(\tilde{T}_{nj} \leq -\sqrt{2 \log p}) \geq 1\bigg)$$

$$=: \Pr(\mathcal{F}_1) + \Pr(\mathcal{F}_2). \tag{1.35}$$

For any $\epsilon > 0$, we can bound the first term by

$$\Pr(\mathcal{F}_1) = \Pr\bigg(\sum_{j \in \mathcal{H}_0} I(\tilde{S}_{nj} + \tilde{T}_{nj} - \tilde{S}_{nj} \geq \sqrt{2 \log p}) \geq 1\bigg)$$

$$\leq \Pr\bigg(\sum_{j \in \mathcal{H}_0} I(\tilde{S}_{nj} \geq \sqrt{2 \log p} - \epsilon) \geq 1\bigg) + \Pr\bigg(\max_{j \in \mathcal{H}_0}|\tilde{T}_{nj} - \tilde{S}_{nj}| \geq \epsilon\bigg)$$

$$\leq p \max_{j \in \mathcal{H}_0}\Pr\bigg(\tilde{S}_{nj} \geq \sqrt{2 \log p} - \epsilon\bigg) + \Pr\bigg(\max_{j \in \mathcal{H}_0}|\tilde{T}_{nj} - \tilde{S}_{nj}| \geq \epsilon\bigg).$$

By Lemma A.4, we know that $\Pr(\max_{j \in \mathcal{H}_0}|\tilde{T}_{nj} - \tilde{S}_{nj}| \geq \epsilon) \to 0$. For simplify, we rewrite $\tilde{S}_{nj} = \sum_{i=1}^n \xi_{ij}/\sqrt{n}$. Since $\mathbb{E}(\xi_{ij}) = 0$, $\mathrm{Var}(\xi_{ij}) = 1$ and $\{\xi_{ij}\}_{i=1}^n$ is a i.i.d. sequence, by Lemma 6.1 of [34], we have $\sup_{0 \leq t \leq 2\sqrt{\log p}}\left|\frac{\Pr(|\tilde{S}_{nj}| \geq t)}{G(t)} - 1\right| \leq C(\log p)^{-1}$. Now let $t = \sqrt{2 \log p} - \epsilon$, we have

$$\Pr\big(\tilde{S}_{nj} \geq \sqrt{2 \log p} - \epsilon\big) \leq G(\sqrt{2 \log p} - \epsilon) + C\frac{G(\sqrt{2 \log p} - \epsilon)}{\log p}$$

for $j \in \mathcal{H}_0$ uniformly. Hence $p \max_{j \in \mathcal{H}_0}\Pr\big(\tilde{S}_{nj} \geq \sqrt{2 \log p} - \epsilon\big) \leq (C+1)pG(\sqrt{2 \log p} - \epsilon)$, which goes to zero as $(n, p) \to \infty$. By symmetry, we know that $\Pr(\mathcal{F}_2)$ in (1.35) also goes to 0. Therefore (1.34) is proved.

Now consider the case when $0 \leq \hat{t} \leq b_p$ holds. We have

$$\mathrm{FDP}(\hat{t}) = \frac{\sum_{j \in \mathcal{H}_0} I(|\tilde{T}_{nj}| \geq \hat{t})}{\max\{\sum_{j=1}^p I(|\tilde{T}_{nj}| \geq \hat{t}), 1\}} \leq \frac{q_0 G(\hat{t})}{\max\{\sum_{j=1}^p I(|\tilde{T}_{nj}| \geq \hat{t}), 1\}}(1 + A_p),$$

where $A_p = \sup_{0 \leq t \leq b_p}\left|\frac{\sum_{j \in \mathcal{H}_0} I(|\tilde{T}_{nj}| \geq t)}{q_0 G(t)} - 1\right|$. Note that by definition $\frac{q_0 G(\hat{t})}{\max\{\sum_{j=1}^p I(|\tilde{T}_{nj}| \geq \hat{t}), 1\}} \leq \frac{q_0 \alpha}{p}$. The proof is complete if $A_p \to 0$ in probability, which has been shown by Theorem A.3.

## A.10 Some technical lemmas

For the convenience of the proof, we introduce some definitions. Denote $U_{nj}$ as

$$U_{nj} = \frac{1}{n(n-1)} \sum_{1 \leq i \neq \tilde{i} \leq n} h_j(Y_i, \mathbf{X}_i; Y_{\tilde{i}}, \mathbf{X}_{\tilde{i}}),$$

where

$$h_j(Y_i, \mathbf{X}_i; Y_{\tilde{i}}, \mathbf{X}_{\tilde{i}}) = \frac{1}{2} \left[ \left\{ I(Y_{\tilde{i}} \leq Y_i) - \frac{1}{2} - \mathbf{Z}_{ij}^\top \boldsymbol{\gamma}_j \right\} \left( X_{ij} - \mathbf{Z}_{ij}^\top \boldsymbol{\theta}_j \right) \right. \tag{1.36}$$
$$\left. + \left\{ I(Y_i \leq Y_{\tilde{i}}) - \frac{1}{2} - \mathbf{Z}_{\tilde{i}j}^\top \boldsymbol{\gamma}_j \right\} \left( X_{\tilde{i}j} - \mathbf{Z}_{\tilde{i}j}^\top \boldsymbol{\theta}_j \right) \right].$$

And denote $\widetilde{U}_{nj} =: \sigma_j^{-1} U_{nj}$. $U_{nj}$ plays an important role in the proof of main results. Let $s_X$ be the sparsity level for parameter $\boldsymbol{\theta}_j$ as $s_X = \max_{1 \leq j \leq p} \|\boldsymbol{\theta}_j\|_0$.

**Lemma A.1.** *Under Assumptions A.1 and A.2, the $\hat{\boldsymbol{\theta}}_j$ defined in (2.5) satisfies*

$$\|\hat{\boldsymbol{\theta}}_j - \boldsymbol{\theta}_j\|_1 \leq c\lambda_X s_X$$

*with probability at least $1 - c_1 \exp(-c_2 \log p)$. Here, $(c, c_1, c_2)$ are universal constants.*

**Lemma A.2.** *Under assumptions in Theorem 3.1, $\hat{\boldsymbol{\beta}}_h$ defined in (3.11) satisfies*

$$\|\hat{\boldsymbol{\beta}}_h - \tilde{\boldsymbol{\beta}}_h\|_2 \leq c_0 \lambda_Y \sqrt{s_Y}, \quad and \quad \|\hat{\boldsymbol{\beta}}_h - \tilde{\boldsymbol{\beta}}_h\|_1 \leq c_0' \lambda_Y s_Y$$

*with probability at least $1 - c_1 \exp(-c_2 \log p)$. Here $\tilde{\boldsymbol{\beta}}_h = \arg\min_{\boldsymbol{\beta}_h \in \mathbb{R}^p} \mathbb{E}\mathcal{L}_n(\boldsymbol{\beta}_h)$ with $\mathcal{L}_n(\boldsymbol{\beta}_h)$ defined in equation (2.6).*

*Proof.* The main part of the proof is to verify RSC condition in assumption A.3 and get exponential inequality of $\|\nabla\mathcal{L}_n(\boldsymbol{\beta}_h)\|_\infty$. Based on these results, we use Lemma A.5 to prove theorem A.2.

Thus, we focus on RSC condition at the beginning. Note that for $\Delta \in \mathbb{R}^p$,

$$\mathcal{E}_n(\Delta) = \{\nabla\mathcal{L}_n(\boldsymbol{\beta}_h + \Delta) - \nabla\mathcal{L}_n(\boldsymbol{\beta}_h)\}^\top \Delta$$
$$= \frac{1}{n} \sum_{i=1}^n (\mathbf{X}_i^\top \Delta)^2.$$

From Proportion 2 in [38], we then have

$$\mathcal{E}_n(\Delta) \geq \alpha_1 \|\Delta\|_2^2 - \tau_1 \sqrt{\frac{\log p}{n}} \|\Delta\|_1 \|\Delta\|_2 \qquad \forall \|\Delta\|_2 \leq 1 \tag{1.37}$$

with probability at least $1 - c_1 \exp\{-c_2 n\}$ for an appropriate choice of $\alpha_1$. By the arithmetic mean-geometric mean inequality,

$$\tau_1 \sqrt{\frac{\log p}{n}} \|\Delta\|_1 \|\Delta\|_2 \leq \frac{\alpha_1}{2} \|\Delta\|_2^2 + \frac{\tau_1^2}{2\alpha_1} \frac{\log p}{n} \|\Delta\|_1^2,$$

and consequently,

$$\mathcal{E}_n(\Delta) \geq \frac{\alpha_1}{2} \|\Delta\|_2^2 - \frac{\tau_1^2}{2\alpha_1} \frac{\log p}{n} \|\Delta\|_1^2,$$

which establishes (1.22) in assumption A.3. As square loss function is convex, condition (1.23) then follows via Lemma 8 in [35]. So we verify RSC condition completely.

Next, we construct exponential inequalities of $\nabla\mathcal{L}_n(\boldsymbol{\beta}_h)$. By conditions of lemma A.5, it suffices to show that there exist universal constants $(c, c_1, c_2)$ such that

$$\Pr\left( \|\nabla\mathcal{L}_n(\tilde{\boldsymbol{\beta}}_h)\|_\infty \geq c\sqrt{\frac{\log p}{n}} \right) \leq c_1 \exp(-c_2 \log p). \tag{1.38}$$

By lemma A.7, we can choose proper $(c, c_1, c_2)$ such that both RSC condition and inequality (1.38) are satisfied. As $\lambda_Y$ is proportional to $\sqrt{\frac{\log p}{n}}$, the claimed $l_1$ and $l_2$ bounds then follow directly from Lemma A.5 by choosing $R$ proportional to $\frac{1}{\lambda_Y}$. $\qquad\square$

**Lemma A.3.** *Under the assumptions in Theorem 3.2,*

$$\lim_{n\to\infty} \sup_{\boldsymbol{\beta}_h\in\Omega_{\mathcal{G}}^0} \sup_{t\in\mathbb{R}} \big|\Pr\big(\max_{j\in\mathcal{G}}|\widetilde{T}_{nj}| \le t\big) - \Pr\big(\|\mathbf{N}_{\mathcal{G}}\|_\infty \le t\big)\big| = 0,$$

*where* $\mathbf{N}_{\mathcal{G}} \sim N_{p_0}(\mathbf{0}, \boldsymbol{\Lambda}_{\mathcal{G}})$.

*Proof.* By Lemma A.16, we obtain

$$\big|\max_{j\in\mathcal{G}}|\widetilde{T}_{nj}| - \max_{j\in\mathcal{G}}|\widetilde{S}_{nj}|\big| \le \max_{j\in\mathcal{G}}|\widetilde{T}_{nj} - \widetilde{S}_{nj}| \le Cq(n,p)$$

for some constant $C$, with probability 1 as $(n,p) \to \infty$. And $q(n,p)$ is a sequence satisfying $q(n,p) = o((\log p_0)^{-1/2})$. This implies that

$$\Pr\big(\max_{j\in\mathcal{G}}|\widetilde{T}_{nj}| \le t\big) - \Pr\big(\|\mathbf{N}_{\mathcal{G}}\|_\infty \le t\big) \le \Pr\big(\max_{j\in\mathcal{G}}|\widetilde{S}_{nj}| \le t + Cq(n,p)\big) - \Pr\big(\|\mathbf{N}_{\mathcal{G}}\|_\infty \le t + Cq(n,p)\big)$$
$$+ \Pr\big(\|\mathbf{N}_{\mathcal{G}}\|_\infty \le t + Cq(n,p)\big) - \Pr\big(\|\mathbf{N}_{\mathcal{G}}\|_\infty \le t\big)$$
$$=:I_1 + I_2.$$

By *sub-Gaussian* condition in Assumption A.1 and Assumption A.8, $\limsup_{n\to\infty} \sup_{t\in\mathbb{R}} I_1 \le 0$. By the Gaussian anti-concentration inequality in Lemma A.20, we derive that $\sup_{t\in\mathbb{R}} I_2 \le Cq(n,p)\sqrt{1 \vee \log(p_0/q(n,p))}$ for some constant $C$. As $q(n,p) = o((\log p_0)^{-1/2})$, we derive that

$$\limsup_{n\to\infty} \sup_{t\in\mathbb{R}} \big(\Pr\big(\max_{j\in\mathcal{G}}|\widetilde{T}_{nj}| \le t\big) - \Pr\big(\|\mathbf{N}_{\mathcal{G}}\|_\infty \le t\big)\big) \le 0.$$

Similarly,

$$\liminf_{n\to\infty} \inf_{t\in\mathbb{R}} \big(\Pr\big(\max_{j\in\mathcal{G}}|\widetilde{T}_{nj}| \le t\big) - \Pr\big(\|\mathbf{N}_{\mathcal{G}}\|_\infty \le t\big)\big) \ge 0.$$

This completes the proof. $\qquad\square$

**Lemma A.4.** *Under the assumptions in Theorem A.3,*

$$\max_{j\in\mathcal{H}_0} |\widetilde{T}_{nj} - \widetilde{S}_{nj}| = o_p\big(\frac{1}{\sqrt{\log p}}\big).$$

*Proof.* The proof of this Lemma is similar to Lemma A.16. We only need replace $\mathcal{G}$ with $\mathcal{H}_0$ and replace $p_0$ with $p$. $\qquad\square$

**Lemma A.5.** *(Theorem 1 in [35]) Suppose the regularizer $\rho_\lambda$ satisfies Assumption A.2, the empirical loss $\mathcal{L}_n$ satisfies the RSC condition with $\frac{3}{4}\mu < \alpha_1$, and $\beta^*$ is feasible for the objective. Consider any choice of $\lambda$ such that*

$$\frac{4}{L} \cdot \max\bigg\{\|\nabla\mathcal{L}_n(\beta^*)\|_\infty, \alpha_2\sqrt{\frac{\log p}{n}}\bigg\} \le \lambda \le \frac{\alpha_2}{6RL} \tag{1.39}$$

*and suppose $n \ge \frac{16R^2\max(\tau_1^2,\tau_2^2)}{\alpha_2^2}\log p$. Then any vector $\tilde{\beta}$ satisfying the first-order necessary condition (3.11) satisfies the error bounds*

$$\|\hat{\beta} - \beta^*\|_2 \le \frac{6\lambda L\sqrt{k}}{4\alpha_1 - 3\mu}, \quad \text{and} \quad \|\hat{\beta} - \beta^*\|_1 \le \frac{24\lambda Lk}{4\alpha_1 - 3\mu} \tag{1.40}$$

*where $k = \|\beta^*\|_0$.*

**Lemma A.6.** *Consider a U-statistics*

$$U = \frac{1}{n(n-1)} \sum_{i \neq j} h(Z_i, Z_j)$$

*with a kernel h based on i.i.d. random variables $Z_1, \ldots, Z_n$. Then for $\forall s \in \mathbb{R}$,*

$$\mathbb{E} \exp\{s(U - \mathbb{E}U)\} \leq \mathbb{E} \exp\left[\frac{s}{N} \sum_{i=1}^{N} \{h(Z_i, Z_{i+N}) - \mathbb{E}U\}\right]$$

*where $N = \lfloor \frac{n}{2} \rfloor$, which represents the greatest integer less than or equal to $\frac{n}{2}$.*

*Proof.* It can be seen in the proof of Lemma 14 in [43]. □

**Lemma A.7.** *Under the Assumptions A.1 and A.2, there exist universal constants $(c, c_1, c_2)$ such that*

$$\Pr\left(\|\nabla \mathcal{L}_n(\tilde{\boldsymbol{\beta}}_h)\|_\infty \geq c\sqrt{\frac{\log p}{n}}\right) \leq c_1 \exp(-c_2 \log p), \tag{1.41}$$

*where $\nabla \mathcal{L}_n(\tilde{\boldsymbol{\beta}}_h)$ is the gradient of $\mathcal{L}_n(\tilde{\boldsymbol{\beta}}_h)$, which is a p-dimension column vector.*

*Proof.* Suppose $\nabla_k \mathcal{L}_n(\tilde{\boldsymbol{\beta}}_h)$ is the k-th component of $\nabla \mathcal{L}_n(\tilde{\boldsymbol{\beta}}_h)$. Note that

$$
\begin{aligned}
\nabla_k \mathcal{L}_n(\tilde{\boldsymbol{\beta}}_h) &= \frac{1}{n} \sum_{i=1}^{n} X_{ik}\left[\mathbf{X}_i^\top \tilde{\boldsymbol{\beta}}_h - \{F_n(Y_i) - \frac{1}{2}\}\right] \\
&= \frac{1}{n} \sum_{i=1}^{n} X_{ik}\left[\mathbf{X}_i^\top \tilde{\boldsymbol{\beta}}_h - \{F(Y_i) - \frac{1}{2}\}\right] - X_{ik}\{F_n(Y_i) - F(Y_i)\} \\
&= \frac{1}{n} \sum_{i=1}^{n} X_{ik}\left[\mathbf{X}_i^\top \tilde{\boldsymbol{\beta}}_h - \{F(Y_i) - \frac{1}{2}\}\right] - \frac{1}{n^2} \sum_{i=1}^{n} \sum_{\tilde{i}=1}^{n} X_{ik}\{I(Y_{\tilde{i}} \leq Y_i) - F(Y_i)\} \\
&= \frac{1}{n} \sum_{i=1}^{n} X_{ik}\left[\mathbf{X}_i^\top \tilde{\boldsymbol{\beta}}_h - \{F(Y_i) - \frac{1}{2}\}\right] - \frac{n-1}{n} \cdot \frac{1}{n(n-1)} \sum_{1 \leq i \neq \tilde{i} \leq n} X_{ik}\{I(Y_{\tilde{i}} \leq Y_i) - F(Y_i)\} \\
&\quad - \frac{1}{n^2} \sum_{i=1}^{n} X_{ik}\{1 - F(Y_i)\},
\end{aligned}
$$

where $I(\cdot)$ is an indicator function. So we decompose $\nabla_k \mathcal{L}_n(\tilde{\boldsymbol{\beta}}_h)$ into three parts:

$$\nabla_k \mathcal{L}_n(\tilde{\boldsymbol{\beta}}_h) = \frac{1}{n} \sum_{i=1}^{n} X_{ik}\left[\mathbf{X}_i^\top \tilde{\boldsymbol{\beta}}_h - \{F(Y_i) - \frac{1}{2}\}\right] - \frac{n-1}{n} A_k - \frac{1}{n^2} \sum_{i=1}^{n} X_{ik}\{1 - F(Y_i)\},$$

where $A_k$ is a U-statistics with kernel $h_k(y_1, x_{1k}; y_2, x_{2k}) = \frac{1}{2}\left[x_{1k}\{I(y_2 \leq y_1) - F(y_1)\} + x_{2k}\{I(y_1 \leq y_2) - F(y_2)\}\right]$. Then

$$\Pr\left(|\nabla_k \mathcal{L}_n(\tilde{\boldsymbol{\beta}}_h)| \geq c\sqrt{\frac{\log p}{n}}\right) \leq \Pr\left(|A_k| \geq t\right) \tag{1.42}$$

$$+ \Pr\left(\left|\frac{1}{n} \sum_{i=1}^{n} X_{ik}\left[\mathbf{X}_i^\top \tilde{\boldsymbol{\beta}}_h - \{F(Y_i) - \frac{1}{2}\}\right]\right| \geq t\right) \tag{1.43}$$

$$+ \Pr\left(\left|\frac{1}{n^2} \sum_{i=1}^{n} X_{ik}\{1 - F(Y_i)\}\right| \geq t\right) \tag{1.44}$$

with $t = \frac{c}{3}\sqrt{\frac{\log p}{n}}$.

Firstly, for probability (1.42). For $\forall s \in \mathbb{R}$

$$\Pr(A_k \geq t) \overset{(1)}{\leq} \exp(-st)\mathbb{E}\{\exp(sA_k)\}$$

$$\overset{(2)}{\leq} \exp(-st)\mathbb{E}\left[\exp\left\{\frac{s}{N}\sum_{i=1}^{N} h_k(Y_i, X_{ik}; Y_{N+i}, X_{(N+i)k})\right\}\right]$$

$$\leq \exp(-st)\mathbb{E}\left[\exp\left\{\frac{s}{2N}\sum_{i=1}^{N}(X_{ik} + X_{(N+i)k})\right\}\right]$$

$$\leq \exp\left(-st + \frac{C_1 K^2}{2N}s^2\right),$$

where $C_1$ is an absolute constant, $K = \max_{1 \leq k \leq p}\|X_{1k}\|_{\psi_2}$ and $N = \lfloor\frac{n}{2}\rfloor$. Note that (1) holds by Markov inequality, (2) holds by Lemma A.6. Denote $s = \frac{Nt}{C_1 K^2}$, we have

$$\Pr(A_k \geq t) \leq \exp\left(-\frac{Nt^2}{2C_1 K^2}\right).$$

Similarly, we have

$$\Pr(|A_k| \geq t) \leq 2\exp\left(-\frac{Nt^2}{2C_1 K^2}\right). \tag{1.45}$$

Secondly, for probability (1.43). As both $\{\mathbf{X}_i^\top \tilde{\boldsymbol{\beta}}_h - \{F(Y_i) - \frac{1}{2}\}\}_{i=1}^n$ and $\{X_{ik}\}_{i=1}^n$ are *sub-Gaussian* random variables, $\{X_{ik}[\mathbf{X}_i^\top \tilde{\boldsymbol{\beta}}_h - \{F(Y_i) - \frac{1}{2}\}]\}_{i=1}^n$ is a zero mean *sub-Exponential* random variable sequence with *sub-Exponential* norm $\|X_{ik}[\mathbf{X}_i^\top \tilde{\boldsymbol{\beta}}_h - \{F(Y_i) - \frac{1}{2}\}]\|_{\psi_1} \leq KK'$, where $K' = \|\mathbf{X}_1^\top \tilde{\boldsymbol{\beta}}_h - \{F(Y_1) - \frac{1}{2}\}\|_{\psi_2}$. Then by inequality for r.v. with *sub-exponential* sum (Corollary 2.8.3 in [49]),

$$\Pr\left(\left|\frac{1}{n}\sum_{i=1}^{n} X_{ik}[\mathbf{X}_i^\top \tilde{\boldsymbol{\beta}}_h - \{F(Y_i) - \frac{1}{2}\}]\right| \geq t\right) \leq 2\exp\left\{-C_2\min\left(\frac{nt^2}{K^2 K'^2}, \frac{nt}{KK'}\right)\right\}, \tag{1.46}$$

where $C_2$ is an absolute constant.

Thirdly, for probability (1.44). Using general Hoeffding's inequality (Theorem 2.6.3 in [49]) directly,

$$\Pr\left(\left|\frac{1}{n^2}\sum_{i=1}^{n} X_{ik}\{1 - F(Y_i)\}\right| \geq t\right) \leq \Pr\left(\left|\frac{1}{n^2}\sum_{i=1}^{n} X_{ik}\right| \geq t\right) \leq 2\exp\left(-C_3\frac{n^3 t^2}{K^2}\right), \tag{1.47}$$

where $C_3$ is an absolute constant. As $\exp(-n^3) = o(\exp(-n))$, probability (1.44) is trivial compared to other two terms. Combining inequalities (1.45), (1.46) and (1.47), we have

$$\Pr\left(\|\nabla\mathcal{L}_n(\tilde{\boldsymbol{\beta}}_h)\|_\infty \geq c\sqrt{\frac{\log p}{n}}\right) = \Pr\left(\max_{1 \leq k \leq p}|\nabla_k\mathcal{L}_n(\tilde{\boldsymbol{\beta}}_h)| \geq c\sqrt{\frac{\log p}{n}}\right)$$

$$\leq p\max_{1 \leq k \leq p}\Pr\left(|\nabla_k\mathcal{L}_n(\tilde{\boldsymbol{\beta}}_h)| \geq c\sqrt{\frac{\log p}{n}}\right)$$

$$\leq 4p\exp(-C_4\log p),$$

where $C_4 = \min\{\frac{c^2}{36C_1 K^2}, \frac{C_2 c^2}{9K^2 K'^2}\}$. For a given constant $c$, we can choose arbitrary constants $c_1 \geq 4$, $c_2 \leq C_4 - 1$ and inequality (1.41) is satisfied. □

**Lemma A.8.** *Under the assumptions in theorem 3.1, then*

$$\boldsymbol{\beta}_h = \frac{Cov(F(Y), \boldsymbol{\beta}^\top \mathbf{X})}{\boldsymbol{\beta}^\top \Sigma \boldsymbol{\beta}}\boldsymbol{\beta}.$$

*Proof.* The proof of this lemma is similar to the proof of theorem 2.1 in [32]. Using Jensen's inequality and LC condition, we can get this result. We omit the proof. □

**Lemma A.9.** *Suppose $p_{\lambda_X}$ satisfies Assumption A.2. And under the Assumption A.1, we have*

$$\left\| \frac{1}{n} \sum_{i=1}^{n} \mathbf{Z}_{ij}(X_{ij} - \mathbf{Z}_{ij}^\top \hat{\boldsymbol{\theta}}_j) \right\|_\infty \leq L\lambda_X,$$

*where $\hat{\boldsymbol{\theta}}_j$ is defined in (2.5) and $L$ is a constant only depend on $p_{\lambda_X}$.*

*Proof.* By the property of $\hat{\boldsymbol{\theta}}_j$, we have

$$\frac{1}{n} \sum_{i=1}^{n} Z_{ijk}(X_{ij} - \mathbf{Z}_{ij}^\top \hat{\boldsymbol{\theta}}_j) = \nabla p_{\lambda_X}(\hat{\theta}_{jk})$$

with probability 1. Under assumption A.2 and by Lemma 4(a) in [35], all subgradients and derivatives of $p_{\lambda_X}$ are bounded in magnitude by $L\lambda_X$, that is,

$$\max_{1 \leq k \leq p} |\nabla p_{\lambda_X}(\hat{\theta}_{jk})| \leq L\lambda_X.$$

And this lemma is proved. □

**Lemma A.10.** *Under the Assumptions A.1 and A.2, for $\boldsymbol{\beta}_h \in \{\boldsymbol{\beta}_h : \|\boldsymbol{\beta}_h\|_\infty \lesssim \sqrt{\log p/n}\}$, there exist universal constants $(c, c_1, c_2)$ such that*

$$\Pr\left( \left\| \frac{1}{n} \sum_{i=1}^{n} \mathbf{Z}_{ij}\{F_n(Y_i) - 1/2 - \mathbf{Z}_{ij}^\top \boldsymbol{\gamma}_j\} \right\|_\infty \geq c\sqrt{\frac{\log p}{n}} \right) \leq c_1 \exp(-c_2 \log p).$$

*Proof.* At first we decompose $\frac{1}{n}\sum_{i=1}^n \mathbf{Z}_{ij}\{F_n(Y_i) - 1/2 - \mathbf{Z}_{ij}^\top \boldsymbol{\gamma}_j\}$ as

$$\frac{1}{n} \sum_{i=1}^{n} \mathbf{Z}_{ij}\{F_n(Y_i) - 1/2 - \mathbf{Z}_{ij}^\top \boldsymbol{\gamma}_j\} = \nabla \mathcal{L}_n(\boldsymbol{\beta}_h) + \frac{1}{n} \sum_{i=1}^{n} \beta_{hj} X_{ij} \mathbf{Z}_{ij}$$

$$= \nabla \mathcal{L}_n(\boldsymbol{\beta}_h) + \frac{1}{n} \sum_{i=1}^{n} \beta_{hj}\{X_{ij}\mathbf{Z}_{ij} - \mathbb{E}(X_j \mathbf{Z}_j)\} + \beta_{hj}\mathbb{E}(X_j \mathbf{Z}_j)$$

$$=: I_1 + I_2 + I_3.$$

If $\|\boldsymbol{\beta}_h\|_\infty = 0$, then $I_2$, $I_3 = 0$. Thus this Lemma is proved by Lemma A.7. So in the following proof, suppose $\|\boldsymbol{\beta}_h\|_\infty > 0$, that is, there exists $1 \leq j \leq p$, such that $|\boldsymbol{\beta}_{hj}| > 0$. As $\|\boldsymbol{\beta}_h\|_\infty \lesssim \sqrt{\log p/n}$, there exist a constant $c_3 > 0$ such that $\|\boldsymbol{\beta}_h\|_\infty \leq c_3\sqrt{\log p/n}$. Denote $c' = c/3$, $c'_1 = c_1/3$, we have

$$\Pr\left( \left\| \frac{1}{n} \sum_{i=1}^{n} \mathbf{Z}_{ij}\{F_n(Y_i) - 1/2 - \mathbf{Z}_{ij}^\top \boldsymbol{\gamma}_j\} \right\|_\infty \geq c\sqrt{\frac{\log p}{n}} \right) \leq \sum_{k=1}^{3} \Pr\left( \|I_k\|_\infty \geq c'\sqrt{\frac{\log p}{n}} \right).$$

So it suffices to prove $\Pr(\|I_k\|_\infty \geq c'\sqrt{\log p/n}) \leq c'_1 \exp(-c_2 \log p)$, $k = 1, 2, 3$.

For $k = 1$, this inequality holds by lemma A.7 and choosing proper constants.

For $k = 2$, by sub-Gaussian Assumption in Assumption A.1 and applying Bernstein inequality(Theorem 2.8.2 in [49]), we derive

$$\Pr\left( \|I_2\|_\infty \geq c'\sqrt{\frac{\log p}{n}} \right)$$

$$\leq p \max_{1 \leq k \leq p} \Pr\left( \left| \frac{\beta_{hj}}{n} \sum_{i=1}^{n} \{X_{ij}Z_{ijk} - \mathbb{E}(X_j Z_{jk})\} \right| \geq c'\sqrt{\frac{\log p}{n}} \right)$$

$$\leq 2p \max_{1 \leq k \leq p} \exp\left\{ -c_4 \min\left( \frac{c'^2 \log p}{\beta_{hj}^2 K^2 n}, \frac{C\sqrt{\log p}}{|\beta_{hj}|K\sqrt{n}} \right) n \right\}$$

$$\leq 2p \exp\left[ -c_4 \min\left\{ \left( \frac{c'}{c_3 K} \right)^2, \frac{c'}{c_3 K} \right\} n \right],$$

where $c_4$ is a constant independent of $\boldsymbol{\beta}_h$ and $K = \max_{1 \le k \le p} \|X_j Z_{jk}\|_{\psi_1}$. Thus this inequality holds as $n \gg \log p$.

For $k = 3$. As $X_j, j = 1, \ldots, p$ are all sub-Gaussian with uniform bounded sub-Gaussian norm, then it follows that $\|I_3\|_\infty \le Cc_3\sqrt{\log p/n}$. In summary, this Lemma is proved. $\qquad \square$

**Lemma A.11.** *Let $X_1, \ldots, X_n$ be i.i.d mean zero random variables. If there exist constants $L_1$ and $L_2$, such that $\Pr(|X_i| \ge x) \le L_1 \exp(-L_2 x^r)$ for some $r > 0$, then for $x \ge \sqrt{8\mathbb{E}(X_i^2)/n}$*

$$\Pr\left(\left|\frac{1}{n}\sum_{i=1}^{n}X_i\right| \ge x\right) \le 4\exp\left\{-\frac{1}{8}n^{r/(2+r)}x^{2r/(2+r)}\right\} + 4nL_1\exp\left\{-\frac{L_2 n^{r/(2+r)}x^{2r/(2+r)}}{2^r}\right\}.$$

*Proof.* Details of the proof can be seen in [40]. $\qquad \square$

**Lemma A.12.** *For a sequence $\eta(n,p) \to \infty$ as $(n,p) \to \infty$, we have*

$$\Pr\left(\sup_{t \in \mathbb{R}}\left|F_n(t) - F(t)\right| > \frac{\eta(n,p)}{\sqrt{n}}\right) = o(1).$$

*Proof.* We can get this result by using DKW inequality([48]) directly. $\qquad \square$

**Lemma A.13.** *Under the Assumption A.1 and A.2, $\max_{i,j}(X_{ij} - \mathbf{Z}_{ij}^\top\boldsymbol{\theta}_j)^2$, $\max_{i,j}e_{i,j}^2$ and $\max_{i,j}|(X_{ij} - \mathbf{Z}_{ij}^\top\boldsymbol{\theta}_j)e_{i,j}|$ are all $O_p(\log(np))$. Both $\max_j \frac{1}{n}\sum_{i=1}^{n}\{\mathbf{Z}_{ij}^\top(\boldsymbol{\theta}_j - \hat{\boldsymbol{\theta}}_j)\}^2$ and $\max_j \frac{1}{n}\sum_{i=1}^{n}(\hat{e}_{ij} - e_{ij})^2$ are $O_p(s\frac{\log p}{n})$. We abbreviate $1 \le i \le n$, $j \in \mathcal{G}$ as $i, j$ respectively.*

*Proof.* For a sufficient large constant $C$, by sub-Gaussian assumption we have

$$\Pr\left(\max_{i,j}(X_{ij} - \mathbf{Z}_{ij}^\top\boldsymbol{\theta}_j)^2 > C^2 \log(np)\right)$$
$$=\Pr\left(\max_{i,j}|X_{ij} - \mathbf{Z}_{ij}^\top\boldsymbol{\theta}_j| > C\sqrt{\log(np)}\right)$$
$$\le np\max_{i,j}\Pr\left(|X_{ij} - \mathbf{Z}_{ij}^\top\boldsymbol{\theta}_j| > C\sqrt{\log(np)}\right)$$
$$\le 2np\exp\left\{-C^2 \log(np)/K_1^2\right\} = o(1),$$

where $K_1$ is a constant independent of the choice of $i, j$. So $\max_{i,j}(X_{ij} - \mathbf{Z}_{ij}^\top\boldsymbol{\theta}_j)^2 = O_p(\log(np))$ is proved. By same argument, we can prove both $\max_{i,j}e_{i,j}^2$ and $\max_{i,j}|(X_{ij} - \mathbf{Z}_{ij}^\top\boldsymbol{\theta}_j)e_{i,j}|$ are $O_p(\log(np))$. Applying Theorem 2 in [35], it follows directly that $\max_j \frac{1}{n}\sum_{i=1}^{n}[\mathbf{Z}_{ij}^\top(\boldsymbol{\theta}_j - \hat{\boldsymbol{\theta}}_j)]^2 = O_p(s\frac{\log p}{n})$. But for $\max_j \frac{1}{n}\sum_{i=1}^{n}(\hat{e}_{ij} - e_{ij})^2$, there is something different. Note that for a sufficient large constant $C > 0$,

$$\Pr\left(\max_j \frac{1}{n}\sum_{i=1}^{n}(\hat{e}_{ij} - e_{ij})^2 > 4C\frac{s\log p}{n}\right)$$
$$=\Pr\left(\max_j \frac{1}{n}\sum_{i=1}^{n}\{F_n(Y_i) - F(Y_i) - \mathbf{Z}_{ij}^\top(\hat{\boldsymbol{\gamma}}_j - \boldsymbol{\gamma}_j)\}^2 > 4C\frac{s\log p}{n}\right)$$
$$\le\Pr\left(\max_j \frac{1}{n}\sum_{i=1}^{n}\{F_n(Y_i) - F(Y_i)\}^2 + \{\mathbf{Z}_{ij}^\top(\hat{\boldsymbol{\gamma}}_j - \boldsymbol{\gamma}_j)\}^2 > 2C\frac{s\log p}{n}\right)$$
$$\le\Pr\left(\sup_{t \in \mathbb{R}}|F_n(t) - F(t)| > \sqrt{\frac{\log p}{n}}\right) + \Pr\left(\max_j \frac{1}{n}\sum_{i=1}^{n}\{\mathbf{Z}_{ij}^\top(\hat{\boldsymbol{\gamma}}_j - \boldsymbol{\gamma}_j)\}^2 > C\frac{s\log p}{n}\right).$$

The first term of last inequality is $o(1)$ by lemma A.12 with $\eta(n,p) = \sqrt{\log p}$. The second term is also $o(1)$ by applying Theorem 2 in [35] again. In summary, this lemma is proved. $\qquad \square$

**Lemma A.14.** *Under the Assumptions A.1 and A.2,*

$$\max_{j \in \mathcal{G}} |\sigma_j^2 - \hat{\sigma}_j^2| = o_p\big(r_2(n, p, p_0, s)\big),$$

*where*

$$r_2(n, p, p_0, s) = \sqrt{\frac{(\log np_0)^5}{n}} \vee \sqrt{\frac{s \log p \log(np)}{n}} \vee \sqrt{\frac{(s \log p)^{3/2}\sqrt{\log np}}{n}} \vee \sqrt{\frac{(s \log p)^2}{n}}. \tag{1.48}$$

*Proof.* To prove this lemma, we define

$$\tilde{\sigma}_j^2 = \frac{1}{n} \sum_{i=1}^n \big\{ (X_{ij} - \mathbf{Z}_{ij}^\top \boldsymbol{\theta}_j) e_{ij} + m_j(Y_i) \big\}^2,$$

where $e_{ij} = F(Y_i) - 1/2 - \mathbf{Z}_{ij}^\top \boldsymbol{\gamma}_j$, $m_j(Y_i) = \mathbb{E}_{\boldsymbol{\beta}_h}[(X_j - \mathbf{Z}_j^\top \boldsymbol{\theta}_j)\{I(Y \geq Y_i) - F(Y)\}]$ and $\tilde{\sigma}_j^2$ is an unbiased estimator of $\sigma_j^2$. It follows by the triangle inequality that $\max_{j \in \mathcal{G}} |\sigma_j^2 - \hat{\sigma}_j^2| \leq \max_{j \in \mathcal{G}} |\sigma_j^2 - \tilde{\sigma}_j^2| + \max_{j \in \mathcal{G}} |\tilde{\sigma}_j^2 - \hat{\sigma}_j^2|$. So it suffices to prove both terms are $o_p\big((\log p_0)^{-1}\big)$.

We begin with estimation of the rate of $\max_{j \in \mathcal{G}} |\sigma_j^2 - \tilde{\sigma}_j^2|$. Note that

$$\max_{j \in \mathcal{G}} |\sigma_j^2 - \tilde{\sigma}_j^2| = \max_{j \in \mathcal{G}} \frac{1}{n} \sum_{i=1}^n \Big[ \big\{ (X_{ij} - \mathbf{Z}_{ij}^\top \boldsymbol{\theta}_j) e_{ij} + m_j(Y_i) \big\}^2 - \sigma_j^2 \Big].$$

Since $\Pr_{\boldsymbol{\beta}_h}(\max_{j \in \mathcal{G}} |\sigma_j^2 - \tilde{\sigma}_j^2| \geq t) \leq p_0 \max_{j \in \mathcal{G}} \Pr_{\boldsymbol{\beta}_h}(|\sigma_j^2 - \tilde{\sigma}_j^2| \geq t)$ and for $\forall t \geq \sigma_j^2$,

$$\Pr_{\boldsymbol{\beta}_h}\left( \left| \big\{ (X_{ij} - \mathbf{Z}_{ij}^\top \boldsymbol{\theta}_j) e_{ij} + m_j(Y_i) \big\}^2 - \sigma_j^2 \right| \geq t \right)$$

$$\leq \Pr_{\boldsymbol{\beta}_h}\left( \left| \big\{ (X_{ij} - \mathbf{Z}_{ij}^\top \boldsymbol{\theta}_j) e_{ij} + m_j(Y_i) \big\}^2 \right| \geq t - \sigma_j^2 \right)$$

$$= \Pr_{\boldsymbol{\beta}_h}\left( \left| (X_{ij} - \mathbf{Z}_{ij}^\top \boldsymbol{\theta}_j) e_{ij} + m_j(Y_i) \right| \geq \sqrt{t - \sigma_j^2} \right).$$

By sub-Gaussian assumption, $(X_{ij} - \mathbf{Z}_{ij}^\top \boldsymbol{\theta}_j) e_{ij} + m_j(Y_i)$ is a *sub-Exponential* random variable with uniform bounded *sub-Exponential* norm. When $t$ is large enough, by the definition of *sub-Exponential* random variable, we could find some constants $L_1, L_2$ such that $\Pr_{\boldsymbol{\beta}_h}\big( |(X_{ij} - \mathbf{Z}_{ij}^\top \boldsymbol{\theta}_j) e_{ij} + m_j(Y_i)| \geq \sqrt{t - \sigma_j^2} \big) \leq L_1 \exp(-L_2 \sqrt{t})$. Thus it follows by Lemma A.11 for some $t \geq C\sqrt{(\log np_0)^5/n}$ with sufficient large constant $C$,

$$p_0 \max_{j \in \mathcal{G}} \Pr_{\boldsymbol{\beta}_h}\big( |\sigma_j^2 - \tilde{\sigma}_j^2| \geq t \big) \leq 4p_0 \exp(-n^{1/5} t^{2/5}/8) + 4np_0 C_1 \exp(-C_2 n^{1/5} t^{2/5}) = O_p\big((np_0)^{-1}\big)$$

for some constants $C_1$ and $C_2$. Thus, $\max_{j \in \mathcal{G}} |\sigma_j^2 - \tilde{\sigma}_j^2| = O_p\big(\sqrt{(\log np_0)^5/n}\big)$.

Next we estimate the rate of $\max_{j\in\mathcal{G}}\left|\tilde{\sigma}_j^2 - \hat{\sigma}_j^2\right|$. By definitions,

$$
\begin{aligned}
&\max_{j\in\mathcal{G}}\left|\tilde{\sigma}_j^2 - \hat{\sigma}_j^2\right| \\
&= \max_{j\in\mathcal{G}}\left|\frac{1}{n}\sum_{i=1}^n\left\{(X_{ij}-\mathbf{Z}_{ij}^\top\hat{\boldsymbol{\theta}}_j)\hat{e}_{ij}+\hat{m}_j(Y_i)\right\}^2 - \left\{(X_{ij}-\mathbf{Z}_{ij}^\top\boldsymbol{\theta}_j)e_{ij}+m_j(Y_i)\right\}^2\right| \\
&\leq \max_{j\in\mathcal{G}}\left|\frac{1}{n}\sum_{i=1}^n\left\{(X_{ij}-\mathbf{Z}_{ij}^\top\hat{\boldsymbol{\theta}}_j)\hat{e}_{ij}+\hat{m}_j(Y_i)-(X_{ij}-\mathbf{Z}_{ij}^\top\boldsymbol{\theta}_j)e_{ij}-m_j(Y_i)\right\}^2\right| \\
&\quad + \max_{j\in\mathcal{G}}\left|\frac{2}{n}\sum_{i=1}^n\left\{(X_{ij}-\mathbf{Z}_{ij}^\top\boldsymbol{\theta}_j)e_{ij}+m_j(Y_i)\right\}\left\{(X_{ij}-\mathbf{Z}_{ij}^\top\hat{\boldsymbol{\theta}}_j)\hat{e}_{ij}+\hat{m}_j(Y_i)-(X_{ij}-\mathbf{Z}_{ij}^\top\boldsymbol{\theta}_j)e_{ij}-m_j(Y_i)\right\}\right| \\
&\leq \max_{j\in\mathcal{G}}\left|\frac{2}{n}\sum_{i=1}^n\left\{(X_{ij}-\mathbf{Z}_{ij}^\top\hat{\boldsymbol{\theta}}_j)\hat{e}_{ij}-(X_{ij}-\mathbf{Z}_{ij}^\top\boldsymbol{\theta}_j)e_{ij}\right\}^2\right| + \max_{j\in\mathcal{G}}\left|\frac{2}{n}\sum_{i=1}^n\left\{\hat{m}_j(Y_i)-m_j(Y_i)\right\}^2\right| \\
&\quad + \max_{j\in\mathcal{G}}\left|\frac{2}{n}\sum_{i=1}^n\left\{(X_{ij}-\mathbf{Z}_{ij}^\top\boldsymbol{\theta}_j)e_{ij}+m_j(Y_i)\right\}\left\{(X_{ij}-\mathbf{Z}_{ij}^\top\hat{\boldsymbol{\theta}}_j)\hat{e}_{ij}+\hat{m}_j(Y_i)-(X_{ij}-\mathbf{Z}_{ij}^\top\boldsymbol{\theta}_j)e_{ij}-m_j(Y_i)\right\}\right| \\
&=: 2(I_1+I_2+I_3).
\end{aligned}
$$

So it suffices to estimate the converge rate of $I_i$, $i=1,2,3$ respectively. Decomposing $I_1$, it follows that

$$
I_1 \leq \sum_{\nu=1}^6 I_{1\nu},
$$

where $I_{1\nu}$ is given by

$$
\begin{aligned}
I_{11} &= \max_{j\in\mathcal{G}}\frac{1}{n}\sum_{i=1}^n(X_{ij}-\mathbf{Z}_{ij}^\top\boldsymbol{\theta}_j)^2(\hat{e}_{ij}-e_{ij})^2, \\
I_{12} &= \max_{j\in\mathcal{G}}\frac{1}{n}\sum_{i=1}^n\left\{\mathbf{Z}_{ij}^\top(\boldsymbol{\theta}_j-\hat{\boldsymbol{\theta}}_j)\right\}^2 e_{ij}^2, \\
I_{13} &= \max_{j\in\mathcal{G}}\frac{1}{n}\sum_{i=1}^n\left\{\mathbf{Z}_{ij}^\top(\boldsymbol{\theta}_j-\hat{\boldsymbol{\theta}}_j)\right\}^2(\hat{e}_{ij}-e_{ij})^2, \\
I_{14} &= \max_{j\in\mathcal{G}}\left|\frac{2}{n}\sum_{i=1}^n(X_{ij}-\mathbf{Z}_{ij}^\top\boldsymbol{\theta}_j)e_{ij}\mathbf{Z}_{ij}^\top(\boldsymbol{\theta}_j-\hat{\boldsymbol{\theta}}_j)(\hat{e}_{ij}-e_{ij})\right|, \\
I_{15} &= \max_{j\in\mathcal{G}}\left|\frac{2}{n}\sum_{i=1}^n(X_{ij}-\mathbf{Z}_{ij}^\top\boldsymbol{\theta}_j)\mathbf{Z}_{ij}^\top(\boldsymbol{\theta}_j-\hat{\boldsymbol{\theta}}_j)(\hat{e}_{ij}-e_{ij})^2\right|, \\
I_{16} &= \max_{j\in\mathcal{G}}\left|\frac{2}{n}\sum_{i=1}^n\left\{\mathbf{Z}_{ij}^\top(\boldsymbol{\theta}_j-\hat{\boldsymbol{\theta}}_j)\right\}^2 e_{ij}(\hat{e}_{ij}-e_{ij})\right|.
\end{aligned}
$$

Then we deal with $I_{1\nu}$, $\nu=1,\ldots,6$ respectively. For simplify, in the rest of proof, we abbreviate $1\leq i\leq n$, $j\in\mathcal{G}$ as $i,j$ respectively.

For $I_{11}$, we have $I_{11} \leq \max_{i,j}(X_{ij}-\mathbf{Z}_{ij}^\top\boldsymbol{\theta}_j)^2\max_j\frac{1}{n}\sum_{i=1}^n(\hat{e}_{ij}-e_{ij})^2 = O_p(\log(np))O_p(n^{-1}s\log p) = O_p(n^{-1}s\log p\log(np))$ by lemma A.13. Similarly, we have $I_{12} \leq \max_{i,j}e_{ij}^2\max_j\frac{1}{n}\sum_{i=1}^n\left\{\mathbf{Z}_{ij}^\top(\boldsymbol{\theta}_j-\hat{\boldsymbol{\theta}}_j)\right\}^2 = O_p(n^{-1}s\log p\log(np))$.

For $I_{13}$, by Cauchy-Schwartz inequality and lemma A.13, it holds that

$$I_{13} \leq \frac{1}{n}\sqrt{\max_j \sum_{i=1}^n \{\mathbf{Z}_{ij}^\top(\boldsymbol{\theta}_j - \hat{\boldsymbol{\theta}}_j)\}^4}\sqrt{\max_j \sum_{i=1}^n (\hat{e}_{ij} - e_{ij})^4}$$

$$\leq n\max_j \frac{1}{n}\sum_{i=1}^n \{\mathbf{Z}_{ij}^\top(\boldsymbol{\theta}_j - \hat{\boldsymbol{\theta}}_j)\}^2 \max_j \frac{1}{n}\sum_{i=1}^n (\hat{e}_{ij} - e_{ij})^2$$

$$= O_p\left(n\left(s\frac{\log p}{n}\right)^2\right) = O_p\left(\frac{(s\log p)^2}{n}\right).$$

Similarly, we have

$$I_{14} \leq 2\max_{i,j}\left|(X_{ij} - \mathbf{Z}_{ij}^\top\boldsymbol{\theta}_j)e_{ij}\right|\sqrt{\max_j \frac{1}{n}\sum_{i=1}^n \{\mathbf{Z}_{ij}^\top(\boldsymbol{\theta}_j - \hat{\boldsymbol{\theta}}_j)\}^2}\sqrt{\max_j \frac{1}{n}\sum_{i=1}^n (\hat{e}_{ij} - e_{ij})^2}$$

$$= O_p(n^{-1}s\log p\log(np)),$$

$$I_{15} \leq 2\sqrt{n}\max_{i,j}\left|X_{ij} - \mathbf{Z}_{ij}^\top\boldsymbol{\theta}_j\right|\sqrt{\max_j \frac{1}{n}\sum_{i=1}^n \{\mathbf{Z}_{ij}^\top(\boldsymbol{\theta}_j - \hat{\boldsymbol{\theta}}_j)\}^2 \max_j \frac{1}{n}\sum_{i=1}^n (\hat{e}_{ij} - e_{ij})^2}$$

$$= O_p\left(\frac{(s\log p)^{3/2}\sqrt{\log(np)}}{n}\right),$$

$$I_{16} \leq 2\sqrt{n}\max_{i,j}\left|e_{ij}\right|\max_j \frac{1}{n}\sum_{i=1}^n \{\mathbf{Z}_{ij}^\top(\boldsymbol{\theta}_j - \hat{\boldsymbol{\theta}}_j)\}^2\sqrt{\max_j \frac{1}{n}\sum_{i=1}^n (\hat{e}_{ij} - e_{ij})^2}$$

$$= O_p\left(\frac{(s\log p)^{3/2}\sqrt{\log(np)}}{n}\right).$$

Next we deal with $I_2$. By definition and Hoeffding's inequality we have

$$I_2 = \max_j \frac{1}{n}\sum_{i=1}^n \left[\frac{1}{n}\sum_{\tilde{i}=1}^n (X_{\tilde{i}j} - \mathbf{Z}_{\tilde{i}j}^\top\hat{\boldsymbol{\theta}}_j)\{I(Y_{\tilde{i}} \geq Y_i) - F_n(Y_i)\} - m_j(Y_i)\right]^2$$

$$\leq \max_j \frac{1}{n}\sum_{i=1}^n \left[\frac{1}{n}\sum_{\tilde{i}=1}^n (X_{\tilde{i}j} - \mathbf{Z}_{\tilde{i}j}^\top\hat{\boldsymbol{\theta}}_j)\{I(Y_{\tilde{i}} \geq Y_i) - F_n(Y_i)\}\right]^2$$

$$\leq \max_j \left\{\frac{1}{n}\sum_{\tilde{i}=1}^n (X_{\tilde{i}j} - \mathbf{Z}_{\tilde{i}j}^\top\hat{\boldsymbol{\theta}}_j)\right\}^2$$

$$= O_p(n^{-1}\log p).$$

In summary, we have that $I_1 + I_2 = O_p(r(n,p,s))$, where

$$r(n,p,s) = \frac{s\log p\log(np)}{n} \vee \frac{(s\log p)^{3/2}\sqrt{\log np}}{n} \vee \frac{(s\log p)^2}{n} \vee \frac{\log p}{n}.$$

Similarly, we can show that $I_3 = O_p(\sqrt{r(n,p,s)})$, it follows from Cauchy-Schwartz inequality that

$$I_3 \leq \sqrt{\max_j \tilde{\sigma}_j^2}\sqrt{2(I_1 + I_2)}.$$

Thus we have $\max_{j\in\mathcal{G}}\left|\tilde{\sigma}_j^2 - \hat{\sigma}_j^2\right| = O_p(\sqrt{r(n,p,s)})$ and totally,

$$\max_{j\in\mathcal{G}}\left|\sigma_j^2 - \hat{\sigma}_j^2\right| = O_p\left(\sqrt{\frac{(\log np_0)^5}{n}} \vee \sqrt{r(n,p,s)}\right)$$

$$= O_p\left(\sqrt{\frac{(\log np_0)^5}{n}} \vee \sqrt{\frac{s\log p\log(np)}{n}} \vee \sqrt{\frac{(s\log p)^{3/2}\sqrt{\log np}}{n}} \vee \sqrt{\frac{(s\log p)^2}{n}}\right)$$

$$= O_p\left(r_2(n,p,p_0,s)\right).$$

Thus this lemma is proved. $\qquad\square$

**Lemma A.15.** *Under the Assumptions A.1,*

$$\max_{j\in\mathcal{G}}\left|S_{nj} - \sqrt{n}\beta_{hj}\delta_j\right| = O_p\big((\log p_0)^{1/2}\big).$$

*Proof.* For simplify, we rewrite $S_{nj} - \sqrt{n}\beta_{hj}\delta_j = \sum_{i=1}^n \xi_{ij}/\sqrt{n}$. Under assumption A.1, it's easy to show that $\{\xi_{ij}\}_{i=1}^n$ is an i.i.d. zero mean $sub\text{-}Exponential$ r.v. sequence. So we can prove this lemma by using Bernstein's inequality and Bonferroni's inequality. $\square$

**Lemma A.16.** *Under the Assumptions A.1 and A.2,*

$$\max_{j\in\mathcal{G}}|(\widetilde{T}_{nj} - \hat{\sigma}_j^{-1}\sqrt{n}\beta_{hj}\delta_j) - (\widetilde{S}_{nj} - \sigma_j^{-1}\sqrt{n}\beta_{hj}\delta_j)| = O_p\big(r(n,p,p_0,s)\big)$$

*uniformly for $\boldsymbol{\beta}_h \in \Omega \cap \{\boldsymbol{\beta}_h : \|\boldsymbol{\beta}_h\|_\infty \lesssim \sqrt{\log p/n}\}$ with*

$$r(n,p,p_0,s) = r_1(n,p,p_0,s) \vee r_2(n,p,p_0,s)(\log p_0)^{1/2}. \tag{1.49}$$

*Where*

$$r_1(n,p,p_0,s) = \frac{s\log p}{\sqrt{n}} \vee \frac{(\log p_0)^{3/2}\log(np_0)}{n} \vee \frac{\log p_0}{\sqrt{n}},$$

*and $r_2(n,p,p_0,s)$ is defined in (1.48).*

*Proof.* First we decompose $(\widetilde{T}_{nj} - \hat{\sigma}_j^{-1}\sqrt{n}\beta_{hj}\delta_j) - (\widetilde{S}_{nj} - \sigma_j^{-1}\sqrt{n}\beta_{hj}\delta_j)$ as:

$$(\widetilde{T}_{nj} - \hat{\sigma}_j^{-1}\sqrt{n}\beta_{hj}\delta_j) - (\widetilde{S}_{nj} - \sigma_j^{-1}\sqrt{n}\beta_{hj}\delta_j)$$
$$=\hat{\sigma}_j^{-1}(T_{nj} - S_{nj}) + (\hat{\sigma}_j^{-1} - \sigma_j^{-1})(S_{nj} - \sqrt{n}\beta_{hj}\delta_j)$$
$$=: A_{1j} + A_{2j}.$$

Next we deal with $\max_{j\in\mathcal{G}}|A_{ij}|, i = 1,2$ respectively. We handle $\max_{j\in\mathcal{G}}|A_{1j}|$ at first. We decompose $A_{1j}$ as

$$A_{1j} = \hat{\sigma}_j^{-1}\left[\frac{1}{n^{3/2}}\sum_{i=1}^n(\frac{1}{2} - \mathbf{Z}_{ij}^\top\boldsymbol{\gamma}_j)(X_{ij} - \mathbf{Z}_{ij}^\top\boldsymbol{\theta}_j) + \frac{(\boldsymbol{\theta}_j - \hat{\boldsymbol{\theta}}_j)^\top}{\sqrt{n}}\sum_{i=1}^n \mathbf{Z}_{ij}\{F_n(Y_i) - 1/2 - \mathbf{Z}_{ij}^\top\boldsymbol{\gamma}_j\}\right.$$

$$\left. + \frac{(\boldsymbol{\gamma}_j - \hat{\boldsymbol{\gamma}}_j)^\top}{\sqrt{n}}\sum_{i=1}^n \mathbf{Z}_{ij}(X_{ij} - \mathbf{Z}_{ij}^\top\hat{\boldsymbol{\theta}}_j) - n^{-1/2}U_n + (\sqrt{n}U_{nj} - S_{nj})\right]$$

$$=: \hat{\sigma}_j^{-1}\big(I_{1j} + I_{2j} + I_{3j} + I_{4j} + I_{5j}\big)$$

where $U_{nj}$ is defined in (1.24). So it suffices to prove $\max_{j\in\mathcal{G}}|I_{ij}| = O_p(r_1(n,p,p_0,s)), i = 1,2,3,4,5$ respectively. For a sufficient small constant $c' > 0$ and a sufficient large constant $C > 0$, we have following results.

Step 1, for $i = 1$. $\max_{j\in\mathcal{G}}|I_{ij}| = O_p(r_1(n,p,p_0,s))$ holds by sub-Gaussian Assumption and Bernstein inequality. The proof is similar to step 2 in the proof of Lemma A.10, we omit it.

Step 2, for $i = 2$. By Proposition A.1 and Lemma A.10, we have

$$\Pr(|I_{2j}| \geq c\sqrt{n}\lambda_X\lambda_Y s_X) \leq c_1\exp(-c_2 p),$$

where $(c, c_1, c_2)$ are constants defined in these lemma and independent of the choice of $j$. We can derive that $\sqrt{n}\lambda_X\lambda_Y s_X = O(s\log p/\sqrt{n})$ by the fact that $\lambda_X,\ \lambda_Y \asymp \sqrt{\log p/n}$

Step 3, for $i = 3$. It's similar to step 2 by using Theorem 3.1 and Lemma A.9. We omit it.

Step 4, for $i = 4$. Note that $I_{4j} = n^{-1/2}(U_{nj} - \beta_{hj}\delta_j) + n^{-1/2}\beta_{hj}\delta_j$ and $n^{-1/2}\max_{j\in\mathcal{G}}|\beta_{hj}\delta_j| = O(n^{-1}\sqrt{\log p})$. It suffices to deal with $n^{-1/2}(U_{nj} - \beta_{hj}\delta_j)$. We begin the proof in the same way as we argue in the proof of lemma A.7. Applying lemma A.6, for $\forall s \in \mathbb{R}$ we have

$$\Pr(U_{nj} - \beta_{hj}\delta_j \geq t) \leq \exp(-st)\mathbb{E}\left[\exp\left\{\frac{s}{N}\sum_{i=1}^N h_j(Y_i, \mathbf{X}_i; Y_{N+i}, \mathbf{X}_{N+i}) - \beta_{hj}\delta_j\right\}\right], \tag{1.50}$$

where $N = \lfloor n/2 \rfloor$ and $t \in \mathbb{R}^+$. Denote $W_{ij} = h_j(Y_i, \mathbf{X}_i; Y_{N+i}, \mathbf{X}_{N+i}) - \beta_{hj}\delta_j$, $i = 1, \ldots, N$. By the *sub-Gaussian* assumption made in assumption A.1, $\{W_{ij}\}_{i=1}^N$ is an i.i.d. *sub-Exponential* sequence and $\max_{1 \le j \le p} \|W_{\cdot j}\|_{\psi_1} < \infty$. Thus by the same way used in the proof of Bernstein inequality([49]), we derive

$$\max_{j \in \mathcal{G}} \Pr(|U_{nj} - \beta_{hj}\delta_j| \ge t) \le 2\exp\left[-c\min\left\{\frac{t^2}{(\max_{1 \le j \le p}\|W_{\cdot j}\|_{\psi_1})^2}, \frac{t}{\max_{1 \le j \le p}\|W_{\cdot j}\|_{\psi_1}}\right\}n\right],$$

where $c$ is a constant. Denote $t = Cr_1(n, p, p_0, s)$, we prove it as $n \gg t^{-2}$.

Step 5, for $i = 5$. By the theory of $U$-statistics, $U_{nj} - S_{nj}/\sqrt{n}$ is a two-order canonical $U$-statistic. Thus $\max_{j \in \mathcal{G}} |I_{5j}| = O_p\left(\{n^{-1}(\log p_0)^{3/2}\log(np_0) + n^{-1/2}\log p_0\}\right)$ by Lemma A.21. In summary, we derive that $\max_{j \in \mathcal{G}} |A_{1j}| = O_p(r_1(n, p, p_0, s))$ since $\hat\sigma_j \ge C_{\min} - \psi_n \ge C_{\min}/2 > 0$ for $n$ large enough.

Next we prove $\max_{j \in \mathcal{G}} |A_{2j}| = o_p(r_2(n, p, p_0, s)(\log p_0)^{1/2})$. For $\forall c' > 0$ we have

$$\Pr_{\boldsymbol\beta_h}\left(\max_{j \in \mathcal{G}} |A_{2j}| \ge \frac{4}{3}C'^{-3}c'Cr_2(n, p, p_0, s)(\log p_0)^{1/2}\right)$$

$$\le \Pr_{\boldsymbol\beta_h}\left(\max_{j \in \mathcal{G}}|\sigma_j^2 - \hat\sigma_j^2| \ge c'r_2(n, p, p_0, s)\right) + \Pr_{\boldsymbol\beta_h}\left(\max_{j \in \mathcal{G}}|S_{nj} - \sqrt{n}\beta_{hj}\delta_j| \ge C(\log p_0)^{1/2}\right)$$

$$= o(1).$$

The last equality holds by Lemma A.14 and A.15. Note that all the universal constants do not depend on $n$, $p$, $\boldsymbol\beta_h$ and $\boldsymbol\theta_j$, $j = 1, \ldots, p$. Thus this lemma is proved. $\qquad\square$

**Lemma A.17.** *Under the Assumptions A.1 and A.2,*

$$\max_{j \in \mathcal{G}} |\widetilde{T}_{nj} - \sqrt{n}\widetilde{U}_{nj}| = O_p(r(n, p, p_0, s)).$$

*Proof.* The proof of this Lemma is similar to Lemma A.16. $\qquad\square$

**Lemma A.18.** *Under the Assumptions in theorem A.1, if $s = o\left(\sqrt{n}/(\log p(\log np_0)^{3/2})\right)$, then*

$$\Pr_w\left(\left|\sqrt{M_{n,\mathcal{G}}^\sharp} - \sqrt{L_{n,\mathcal{G}}^\sharp}\right| \ge C(\log(np_0))^{-1/2}\right) \to 0 \text{ in probability},$$

*where $L_{n,\mathcal{G}}^\sharp$ is defined in* (1.27) *in the main text and $C$ is a sufficient large constant.*

*Proof.* By definition,

$$\left|\sqrt{M_{n,\mathcal{G}}^\sharp} - \sqrt{L_{n,\mathcal{G}}^\sharp}\right| \le \max_{j \in \mathcal{G}}\left|\frac{2}{n}(\widetilde{T}_{nj} - \sqrt{n}\widetilde{U}_{nj})\sum_{i=1}^n w_i\right|$$

$$+ \max_{j \in \mathcal{G}}\left|\frac{2}{\sqrt{n}(n-1)}\sum_{i=1}^n\sum_{\tilde i \ne i}\left\{\hat\sigma_j^{-1}\hat h_j(Y_i, \mathbf{X}_i; Y_{\tilde i}, \mathbf{X}_{\tilde i}) - \sigma_j^{-1}h_j(Y_i, \mathbf{X}_i; Y_{\tilde i}, \mathbf{X}_{\tilde i})\right\}w_i\right|$$

$$=: I_1 + I_2.$$

Thus it suffices to prove $\Pr_w(I_i \ge C(\log p_0)^{-1/2}) \to 0$ in probability, $i = 1, 2$.

For $I_1$, since $w_i$ is standard Gaussian r.v., the Hoeffding inequality implies that

$$\Pr_w(I_1 \ge C(\log(np_0))^{-1/2}) \le 2\exp\left\{-\frac{C'n(\log(np_0))^{-1}}{\left(\max_{j \in \mathcal{G}}|\widetilde{T}_{nj} - \sqrt{n}\widetilde{U}_{nj}|\right)^2}\right\}$$

where $C'$ is a constant. Applying Lemma A.17, $\Pr_w(I_1 \ge C(\log(np_0))^{-1/2}) = o_p(1)$ is proved.

For $I_2$, the Hoeffding inequality also implies that

$$\Pr_w(I_2 \ge C(\log(np_0))^{-1/2}) \le 2\exp\left\{-\frac{C'n(n-1)^2(\log(np_0))^{-1}}{\max_{j \in \mathcal{G}}\sum_{i=1}^n\left(\sum_{\tilde i \ne i}\{\hat\sigma_j^{-1}\hat h_j - \sigma_j^{-1}h_j\}\right)^2}\right\}.$$

For simplify, we abbreviate $\hat{h}_j(Y_i, \mathbf{X}_i; Y_{\tilde{i}}, \mathbf{X}_{\tilde{i}})$, $h_j(Y_i, \mathbf{X}_i; Y_{\tilde{i}}, \mathbf{X}_{\tilde{i}})$ as $\hat{h}_j$, $h_j$ respectively. In the rest of proof, we abbreviate $\max_{1 \le i \le n}$, $\max_{\tilde{i} \ne i}$, $\max_{j \in \mathcal{G}}$ as $\max_i$, $\max_{\tilde{i}}$, $\max_j$ respectively. To prove $\Pr_w(I_2 \ge C(\log(np_0))^{-1/2}) = o_p(1)$, it suffices to prove that

$$\frac{1}{n(n-1)^2} \max_{j \in \mathcal{G}} \sum_{i=1}^n \left\{ \sum_{\tilde{i} \ne i} (\hat{\sigma}_j^{-1} \hat{h}_j - \sigma_j^{-1} h_j) \right\}^2 = o_p((\log(np_0))^{-1}).$$

By the Hölder inequality,

$$\sqrt{\frac{1}{n(n-1)^2} \max_{j \in \mathcal{G}} \sum_{i=1}^n (\sum_{\tilde{i} \ne i} \{\hat{\sigma}_j^{-1} \hat{h}_j - \sigma_j^{-1} h_j\})^2}$$

$$\le \max_{i,\tilde{i},j} |\hat{\sigma}_j^{-1} \hat{h}_j - \sigma_j^{-1} h_j|$$

$$\le \max_{i,\tilde{i},j} |\hat{\sigma}_j^{-1} \hat{h}_j - \hat{\sigma}_j^{-1} h_j| + \max_{i,\tilde{i},j} |\hat{\sigma}_j^{-1} h_j - \sigma_j^{-1} h_j|$$

$$=: I_{21} + I_{22}.$$

Thus it suffices to prove $I_{2k} = o_p((\log(np_0))^{-1/2})$, $k = 1, 2$ respectively. For $I_{21}$, it holds that

$$I_{21} = \max_{i,j} |(X_{ij} - \mathbf{Z}_{ij}^\top \hat{\boldsymbol{\theta}}_j) \hat{\eta}_{ij} - (X_{ij} - \mathbf{Z}_{ij}^\top \boldsymbol{\theta}_j) \eta_{ij}|$$

$$= \max_{i,j} |(X_{ij} - \mathbf{Z}_{ij}^\top \boldsymbol{\theta}_j)(\hat{\eta}_{ij} - \eta_{ij})| + \max_{i,j} |\mathbf{Z}_{ij}^\top(\hat{\boldsymbol{\theta}}_j - \boldsymbol{\theta}_j)(\hat{\eta}_{ij} - \eta_{ij})| + \max_{i,j} |\mathbf{Z}_{ij}^\top(\hat{\boldsymbol{\theta}}_j - \boldsymbol{\theta}_j) \eta_{ij}|,$$

where $\hat{\eta}_{ij} = K - \mathbf{Z}_{ij}^\top \hat{\boldsymbol{\gamma}}_j$, $\eta_{ij} = K - \mathbf{Z}_{ij}^\top \boldsymbol{\gamma}_j$ and $K = 1/2$ or $-1/2$. Note that $\hat{\eta}_{ij} - \eta_{ij} = \mathbf{Z}_{ij}^\top(\boldsymbol{\gamma}_j - \hat{\boldsymbol{\gamma}}_j)$ and

$$\max_{i,j} |\hat{\eta}_{ij} - \eta_{ij}| \le \max_{i,j} |Z_{ij}| \cdot \|\boldsymbol{\gamma}_j - \hat{\boldsymbol{\gamma}}_j\|_1 = O_p\left(s \sqrt{\frac{\log(np_0) \log p}{n}}\right),$$

where the last step follows from the $s\sqrt{\log p / n}$ convergence of $\|\boldsymbol{\gamma}_j - \hat{\boldsymbol{\gamma}}_j\|_1$ and $\max_{i,j} |Z_{ij}| = O_p(\sqrt{\log(np_0)})$, due to the sub-Gaussian properties of $Z_{ij}$. Following the similar arguments, we have

$$\max_{i,j} |(X_{ij} - \mathbf{Z}_{ij}^\top \boldsymbol{\theta}_j)(\hat{\eta}_{ij} - \eta_{ij})| \le \max_{i,j} |X_{ij} - \mathbf{Z}_{ij}^\top \boldsymbol{\theta}_j| \max_{i,j} |\hat{\eta}_{ij} - \eta_{ij}| = O_p\left(s \sqrt{\frac{(\log(np_0))^2 \log p}{n}}\right),$$

$$\max_{i,j} |\mathbf{Z}_{ij}^\top(\hat{\boldsymbol{\theta}}_j - \boldsymbol{\theta}_j)(\hat{\eta}_{ij} - \eta_{ij})| \le \|\hat{\boldsymbol{\theta}}_j - \boldsymbol{\theta}_j\|_1 \max_{i,j} |Z_{ij}| \max_{i,j} |\hat{\eta}_{ij} - \eta_{ij}| = O_p\left(\frac{s^2 \log(np_0) \log p}{n}\right),$$

$$\max_{i,j} |\mathbf{Z}_{ij}^\top(\hat{\boldsymbol{\theta}}_j - \boldsymbol{\theta}_j) \eta_{ij}| \le \|\hat{\boldsymbol{\theta}}_j - \boldsymbol{\theta}_j\|_1 \max_{i,j} |Z_{ij}| \max_{i,j} |\eta_{ij}| = O_p\left(s \sqrt{\frac{(\log(np_0))^2 \log p}{n}}\right).$$

In summary, we have

$$I_{21} = O_p\left(s \sqrt{\frac{(\log(np_0))^2 \log p}{n}}\right) = o_p((\log(np_0))^{-1/2}),$$

where the last equality holds by Assumption A.7. For $I_{22}$, it holds that

$$I_{22} \le \max_j |\hat{\sigma}_j^{-1} - \sigma_j^{-1}| \max_{i,\tilde{i},j} |h_j| = o_p((\log(np_0))^{-1/2}),$$

where the last inequality holds by Lemma A.14 and $\max_{i,\tilde{i},j} |h_j| = O_p(\log(p_0 n))$. In summary, this lemma is proved. $\square$

**Lemma A.19.** *Let $\boldsymbol{X}_1, \dots, \boldsymbol{X}_n$ be $n$ i.i.d random vectors, where $\boldsymbol{X}_i = (X_{i1}, \dots, X_{id})^\top$. Assume that there are some constants $0 < c_1 < C_1$ such that $\mathbb{E}(X_{ij}^2) \ge c_1$ and $X_{ij}$ is sub-Exponential with $\|X_{ij}\|_{\psi_1} \le C_1$ for all $1 \le j \le d$. If $(\log(dn))^7 / n = o(1)$, then*

$$\lim_{n \to \infty} \sup_{t \in \mathbb{R}} \left| \mathbb{P}\left( \left\| \frac{1}{\sqrt{n}} \sum_{i=1}^n \boldsymbol{X}_i \right\|_\infty \le t \right) - \Pr(\|\mathbf{N}\|_\infty \le t) \right| = 0,$$

*where $\mathbf{N} \sim N_d(\mathbf{0}, \mathbb{E}(\boldsymbol{X}\boldsymbol{X}^\top))$.*

*Proof.* Lemma H.6 in [40], and this lemma is adapted from [13]. □

**Lemma A.20.** *Assume that* $\boldsymbol{X} \sim N_d(0, \boldsymbol{\Sigma})$. *Let* $\sigma_j^2 = \Sigma_{jj}$ *and define* $\sigma_{\min} = \min_j \sigma_j$ *and* $\sigma_{\max} = \max_j \sigma_j$. *Then*

$$\sup_{t \in \mathbb{R}} \left| \Pr(\|\boldsymbol{X}\|_\infty \leq t + \epsilon) - \Pr(\|\boldsymbol{X}\|_\infty \leq t) \right| \leq C\epsilon\sqrt{1 \vee \log(d/\epsilon)},$$

*where* $C$ *is a constant depending on* $\sigma_{\min}$ *and* $\sigma_{\max}$.

*Proof.* Lemma H.5 in [40], and this lemma is adapted from [13]. □

**Lemma A.21.** *(A maximal inequality for canonical U-statistics). Let* $\mathbf{X}_1, \ldots, \mathbf{X}_n$ *be* $n$ *i.i.d.* $p$-*dimension random vectors. Let* $f : \mathbb{R}^p \times \mathbb{R}^p \to \mathbb{R}^d$ *be a symmetric and* **canonical** *kernel such that* $\max_{1 \leq m \leq d} \mathbb{E}\{\exp(|f_m|/B_n)\} \leq 2$, *and* $B_n$ *is a sequence of positive reals. Let* $\mathbf{V}_n = \{n(n-1)\}^{-1} \sum_{1 \leq i \neq j \leq n} f(X_i, X_j)$. *If* $2 \leq d \leq \exp(bn)$ *for some constant* $b > 0$, *then there exists a constant* $C(b) > 0$ *such that*

$$\mathbb{E}(\|\mathbf{V}_n\|_\infty) \leq C(b)B_n\{(n^{-1}\log d)^{3/2}\log(nd) + n^{-1}\log d\}.$$

*That is,* $\|\mathbf{V}_n\|_\infty = O_p(B_n\{(n^{-1}\log d)^{3/2}\log(nd) + n^{-1}\log d\})$.

*Proof.* This lemma is adapted from [12]. □

**Lemma A.22.** *Let* $\mathbf{X}_1, \ldots, \mathbf{X}_n$ *be* $n$ *i.i.d.* $p$-*dimension random vectors and* $\mathbf{h} : \mathbb{R}^p \times \mathbb{R}^p \to \mathbb{R}^d$ *be a symmetric and* **nondegenerate** *kernel such that* $\mathbb{E}|h_k(\mathbf{X}_1, \mathbf{X}_2)| < \infty$ *for all* $k = 1, \ldots, d$. *Consider the U-statistic of order two:*

$$\mathbf{U}_n = \frac{1}{n(n-1)} \sum_{1 \leq i \neq j \leq n} \mathbf{h}(\mathbf{X}_i, \mathbf{X}_j).$$

*Let* $\mathbf{T}_n = \sqrt{n}(\mathbf{U}_n - \boldsymbol{\theta})/2$, *where* $\boldsymbol{\theta} = \mathbb{E}\{\mathbf{h}(\mathbf{X}_1, \mathbf{X}_2)\}$ *is the mean of* $\mathbf{U}_n$. *Define* $\mathbf{g}(\mathbf{x}) = \mathbb{E}\{\mathbf{h}(\mathbf{X}_1, \mathbf{X}_2) \mid \mathbf{X}_1 = \mathbf{x}\}$ *and* $\mathbf{x} \in \mathbb{R}^p$. *Let* $e_1, \ldots, e_n$ *be i.i.d.* $N(0, 1)$ *random variables that are independent of* $\{\mathbf{X}_i\}_{i=1}^n$ *and*

$$\mathbf{T}_n^\sharp = \frac{1}{\sqrt{n}} \sum_{i=1}^n \left\{ \frac{1}{n-1} \sum_{j \neq i} \mathbf{h}(\mathbf{X}_i, \mathbf{X}_j) - \mathbf{U}_n \right\} e_i.$$

*Let* $B_n \geq 1$ *be a sequence of real numbers possibly tending to infinity. We assume that:*

*(1) There exists a constant* $\underline{b} > 0$ *such that* $\mathbb{E}\{g_m^2(\mathbf{X}_1)\} \geq \underline{b}$ *for all* $m = 1, \ldots, d$.

*(2)* $\mathbb{E}\{|h_m(\mathbf{X}_1, \mathbf{X}_2)|^{2+\ell}\} \leq B_n^\ell$ *for* $\ell = 1, 2$ *and for all* $m = 1, \ldots, d$.

*(3)* $\|h_m(\mathbf{X}_1, \mathbf{X}_2)\|_{\psi_1} \leq B_n$ *for all* $m = 1, \ldots, d$.

*Denote* $a_{\|\mathbf{T}_n^\sharp\|_\infty}(\alpha)$ *as the* $\alpha$th *conditional quantile of* $\|\mathbf{T}_n^\sharp\|_\infty$ *given* $\mathbf{X}_1, \ldots, \mathbf{X}_n$, *where* $\alpha \in (0, 1)$. *If* $(\log(dn))^7/n = o(1)$, *then*

$$\lim_{n \to \infty} \sup_{\alpha \in (0,1)} \left| \Pr(\|\mathbf{T}_n\|_\infty \leq a_{\|\mathbf{T}_n^\sharp\|_\infty}(\alpha)) - \alpha \right| = 0. \tag{1.51}$$

*In additional, assume that there exist statistics* $\hat{\mathbf{T}}_n$ *and* $\hat{\mathbf{T}}_n^\sharp$ *such that*

$$\Pr(\left|\|\hat{\mathbf{T}}_n\|_\infty - \|\mathbf{T}_n\|_\infty\right| \geq \xi(n)) \to 0,$$

*and*

$$\Pr_e(\left|\|\hat{\mathbf{T}}_n^\sharp\|_\infty - \|\mathbf{T}_n^\sharp\|_\infty\right| \geq \xi(n)) \to 0 \text{ in probability.}$$

*for some* $\xi(n)$ *depending on* $n$. *If* $\xi(n)\sqrt{1 \vee \log(d/\xi(n))} = o(1)$, *then*

$$\lim_{n \to \infty} \sup_{\alpha \in (0,1)} \left| \Pr(\|\hat{\mathbf{T}}_n\|_\infty \leq a_{\|\hat{\mathbf{T}}_n^\sharp\|_\infty}(\alpha)) - \alpha \right| = 0, \tag{1.52}$$

*where* $a_{\|\hat{\mathbf{T}}_n^\sharp\|_\infty}(\alpha)$ *is the* $\alpha$th *conditional quantile of* $\|\hat{\mathbf{T}}_n^\sharp\|_\infty$ *given* $\mathbf{X}_1, \ldots, \mathbf{X}_n$.

*Proof.* This lemma is adapted from [12] and [13]. □

