# OpenReview forum: "Robust group and simultaneous inferences for high-dimensional single index model"
_NeurIPS.cc/2024/Conference — NeurIPS 2024 poster_

### Official Review · Reviewer_Tg8m · 2024-06-29

**Soundness:** 3
**Presentation:** 3
**Contribution:** 3
**Rating:** 5
**Confidence:** 2

**Summary:**

This paper studies the high-dimensional single index model (SIM), which takes the form $Y=g(X^T\beta, \epsilon)$ with $\epsilon$ and $X$ being orthogonal. Although this model has flexibility and interpretability, its efficiency is adversely affected by outlying observations and heavy-tailed distributions. The paper improves this in the following 3 aspects:

(1) they extend the rank-LASSO procedure to include both convex and non-convex penalties and establish error bound of any local optimum of the empirical objective;

(2) they provide asymptotically honest group inference procedures based on the idea of orthogonalization for testing the joint effect of many predictors;

(3) they develop a multiple testing procedure for determining if the individual coefficients are relevant simultaneously, and show that it is able to control the FDR asymptotically.

**Strengths:**

Provides new ideas that deal with the inefficiency of single index model due to outlying observations and heavy-tailed distributions.

**Weaknesses:**

I think the paper is quite well-written. The only issue may be that the authors can provide more about the background of the problem,

**Questions:**

N/A

---

> ### Author Rebuttal · Authors · 2024-08-05
>
> Thanks very much for your valuable comments.
> ## Weaknesses
>
> The group inference is helpful to decide whether a group of predictors are important or not for the response. If we find a group of predictors are important, we would like to know which specific predictors in the group are significant. For this aim, our developed multiple testing procedure is useful. For instance, researchers may aim to test whether a gene pathway, consisting of high-dimensional genes for the same biological function, is important for a certain clinical outcome, given the other high-dimensional genes. When determining that a certain gene pathway is important, researchers need to further identify specific genes within the pathway which are important for a certain clinical outcome.
>
> According to your helpful suggestion, we will add a real data analysis in the appendix of the revision. For your convenience, we also copy the real data analysis in the following:
>
> We apply our methods on a dataset about riboflavin (vitamin B2) production rate with Bacillus Subtilis. This dataset is made publicly by Buhlmann et al. (2014) and has been analyzed by many authors, for instance Meinshausen et al. (2009), Van de Geer et al. (2014), Javanmard and Montanar (2014), and Fei et al. (2019). The dataset riboflavin can be obtained from the R package $\texttt{hdi}$. It consists of $n = 71$ observations of strains of Bacillus Subtilis and $p = 4088$ covariates, measuring the log-expression levels of 4088 genes. The response variable is the logarithm of the riboflavin production rate.
>
> Our goal is to detect which genes are associated with riboflavin production rate. Like most existing studies, we first reduce ultrahigh-dimension to a moderate high-dimension. Here we pick out first 300 genes by distance correlation based screening Li et al. (2012). We first conduct global testing on these 300 genes. The $p$-value of our group inference procedure is 1.29e-04, indicating that the null hypothesis is rejected and the selected 300 genes are influential for riboflavin production rate. Next, we further use FDR control procedure to select the important genes in these 300 genes.
>
> By implementing our proposed FDR control procedure with the FDR level of 0.1, we identify 10 genes that are significantly associated with the response. That is
>
> $G_{I}$ = {YTGB\_at, YCKE\_at, YXLE\_at, YXLD\_at, YJCJ\_at, XHLA\_at, xepA\_at, YCGO\_at, RPLP\_at, XKDS\_at}.
>
> If the FDR level is set as 0.2, 5 more genes will be selected. That is
>
> $G_{II}$ = $G_I$ $\cup$ {SPOIISA\_at, YHCB\_at, XKDI\_at, YJCF\_at, XHLB\_at}.
>
> We further conduct group inference on the selected subsets $G_{I}, G_{II}$ and their  complement sets $G_I^c, G_{II}^c$. As expected, our group inference procedure finds again that $G_{I}, G_{II}$ are significant while $G_{I}^c, G_{II}^c$ are not.
> The corresponding $p$-values of $G_{I}$, $G_{I}^c$, $G_{II}$ and $G_{II}^c$ are 6.75e-06, 7.26e-01, 9.33e-06 and 9.37e-01, respectively. These results suggest that the genes selected by the FDR control procedure are really influential.
>
> We compare these selected genes with other methods. For example, the multi-sample-splitting method proposed in Meinshausen et al. (2009) identified YXLD\_at; Van de Geer et al. (2014) did not select any gene using the de-sparsified Lasso; Javanmard and Montanar (2014) only selected two genes: YXLD\_at and YXLE\_at and Fei et al. (2019) claimed YCKE\_at, XHLA\_at, YXLD\_at, YDAR\_at and YCGN\_at as significant. From $G_I$ and $G_{II}$, we can see clearly that the gene YXLD\_at is detected by not only Meinshausen et al. (2009), Javanmard and Montanar (2014), Fei et al. (2019), but also by our procedure. Besides, the genes YCKE\_at, YXLE\_at and XHLA\_at which are detected by Javanmard and Montanar (2014) and Fei et al. (2019), are also found by our method. Further, our procedure detects some additional important genes.

---

> > ### Comment · Reviewer_Tg8m · 2024-08-10
> > **keep the score**
> >
> > Thanks for the rebuttal. I tend to keep the score.

---

### Official Review · Reviewer_BqNs · 2024-07-12

**Soundness:** 3
**Presentation:** 3
**Contribution:** 3
**Rating:** 6
**Confidence:** 4

**Summary:**

This paper proposes a robust group hypothesis testing procedure for a high-dimensional single index model based on a data-driven transformation of the response variable. The key observation is that under the the linearity condition (LC) of the predictors, a wide class of single index models can be equivalently (for testing purposes) converted into a linear model with a transformed response variable. The main contribution of this paper is the introduction of the distribution transformation (a data-driven approach) of the response variable, and derive the asymptotic distribution of the resulting test statistic. Numerical performance of the proposed method is evaluated through simulation studies.

**Strengths:**

The proposed method is interesting and potentially useful. The paper is well-written and the presentation is clear.

**Weaknesses:**

The simulation study can be improved.

**Questions:**

1. The introduction of the distribution function $F_n$ in (2.7) is interesting but also a little arbitrary. I suspect that any other functions that can be written as linear combinations of functions of $Y_i$'s can be used in (2.7), for example, the kernel density estimator, the kernel smoother of $Y_i$'s. The resulting test statistic should still be asymptotically equivalent to a U-statistic. Could the authors comment on this?

2. Whenever the term "robustness" is involved, it implies loss of efficiency in some cases. In my opinion, it is important to make a transparent evaluation of the limitations of the proposed procedure. In the simulation study, one can add a comparison to the method with $h(Y)=Y$ as described in section 3.3. It will be informative if one can showcase in which cases which procedure is more powerful, and thus understand which method to use in practice.

3. In the simulation study, I would also suggest a graphical comparison between different methods by gradually increasing $p_{out}$ from $0$ to, say, $0.5$ for a more complete picture.

**Limitations:**

The simulation study can be improved.

---

> ### Author Rebuttal · Authors · 2024-08-05
>
> Thanks very much for your valuable comments.
> ## Weaknesses
> We have added more simulations to illustrate our procedures and compare them with other methods. The simulation settings are summarized in global response, and the simulated results are displayed in the attached pdf file.
> ## Question 1
> As noted by you, there are many possible choices for the transformation function $h(Y)$. There are several reasons for us to choose the distribution function as the transformation function:
> * Firstly, the response-distribution transformation function is bounded. Actually with the equation (2.3), given the widely imposed sub-gaussian assumption on the predictors, any bounded transformation function $h(Y)$ would lead the transformed error term $e$ being sub-gaussian, even if the original error term $\epsilon$ in the single index model comes from Cauchy distribution.
> * As noted by Rejchel and Bogdan (2020), in the empirical distribution function, the term $\sum_{j=1}^n I(Y_j \leq Y_i)$ is the rank of $Y_i$. Since statistics with ranks such as Wilcoxon test and the Kruskall-Wallis ANOVA test, are well-known to be robust, this then intuitively explains why our procedures with distribution function are robust with respect to outliers in response.
> * The distribution function is very easy to estimate and thus our approach is straightforward to implement and understand. While the kernel density estimator, the kernel smoother of $Y_i$’s require additionally tuning bandwidth.
>
> ## Question 2
> According to your valuable suggestion, we have conducted detailed comparisons with the method based on $h(Y) = Y$ as described in section 3.3. The simulation settings are summarized in global response, and the corresponding numerical results are displayed in Figure R.1 in the attached pdf file.
>
> * In Figure R.1, we compare our method with the procedure based on $h(Y)=Y$ when the error term follows the standard normal distribution. In the standard case (normal distribution error, linear model and no response polluted), the performance of test statistic with $h(Y)=Y$ is slightly better than our method with $h(Y)=F(Y)$. However, in other settings, our method with $h(Y)=F(Y)$ performs much better than method with $h(Y)=Y$.
>
> Although $h(Y)=Y$ performs well in the standard case, one cannot know the distribution of the error term or the formula of the model in practice. Thus compared to $h(Y)=Y$, we recommend using our procedure in practice.
> ## Question 3
> We agree that a graphical comparison between different methods by gradually increasing $p_{out}$ from 0 to 0.5 would provide a more complete picture. According to your valuable suggestion, we have conducted detailed simulations for comparisons. The simulation settings are summarized in global response, and the results are displayed in Figure R.1 of the attached pdf file.
>
> We consider three transformation procedures: (1) $h(Y) = F(Y)$ (Our method); (2) $h(Y) = Y$; (3) $h(Y) = \text{sigmoid}(Y) = 1/\\{1+\exp(-Y)\\}$. We vary $p_{out}$ from $\\{0, 0.1, 0.2, 0.3, 0.4, 0.5\\}$ and the simulation results are shown in Figure R.1.
> * Firstly, under the null hypothesis ($G_1$), $h(Y) = \text{sigmoid}(Y)$ cannot control the type I error for Model 2, while other procedures perform well.
> * Secondly, under the alternative hypothesis ($G_2$), the empirical powers of other procedures decrease rapidly with the increase of $p_{out}$, while the powers of our procedure remain stable, which is particularly noticeable when $\delta = 0.5$. This finding indicates that our method has strong robustness when the responses are polluted.
> * Thirdly, our method performs well for both Model 1 and Model 2, indicating that our method is robust across different single-index models.
>
> ## Limitations
> Following your suggestions, we have added more simulations to illustrate our procedures and compare them with other methods. The simulation settings are summarized in global response, and the simulated results are displayed in Figure R.1 of the attached pdf file.

---

> > ### Comment · Reviewer_BqNs · 2024-08-12
> > **response to rebuttal**
> >
> > I want to thank you for addressing my concerns. They are very helpful. Congratulations for a solid work!

---

### Official Review · Reviewer_wCvo · 2024-07-14

**Soundness:** 3
**Presentation:** 3
**Contribution:** 3
**Rating:** 6
**Confidence:** 3

**Summary:**

The paper proposes an algorithm for group inference in single-index models with an unknown link function, that is robust to heavy-tailed noise in the responses. The central idea of the approach is based on the property of elliptical distributions that the linear input-output correlations remain along a fixed direction for any transformation of the labels. The algorithm introduces a linear estimation objective for transformed labels resulting in an estimator converging to the true parameters under sparsity assumptions. Crucially, the obtained estimator yields a robust test for group inference satisfying the orthogonality property, which relaxes the requirements of separation between zero and non-zero coefficients. The obtained test further satisfies honesty, namely the uniform convergence of Type 1 error. The work concludes with numerical tests of the proposed approach.

**Strengths:**

The paper is generally well written and provides an adequate motivation for the problem of group inference under heavy-tailed response, and the idea of orthogonalization. The proposed approach is straightforward to implement and understand. The paper provides an extensive discussion of some important properties of the test such as honesty and power. The estimation error bounds are non-asymptotic and hold for general scalings of p,n. The numerical experiments support the efficiency and robustness of the approach.

**Weaknesses:**

A few limitations of the setup are not explicitly discussed:
1) The robustness is restricted to variability in the noise of the response function, not w.r.t general perturbations to the distribution. The paper also doesn’t provide quantitative bounds on robustness.
2) The condition kappa_h \neq 0 excludes even link functions and in particular the problem of sparse phase retrieval.
3) Theorem 3.1 imposes limitations on the sparsity level that should be explicit. The results in the paper seems to be restricted to constant sparsity levels/

The choice of h=F for robustness appears to be due to F(Y) being uniformly distributed in [0,1] independent of the distribution of Y. This should be more discussed explicitly.

**Questions:**

Questions:

Why are λX, λY not directly described in Theorem 3.1?

Why is the transformation H=F particularly suitable for robustness? Is it because of the distribution of F(Y) being agnostic to the distribution of Y?

For sub-exponential variables, Lemma A.11 appears to be weaker than Bernstein’s inequality.

Could you clarify the difference w.r.t Bernstein’s inequality and related inequalities for random variables with bounded Orlicz norms?

**Limitations:**

Some missing discussion on limitations is described above under “weaknesses”. The work is primarily of a theoretical nature and has no potential negative societal impact.

---

> ### Author Rebuttal · Authors · 2024-08-05
>
> Thanks very much for your valuable comments.
> ## Weakness 1
> As noticed by you, in our paper, we say our procedure is robust since our methods do not need any moment condition for the error term in the single-index model. For your concern, we further discuss the robustness of our procedure based on the tool of efficient influence function. Recall that our test procedure is inspired by the quantity $$I = \Psi(P) = E_{P}[\\{F_{P}(Y) - 1/2 - Z_j^\top\gamma_j\\}(X_j - Z_j^\top\theta_j)],$$where $P$ is distribution of $(X_j,Z_j^\top,Y)^\top$. Next we derive the efficient influence function (EIF) of $I$. Consider $$P_t=t\tilde P+(1-t)P,$$where $t\in[0,1]$, and $\tilde P$ is a point mass at a single observation $\tilde o  = (\tilde x_j,\tilde z_j^\top,\tilde y)^\top$. By some calculation, the EIF for $I$ at observation $\tilde o$ is$$\phi(\tilde o,P)=\frac{d\Psi(P_t)}{dt}\vert_{t=0}=\\{F_P(\tilde y)-1/2-\tilde z_j^\top\gamma_j\\}(\tilde x_j-\tilde z_j^\top\theta_j)+E_P[\\{I(Y\geq\tilde y)-F_p(Y)\\}(X_j-Z_j^\top\theta_j)]-\Psi(P). $$ Since $I(Y\geq\tilde y)$ and $F_P(\tilde y)$ are bounded, then given $(x_j,z_j^\top)^\top$, $\phi(\tilde o,P)$ is bounded for any $\tilde y\in R$. In terms of the EIF, our test statistics are robust with respect to the perturbations in the responses.
> ## Weakness 2
> We agree with you. The condition $\kappa_h\neq0$ does exclude even link functions and in particular the problem of sparse phase retrieval. Neykov et al. (2020) introduced a novel procedure to deal with the problem of sparse phase retrieval when $\kappa_h=0$. We will incorporate their insights in the reivision.
> ## Weakness 3
> For consistency in $L_2$-loss, we require the sparsity level satisfying $s_Y=o(n/\log p)$. For $L_1$-loss, the restriction becomes $s_Y=o(\sqrt{n/\log p})$. The sparsity level is allowed to be diverging with $n$ and $p$.
> ## Weakness 4
> With the equation (2.3), given the widely imposed sub-gaussian assumption on the predictors, any bounded transformation function $h(Y)$ would lead the transformed error term $e$ being sub-gaussian, even if the original error term $\epsilon$ comes from Cauchy distribution. However, the response-distribution transformation is preferred due to the following additional reasons.
>  * As noted by Rejchel and Bogdan (2020), in the empirical distribution function, the term $\sum_{j=1}^n I(Y_j \leq Y_i)$ is the rank of $Y_i$. Since statistics with ranks such as Wilcoxon test and the Kruskall-Wallis ANOVA test, are well-known to be robust, this then intuitively explains why our procedures with distribution function are robust with respect to outliers in response.
> * The distribution function is very easy to estimate, and thus, our approach is straightforward to implement and understand.
> * Lastly, besides being robust, the choice of $h(Y) = F(Y)$ would also have relatively high efficiency. Specifically, we conduct detailed simulations to illustrate this point. The simulation settings are summarized in global response, and the simulated results are displayed in the attached pdf file. Besides distribution function, we also consider $h(Y) = Y$ and $h(Y) = \text{sigmoid}(Y) = 1/\\{1+\exp(-Y)\\}$. Compared to other functions, our procedure has high power performance under nearly all the settings. The better performance compared to $\text{sigmoid}(Y)$ demonstrates that the superiority of our procedure is not merely due to the boundedness of the transformation function.
> ## Question 1
> To simplify the presentation, all main assumptions are given in the A.5 of the Appendix. In Assumption A.1 (ii), we assume that $c\sqrt{\log p/n}\leq \lambda_{X},\lambda_{Y}\leq C\sqrt{\log p/n}$ for some constants $0<c\leq C$.
> ## Question 2
> Please refer to response to weakness 4 for details.
> ## Question 3
> After some careful calculation, we agree with you that for sub-Exponential variables, Lemma A.11 is weaker than Bernstein’s inequality. Let $X_1,\ldots,X_n$ be independent, mean zero, sub-Exponential random variables. Then for $t\geq 0$, Bernstein inequality implies that$$P\\{\vert\frac{1}{n}\sum_{i=1}^{n}X_{i}\bigr\vert\geq t\\}\leq 2\exp\\{-L\min(\frac{t^{2}}{K^{2}},\frac{t}{K})n\\},$$where $L$ is a constant and $K = \max_{1\leq i\leq n}\lVert X_{i}\rVert_{\psi_1}$. And Lemma A.11 implies that$$P\\{\vert\frac{1}{n}\sum_{i=1}^{n}X_i\vert\geq t\\}\leq 4\exp\\{-\frac{1}{8}n^{1/3}t^{2/3}\\} + 4n L_{1}\exp\\{-\frac{L_{2}}{2}n^{1/3}t^{2/3}\\}$$for $t\geq \sqrt{8E(X_i^2)/n}$, where $L_{1}$ and $L_{2}$ are constants. For ease of comparsion, suppose that $t \asymp n^{-c}$ for $c\in(0,1/2]$. Note that $$2\exp\\{-\frac{Lt^{2}n}{K^{2}}\\} \ll  4n L_{1}\exp\\{-\frac{L_{2}}{2}n^{1/3}t^{2/3}\\}$$when $n$ is sufficiently large. That is, the bound of Bernstein's inequality is sharper than the bound of Lemma A.11.
> ## Question 4
> Let $g:[0,\infty)\rightarrow[0,\infty)$ be a non-decreasing convex function with $g(0)=0$. The ``$g$-Orlicz'' norm of a random variable $X$ is $\Vert X\Vert_{g} := \inf\\{\eta>0:E[g(\vert X\vert/\eta)]\leq 1\\}$.
>
> Specifically, let $\psi_r(x)=\exp(x^r) - 1$. The sub-Weibull norm of a r.v. $X$ for any $r>0$ is defined as $\Vert X\Vert_{\psi_{r}}$. The r.v. $X$ follows sub-Weibull distribution for $r>0$ is equivalent to $\Vert X\Vert_{\psi_r}$ is bounded.
> When $r=1$, $X$ is sub-Exponential. In this case, Bernstein inequality is applicable when $r=1$. While Lemma A.11 is applicable for $r>0$. Therefore, Lemma A.11 is a generalization of Bernstein's inequality for general sub-Weibull distributions.
>
> In this article, Lemma A.11 is adopted in the proof of Lemma A.14. In this case, we need to derive the bound of $\max_{j\in\mathcal{G}}\vert\sigma_{j}^{2} - \tilde\sigma_j^2\vert$. For $j\in\mathcal{G}$, $\sigma_j^2-\tilde\sigma_j^2$ can be represented as $\sigma_j^2-\tilde\sigma_j^2=\frac{1}{n}\sum_{i=1}^{n}Z_i$, where $Z_1,\ldots,Z_n$ are zero mean, i.i.d. variables with bounded sub-Weibull norm for $r=1/2$.

---

### Author Rebuttal · Authors · 2024-08-05

Dear Program Chairs, Senior Area Chairs, Area Chairs and Reviewers,

Thank all of you for your insightful comments and valuable suggestions, which have significantly enhanced the quality of this work. In this response, all the comments have been carefully addressed and accommodated. A summary of response is as follows.

* Discussions of robustness with respect to perturbations in the responses. In the revision, we carefully discuss the robustness of our inference procedure based on the tool of efficient influence function. Please see the response to weakness 1 of Reviewer wCvo for details.
* Discussions of the response-distribution transformation function. In the revision, we discuss the reasons why the response-distribution transformation is preferred. Please see responses to weakness 4 of Reviewer wCvo and question 1 of Reviewer BqNs for details.
* Relationships between Lemma A.11 and Bernstein' inequality. Please see responses to questions 3 and 4 of Reviewer wCvo for details.
* Additional simulation studies. In the revision, we have conducted additional simulation studies to illustrate the robustness of our procedure. Please refer to the attached pdf file for details.

    * **The simulation settings are summarized as follows.** We consider the following two models:

         * Model 1: Linear model: $Y = X^\top\beta + \epsilon$.

         * Model 2: Non-linear model: $Y = \exp(X^\top\beta + \epsilon)$.

         The regression coefficients $\beta = (\beta_{1},\beta_{2},\ldots,\beta_{p})^\top$ are generated as $\beta_{j} = \delta$ for $j = 1,\ldots,6$ and $\beta_{j}=0$ otherwise, where $\delta$ can be regarded as a signal strength parameter. We generate the error term from the standard normal distribution. We add outliers to pollute the observations: $p_{out}$ of the responses are picked at random and increased by $m_{out}$-times maximum of original responses. Specifically, the detailed settings for the above parameters are as follows. Firstly, we consider the sample size $n=500$ and the dimension $p=800$. Secondly, we set the signal strength parameter $\delta$ to vary from $\\{0.1,0.3,0.5\\}$. Thirdly, we fix $m_{out}$ to 10. Lastly, we vary $p_{out}$ from $0$ to $0.5$ in increments of $0.1$. For more complete comparisons, we consider three transformation procedures: (1) $h(Y) = F(Y)$ (Our method); (2) $h(Y) = Y$; (3) $h(Y)=\text{sigmoid}(Y)=1/\\{1+\exp(-Y)\\}$. Other simulation settings are the same as described in section 4 of the main text. The simulation results are summarized in Figure R.1 of the attached pdf file.

    * Figure R.1 summarizes the results of empirical type I error and empirical power with the significant level of $\alpha=0.05$ for different methods when the error term follows the standard normal distribution. Firstly, under the null hypothesis ($G_1$), $h(Y) = \text{sigmoid}(Y)$ cannot control the type I error for Model 2, while other procedures perform well. Secondly, under the alternative hypothesis ($G_2$), the empirical powers of other procedures decrease rapidly with the increase of $p_{out}$, while the powers of our procedure remain stable, which is particularly noticeable when $\delta = 0.5$. This finding indicates that our method has strong robustness when the responses are polluted. Thirdly, our method performs well for both Model 1 and Model 2, indicating that our method is robust across different single-index models.

* Discussion about the background of the problem. In the response, we discuss the practical background of the problem and demonstrate the practical value by a real data analysis. Please see the response to weaknesses of Reviewer Tg8m for details.
* All the other comments from the Program Chairs, Senior Area Chairs, Area Chairs and Reviewers are also carefully addressed.

Thank you very much for giving us an opportunity to revise the paper.

---

### Comment · Area_Chair_M66z · 2024-08-08

Dear authors, dear reviewers,

the discussion period has begun as the authors have provided their rebuttals.
I encourage the reviewers to read all the reviews and the corresponding rebuttals: the current period might be an opportunity for further clarification on the paper results and in general to engage in an open and constructive exchange.

Many thanks for your work.
The AC

---

### Decision · Program_Chairs · 2024-09-25

**Decision:**

Accept (poster)

**Comment:**

The focus of the paper is an algorithm for group inference in single-index models which is robust under heavy-tailed noise in high dimension: under some sparsity hypothesis, an equivalent linear estimation objective allows to obtain a robust estimator converging to the true parameters.

The paper received positive feedback from all reviewers because of its clarity and the proposed ideas to deal with the problem of group inference under heavy tails in the response in single-index models. The results are further supported by numerical evidence. During the rebuttal discussion, limitations in the analyzed set-up were addressed and clarified in the resubmitted version, and further numerical evidence was added to the manuscript. I share positive evaluations of the Reviewers and recommend therefore acceptance.